# Dynamic changes in the regulatory T-cell heterogeneity and function by murine IL-2 mutein

Daniel R Lu[2], Hao Wu[1], Ian Driver[2], Sarah Ingersoll[1], Sue Sohn[1], Songli Wang[2], Chi-Ming Li[2], Hyewon Phee[1]

**The therapeutic expansion of Foxp3[+] regulatory T cells (Tregs) shows promise for treating autoimmune and inflammatory disorders. Yet, how this treatment affects the heterogeneity and function of Tregs is not clear. Using single-cell RNA-seq analysis, we characterized 31,908 Tregs from the mice treated with a half-life extended mutant form of murine IL-2 (IL-2 mutein, IL-2M) that preferentially expanded Tregs, or mouse IgG Fc as a control. Cell clustering analysis revealed that IL-2M specifically expands multiple sub-states of Tregs with distinct expression profiles. TCR profiling with single-cell analysis uncovered Treg migration across tissues and transcriptional changes between clonally related Tregs after IL-2M treatment. Finally, we identified IL-2M–expanded Tnfrsf9[+]Il1rl1[+] Tregs with superior suppressive function, highlighting the potential of IL-2M to expand highly suppressive Foxp3[+] Tregs.**

## Introduction

Foxp3[+] regulatory T cells (Tregs) play a fundamental role in immunosuppression and immune tolerance, and there is great interest in harnessing Treg populations to treat autoimmune and inflammatory disorders. The differential expression of transcription factors, costimulatory receptors, chemokine receptors, and secreted effectors in quiescent and activated Tregs suggests that the heterogeneous Treg states exist and perform distinct functions (Zheng et al, 2006; Menning et al, 2007; Schiering et al, 2014). Moreover, nonlymphoid tissue Tregs acquire unique phenotypes different from lymphoid-tissue Tregs, suggesting that the anatomical location of Tregs contributes to their heterogeneity (Sather et al, 2007; Miragaia et al, 2019).

Recently, low-dose Interleukin-2 (IL-2) therapies have been tested to induce tolerance in patients with autoimmunity and inflammatory disorders (Koreth et al, 2011; Saadoun et al, 2011;

Hartemann et al, 2013; Matsuoka et al, 2013; Klatzmann & Abbas, 2015; Yu et al, 2015). Although the low-dose IL-2 therapies expand Tregs, their effect has been limited by concomitant increases in conventional effector T cells and natural killer cells. To improve selectivity and pharmacokinetics of low-dose IL-2, alternative modalities have been considered (Peterson et al, 2018). However, it is not clear how IL-2–based therapies impact Treg heterogeneity in diverse tissues. Because the goal of Treg-targeted therapies is to expand Treg-mediated tolerance at proper anatomical locations, it is critical to understand how IL-2–mediated expansion impacts the phenotypic and functional heterogeneity of Tregs in lymphoid and nonlymphoid tissues.

Thymic-derived Foxp3+ Tregs undergo TCR-dependent antigen priming and activation-induced expansion in lymphoid organs followed by extravasation into peripheral tissues, where they acquire tissue-specific tolerogenic phenotypes. Given the complex migratory patterns of Tregs, it is unclear how IL-2–mediated therapy affects Tregs within and across tissues. TCR sequencing combined with single-cell profiling provides an opportunity to measure IL-2–induced Treg differentiation and movement by tracing the transcriptional conversions and trafficking patterns of clonal lineages.

To better understand the impact of the IL-2–mediated Treg expansion therapy on Foxp3[+] Treg heterogeneity in lymphoid and nonlymphoid tissues, we profiled mouse spleen, lung, and gut Tregs using single-cell RNA-seq (scRNA-seq) with TCR sequencing under murine IL-2 mutein (IL-2M) stimulation or homeostatic (mouse IgG Fc isotype control–treated) conditions. Comparison of resting, primed/activated, and activated Treg states from different tissues revealed unique gene signatures shared between spleen and lung Tregs, as well as distinct activation profiles of gut Tregs. Administration of murine IL-2M dramatically changed the landscape of Tregs in the spleen and the lung, although maintaining tissue-specific identity in the gut. TCR profiling coupled with scRNA-seq revealed gene expression dynamics governing Treg differentiation after IL-2M treatment and uncovered a migratory axis across tissues. In addition, we identified a population of activated Tnfrsf9[+]Il1rl1[+] Tregs in mice that expands after IL-2M and

[1]Department of Oncology and Inflammation, Amgen Research, Amgen Inc, South San Francisco, CA, USA    [2]Genome Analysis Unit, Amgen Research, Amgen Inc, South San Francisco, CA, USA

Correspondence: hyewonp@amgen.com; chimingl@amgen.com
Hao Wu's present address is Pharmacyclics, Sunnyvale, CA, USA
Ian Driver's present address is Gordian Biotechnology, San Francisco, CA, USA
Sarah Ingersoll's present address is Nektar therapeutics, San Francisco, CA, USA

suppresses convention T cells robustly in vitro. Overall, our experiments provide new insights into the relationships between Foxp3[+] Treg activation states and their phenotypic heterogeneity in different tissues during homeostasis and after murine IL-2M stimulation.

# Results

## A half-life–extended mutant form of murine IL2 expands CD25[+]Foxp3[+] Tregs in vivo

To determine the specific role of mouse IL-2 in Foxp3[+] Tregs in mice, a half-life–extended mutant form of murine IL-2 (IL-2 mutein, IL-2M) was generated as a mouse IgG2a Fc fusion protein (Fig S1A). Previously, a human form of long-lived IL2 mutein (human IgG-(human IL-2N88D)$_2$) was reported (Peterson et al, 2018). In this human IL-2 mutein, an effector-silent human IgG1 was fused to a mutant form of human IL2 to increase the half-life. Moreover, the N88D mutation was introduced to human IL2 to decrease its binding to the intermediate affinity IL2 receptor, IL2$\beta\gamma$, whereas maintaining its binding to the high-affinity IL2 receptor, IL2$\alpha\beta\gamma$. For the mouse IL-2 mutein, an effector-silent mouse IgG2a Fc (N297G) (Tao & Morrison, 1989) was fused to a mutant form of IL2 to increase the half-life. Furthermore, D34S and N103D mutations were introduced to the mouse IL2 because both amino acids were described to be critical for IL2's binding to IL2R$\beta$, whereas minimally affecting interaction with IL2R$\alpha$ (Zurawski & Zurawski, 1989; Zurawski et al, 1993). The N103 residue of mouse IL2 corresponds to the N88 of human IL2. In addition to D34S and N103D mutations, two additional mutations (C140A and P51T) were introduced to mouse IL2 to facilitate manufacturability (Klein et al, 2017).

Intracellular staining of phosphorylated STAT5 as a response to IL2 signaling was performed to determine activity of mouse IL-2M in vitro (Fig S1B and C). As expected, IL-2M induced phopsho-STAT5 in CD25+Foxp3+ Tregs that express the high-affinity IL2R$\alpha\beta\gamma$ in a dose-dependent manner, whereas it did not induce phospho-STAT5 in CD25-Foxp3– Tconv cells that express the intermediate IL2R, IL2R$\beta\gamma$ (Fig S1B, *top panel*). In contrast, wild-type recombinant mouse IL2 (rmIL2) induced phospho-STAT5 in CD25-Foxp3– Tconv cells as well as CD25+Foxp3+ Tregs (Fig S1B, *bottom left panel*). Because the mutations introduced were intended to reduce the interaction of mouse IL2 with IL2R$\beta$ but maintain interaction with IL2R$\alpha$ (CD25) to allow IL-2M's binding to the high-affinity IL2R (ILR$\alpha\beta\gamma$) on Tregs, the in vitro activity of IL-2M was attenuated compared with wild-type rmIL2. Furthermore, IL-2M slighted activated phospho-STAT5 in CD25+Foxp3– effector Tconv cells, although the degree of activation was markedly reduced compared with wild-type rmIL2 (Fig S1B, *bottom right panel*).

Administration of murine IL-2M to C57BL/6 mice with different doses revealed that 0.33 mg/kg of murine IL-2M specifically increased Foxp3[+] Tregs in the spleen, lymph nodes, blood, and lung, whereas it did not affect CD25[–]Foxp3[–] T conventional cells (Tconv) (Figs 1A–D and S1D). Murine IL-2M increased percentages and cell numbers of CD25[+]Foxp3[–] activated CD4 T cells slightly at a higher dose (Fig 1A), likely because of the presence of high-affinity IL2R (IL2R$\alpha\beta\gamma$) in CD25-expressing cells (Malek, 2008; Letourneau et al, 2009). Expansion of Foxp3[+] Tregs by IL-2M was comparable with

IL-2/anti-IL2 antibody (JES6-1) conjugate (IL-2C), which was previously reported to expand Tregs (Boyman et al, 2006) (Fig S2A–D).

In vivo–expanded Tregs by IL-2M displayed superiority in suppressing the proliferation of Th1 effector cells (Fig 2A) and generation of IFN$\gamma$ from Th1 effector cells in vitro compared with Foxp3[+] Tregs isolated from spleens of PBS-treated mice (Fig 2B). Although these results revealed that highly suppressive Tregs expanded after IL-2M treatment in vivo, it was not clear how IL-2M impacted on the heterogeneity of Tregs toward functional suppression in lymphoid and nonlymphoid tissues. We, therefore, sought to profile the molecular phenotypes expressed by Tregs to understand how IL-2M elevates Treg expansion and suppression in mice.

## Isolation and scRNA-seq of Tregs from the spleen, lung, and gut

The therapeutic success of Treg expansion depends on the efficacy of Treg immunosuppression at specific anatomical sites. To characterize the heterogeneity of Tregs and assess how IL-2M reshapes this diversity, we performed scRNA-seq on Foxp3[+] Tregs from multiple tissues at steady state and after IL-2M treatment. CD4[+]eGFP[+] single cells were sorted from the spleen, lung, and gut of Foxp3-eGFP mice and single cell libraries were prepared using the 10× Chromium platform (Fig 3A and Table S1). After filtering and cross-sample normalization using Seurat (Butler et al, 2018), we recovered 17,097 spleen Tregs, 10,353 lung Tregs, and 4,458 gut Tregs across three replicates (except gut Tregs treated with IL-2M [n = 2]) with roughly equivalent Tregs in mouse IgG Fc isotype control (Iso)–treated and IL-2M–treated (IL-2M) conditions (16,152 and 15,756 cells, respectively). Single-cell molecule and gene recovery rates showed good concordance between Iso- and IL-2M–treated cells (Fig S3A and B). To establish that we had accurately sorted and sequenced Tregs, we additionally performed scRNA-seq on CD4 conventional T cells sorted using FACS. Comparison of Tregs and Tconvs confirmed that Tregs and Tconvs expressed distinct transcriptional profiles and clustered into distinct groups (Fig S4A). In addition, Tregs expressed higher transcript levels of established Treg genes such as *Foxp3*, *Il2ra*, *Ctla4*, *Ikzf2*, and *Nrp1*, whereas both cell types expressed similar levels of *Cd4* (Fig S4B and C).

## Molecular characterization of resting, primed, and activated Treg cell states across all tissues

Previous scRNA-seq profiling efforts have identified resting, early-activated (primed/activated), and terminal effector subsets along the Treg cell differentiation continuum (Cheng et al, 2012; Gaublomme et al, 2015; Guo et al, 2018; Zemmour et al, 2018; Miragaia et al, 2019), with each *subset* containing multiple transcriptionally defined *cell states*. We sought to organize all Tregs from our dataset into different cellular states to generate a single-cell Treg classification schema, which could then define cellular variation across tissues and after IL-2M stimulation at a detailed resolution. To do this, we first impartially defined cell states using unsupervised clustering, and then identified highly representative signature genes from each cell state to anchor our cell state assignments in the context of known Treg biology.

Cell clustering of all Tregs using the graph-based Louvain algorithm from Seurat (Satija et al, 2015) identified three established

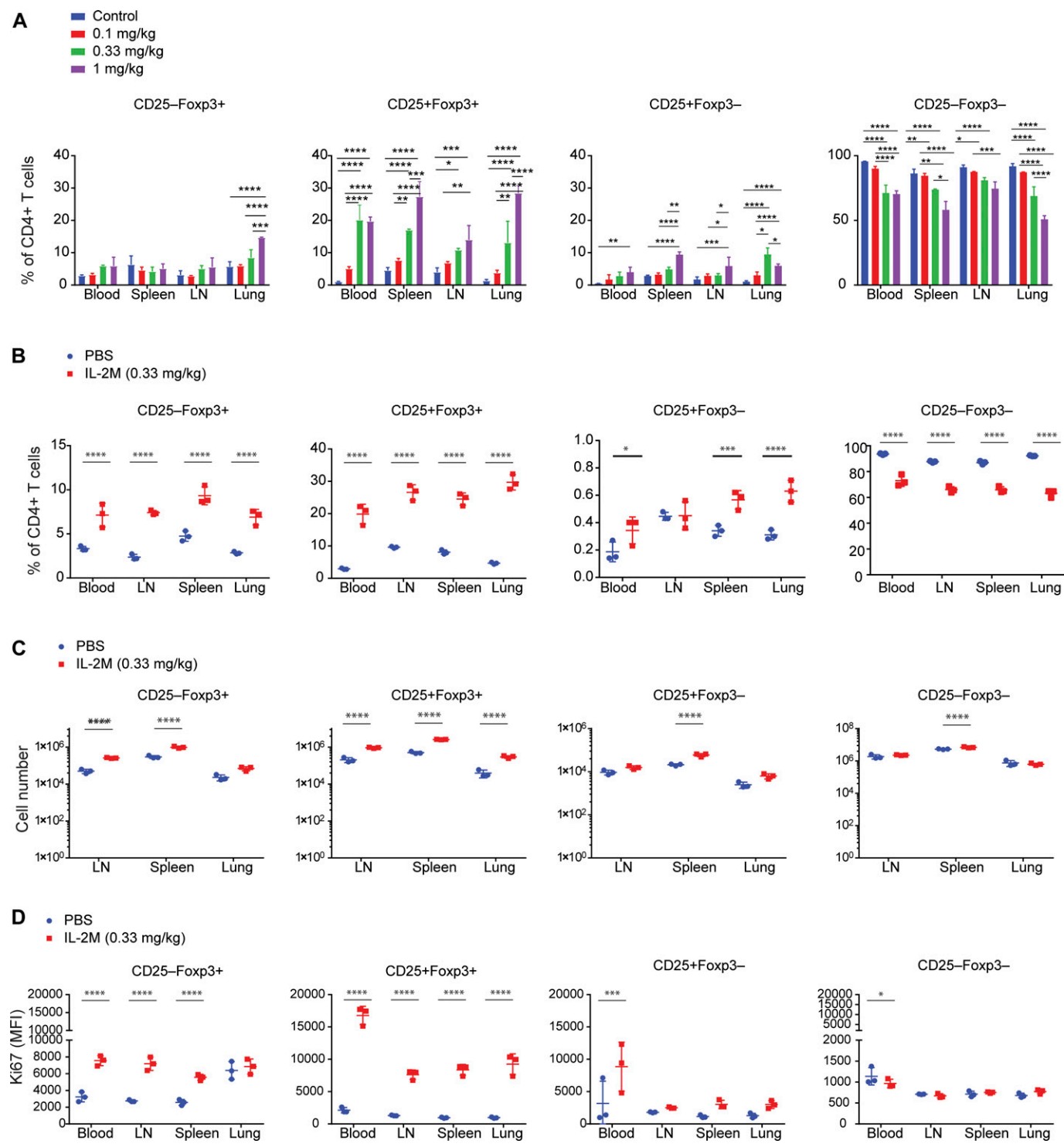

**Figure 1. A half-life–extended mutant form of the murine IL-2M expands CD25⁺Foxp3⁺ Tregs in vivo.**
**(A)** Dose-dependent effects of murine IL-2M on the percentages of CD25−Foxp3+ and CD25+Foxp3+ Tregs and CD25+Foxp3− activated Tconv and CD25−Foxp3− Tconv within CD4 T cells in the blood, spleen, lymph nodes, and lung. Experiments were performed using three mice per each group. Control was untreated mice. **(B)** Effects of murine IL-2M (0.33 mg/kg) on the percentages of each population within CD4 T cells from the blood, spleen, lymph nodes, and lung. **(C)** Effects of murine IL-2M (0.33 mg/kg) on the cell numbers of each population from the blood, spleen, lymph nodes, and lung. **(D)** Effects of murine IL-2M (0.33 mg/kg) on the proliferation of each cell population assessed by expression of Ki67 (mean fluorescence intensity). **(B, C, D)** results are representative of at least two independent experiment. Experiments were performed using three mice per each group. Control: PBS-treated mice. Statistics: two-way ANOVA for multiple comparisons. *0.01 < P < 0.05, **0.001 < P < 0.01, ***0.0001 < P < 0.001, ****P < 0.00001.

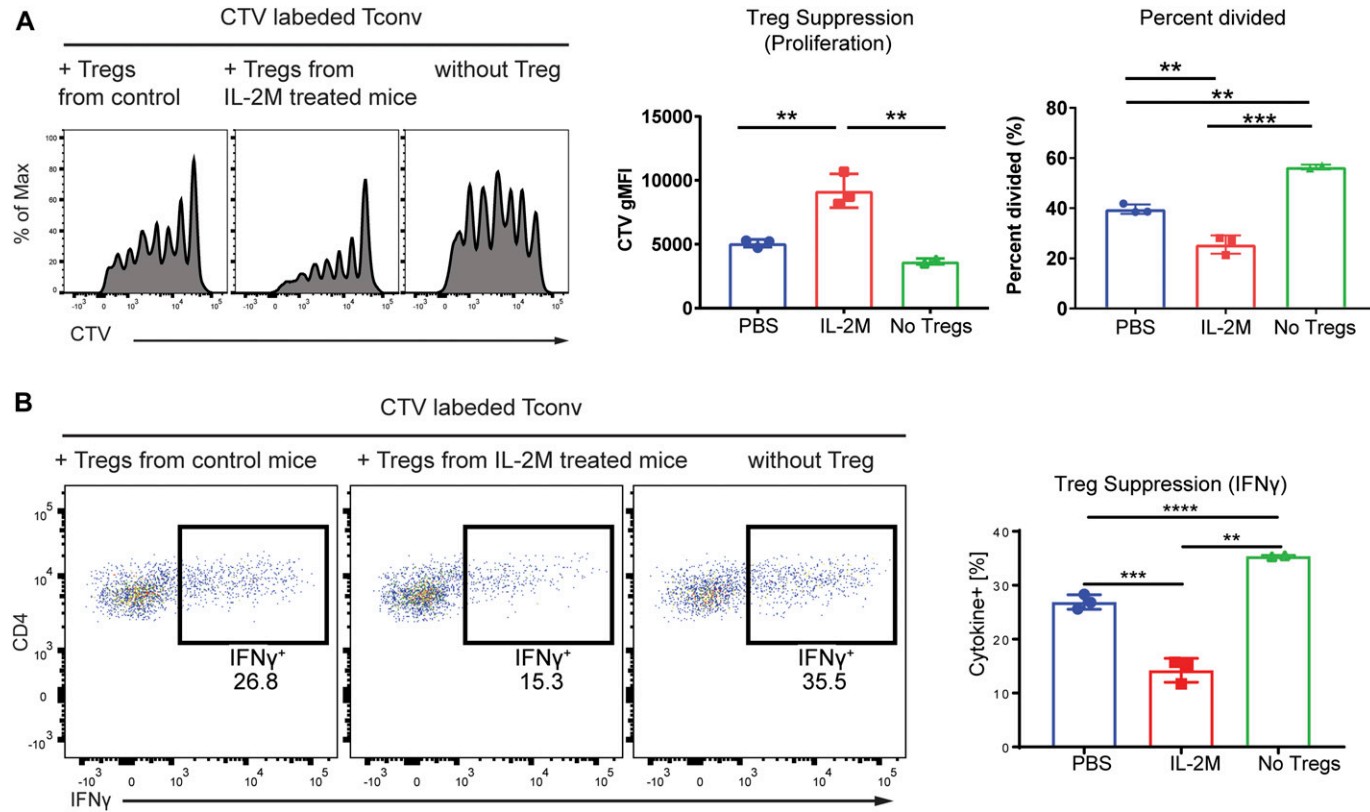

**Figure 2. Superior suppressive activity of CD25+Foxp3+ Tregs expanded by murine IL-2M.**
**(A)** In vitro suppression assays using sorted Foxp3-EGFP+ Tregs from the spleen of either PBS or murine IL-2M–treated Foxp3-EGFP mice. Naïve T cells were MACS sorted using Miltenyi naïve T-cell isolation kit, labeled with cell tracer violet (CTV), and then mixed with irradiated APC in the absence (No Tregs) or presence of Tregs from PBS- or IL-2M–treated mice. Tregs were cocultured then with naïve T cells and APCs under Th1 skewing condition. Shown are representative images of proliferation of Tconv indicated as dilution of CTV (left panel). Suppression of proliferation of Tconv was shown by diluted CTV MFI (middle panel) or percent divided (right panel). **(B)** Suppression of IFNγ generation from Th1 effector cells by Tregs. **(B)** In vitro Treg suppression assay was performed as described in (B), and expression of IFNγ from Tconv was measured by intracellular staining. Results are representative of two independent experiments. Statistics: one-way ANOVA for multiple comparisons. **0.001 < $P$ < 0.01, ***0.0001 < $P$ < 0.001, ****$P$ < 0.00001.

Treg subsets (Fig 3B, top), which could be divided further into 10 Treg cell states of activation (Fig 3B, bottom). These clusters were evenly represented in all replicates, indicating that they represent biological heterogeneity and not batch-specific processing artifacts (Fig S5A and B).

Signature genes were then defined for each cell state to understand transcriptional differences between Treg populations (see the Materials and Methods section) (Figs 3C and S6A). When grouped by resting, primed/activated, and activated Treg subsets, cell states had differential expression of known lymphoid-associated and activated genes. Resting Tregs (C1) have high expression of lymphoid-tissue homing receptors (*Ccr7* and *S1pr1*) and Treg-establishing transcriptional regulators (*Lef1* [Xing et al, 2019], *Satb1* [Kitagawa et al, 2017], and *Klf2* [Pabbisetty et al, 2016]) and low-activation marker expression (*Ctla4*, *Icos*, *Tnfrsf1b*, and *Tnfrsf4*) (Figs 3D and S7A), suggesting that they either have not yet undergone TCR-mediated activation or have a central memory-like phenotype.

Two primed/activated Treg states (C2 and C9) express intermediate levels of both resting and activation gene sets, suggesting they may be transitioning toward an activated phenotype (Figs 3D and S7A). In support of this observation, primed/activated Tregs are

distinguishable from C1 resting Tregs with higher expression of Treg-specific genes that stabilize inhibitory activity of Treg (C2: *Ikzf2* [Kim et al, 2015] and S100 family proteins) or inflammatory response mediators (C9: *Stat1*, *Cxcl10*, and interferon-response genes) (Fig S7B and C). Both C2 and C9 have high *Ms4a4b* expression (Fig S7D), which modulates the activation by regulating proliferation and augmenting co-stimulation through GITR (Howie et al, 2009; Xu et al, 2010). Furthermore, C2- and C9-Tregs could be distinguished from each other, as C2-Tregs express more *Nrp1* (which is expressed highly in natural Tregs [Yadav et al, 2012] and promotes Treg survival by interacting with Sema4a on Tconv cells [Delgoffe et al, 2013]), whereas C9-Tregs uniquely express interferon-response genes (Epeldegui et al, 2015) (*Bst2*, *Tgtp2*, *Ifit1*, and *Isg15*) (Fig S8A and B).

Six Treg clusters (C3-C5, C7, C8, and C10) showed an activated phenotype with the lowest expression of lymphoid-tissue homing markers and highest expression of activation genes (Fig 3D). Whereas activated cell states express high *Tnfrsf9* (encoding for costimulatory activator 4-1BB), each population also expresses distinct genes (Fig 3E). Differential gene expression of these activated clusters compared with C1-resting cluster or C2-primed/

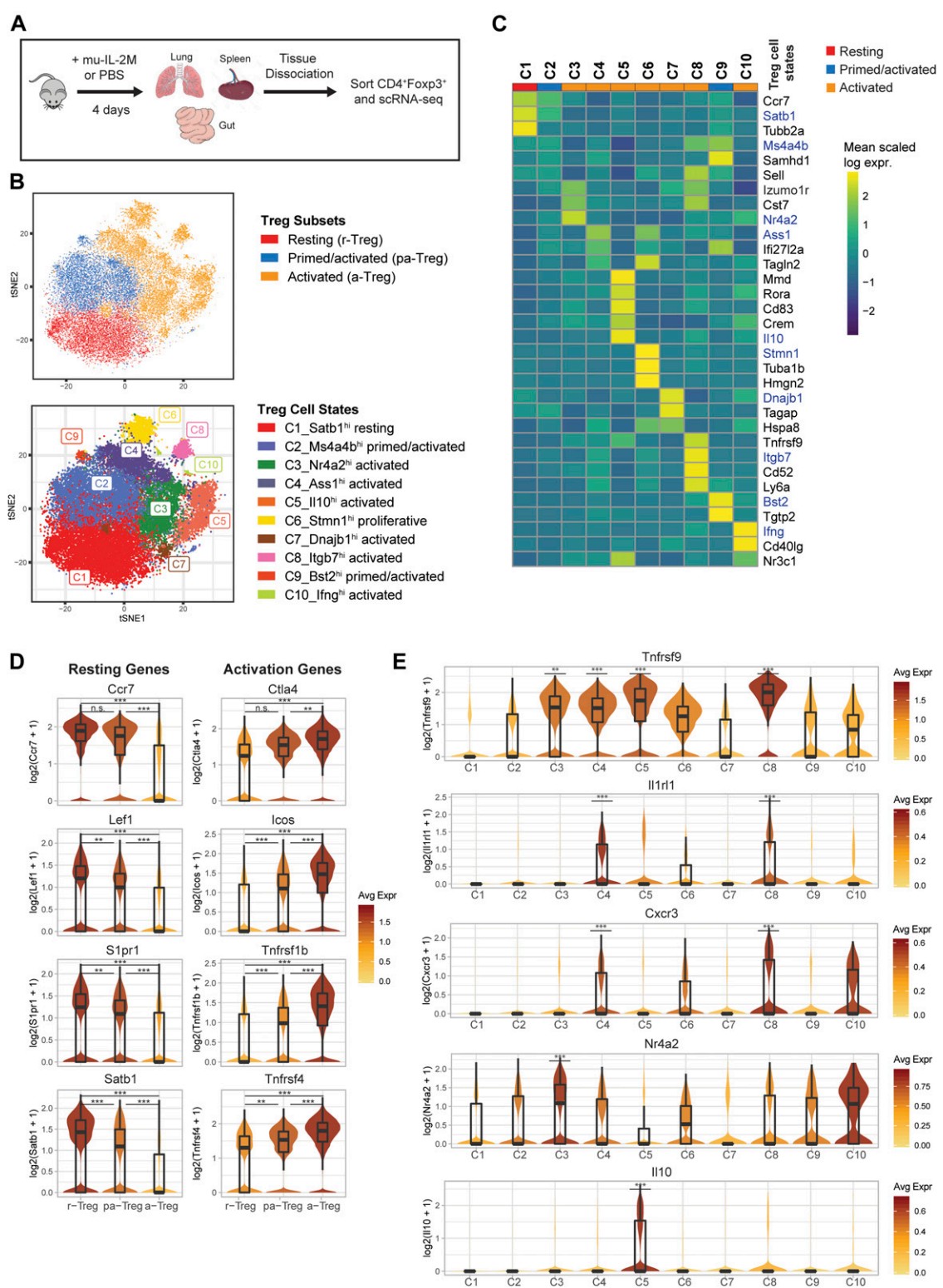

**Figure 3. Classification of heterogeneous Treg resting, primed, and activation states across tissues.**
**(A)** Experimental workflow for interrogation of Tregs in the spleen, lung, and gut after administration of IL-2M or mouse IgG Fc control (Iso) by scRNA-seq. **(B)** t-SNE projection of all Tregs grouped by cell subset (top) or by cell state (bottom) using K-nearest neighbor graph–based clustering. Individual cells are colored by cell state cluster assignments. **(C)** Heat map depicting the mean expression (Z-score) of most representative signature genes (rows) for each cluster (columns). Genes shown in blue text with an asterisk are genes used to characterize Treg states. **(D)** Violin plots depicting gradual down-regulation of classical resting Treg markers (left) and gradual up-regulation of classical activated Treg markers (right) in progressively higher Treg activation states. **(E)** Violin plots showing marker genes associated with sub-populations of activated Tregs. Statistics: Wilcoxon rank sum test. **P < 0.01, ***P < 0.001, n.s. = P > 0.05.

activated cluster or each other was shown (Figs S6B–E, S9A–D, and S10A–C). Both C4 and C8 Tregs express high *Il1rl1* (ST2) compared with all other states (Liao et al, 2010; Bandala-Sanchez et al, 2013; Mahmud et al, 2014; Schiering et al, 2014; Geiger et al, 2016; Siede et al, 2016), whereas they differentially express T-cell activation regulators and effector molecules (C4: *Ass1, Klrg1, Gzmb, and Tagln2* versus C8: *Itgb7, Cd52, and GITR*) (Figs 3E, S6B–E, S9B, and S10B). Similarly, C4 and C8 Tregs express high *CXCR3*, a chemokine receptor regulated by *T-bet* and associated with tissue Tregs that suppress Th1 responses (Koch et al, 2009) (Fig 3E). On the other hand, C3 Tregs express genes required for follicular regulatory phenotype (C3: *Maf* [Wheaton et al, 2017]) and genes that promote Treg survival and persistence (C3: *Nr4a2, Cst7*) (Figs 3E, 4C, S6B, and S9A), whereas C5 Tregs express effectors that promotes suppression (C5: *Il10* [Chaudhry et al, 2011], *Gzma* [Velaga et al, 2015], and *Gzmb* [Cao et al, 2007; Loebbermann et al, 2012]) or tissue protection (C5: *Areg* [Arpaia et al, 2015]) (Figs 3E, 4C, S6B, and S9C). In addition, we identified one proliferative cell state (C6) (*Mki67* and *Top2a*) that co-expresses activation markers (Fig S9D). C7 Tregs are a small population that expresses stress response genes (*Hspa1a* and *Hspa1b*) (Fig S10A). C10 Tregs is a small population as well, expressing genes associated with NK cells (*NKG7, CCl5, CXCR6,* and *Gzmb*) and cytokine (*Ifng*), which is expressed in ex-Tregs (Daniel et al, 2015) (Fig S10C).

## Analysis of Treg clusters at steady-state identified tissue adaptations of Tregs

We next sought to characterize the Treg landscape in the spleen, lung, and gut from isotype control–treated mice using the classification of Treg states described in Fig 3 as a reference criterion. At the tissue level, the spleen and lung share a >40% frequency of C1-resting and minor frequencies of primed/activated and activated Tregs. Conversely, >80% of gut Tregs are activated and the majority are C5-Tregs (Fig 5B–D). Although lymphoid organs, such as the spleen, are known to contain large reservoirs of resting Tregs that can be recruited during immune challenge, the prominence of resting Tregs in the lung is noteworthy and suggests that the lung plays an active role in steady-state immune surveillance.

Activated states of Tregs differ substantially between the tissues, indicative of tissue adaptations acquired by Tregs (Fig 4A and B). The spleen in the unperturbed state uniquely contained more than 30% C3-Nr4a2$^{hi}$–activated Tregs, which express differentiation-promoting transcription factor *Nr4a2* (Sekiya et al, 2011). In contrast, the lung did not contain the C3-Nr4a2$^{hi}$ Tregs as a main activation state. Rather, the lung contains the C4-Ass1$^{hi}$ and C8-Itgb7$^{hi}$–activated clusters and C6-Stmn1$^{hi}$ proliferative cluster. Comparison of C3 and C4 revealed that C3 cluster differentially express *TBC1D4*, a transcription factor highly expressed in follicular regulatory Tregs, suggesting C3-Nr4a2$^+$–activated Tregs include Tfr in the spleen (Fig 4E). The co-expression of immunomodulatory genes (*Izumo1r* [Yamaguchi et al, 2007] and *Nt5e* (*CD73*) [Kobie et al, 2006]) and a reduced activation phenotype (*Tnfrsf9*$^{med}$*Cd83*$^{med}$) without expression of effector proteins compared with other activated states suggest that Nr4a2$^+$ Tregs are activated Tregs that are not terminally differentiated but still have suppressive capabilities. Compared with C1-resting Tregs, C4-cluster Tregs express high

levels of effector molecules (*GzmB, Glrx, and Klrg1*) and chemokine receptor (*Ccr2*), suggesting this cluster may contain terminally differentiated Tregs with tissue-homing receptors (Cheng et al, 2012) (Fig 4C–F).

The gut Tregs contain small proportion of C1-resting Tregs and very little C2 or C9 prime/activated Tregs. Instead, they primarily contain C5- and C7-activated Treg populations. The C5-IL10$^{hi}$ Tregs uniquely express effectors such as IL10 and tissue-repairing Areg (Rubtsov et al, 2008) (Figs 3E and S9C) and proinflammatory transcriptional regulator *Rora* (Schiering et al, 2014) and elevated levels of gut-homing chemokine receptor *Ccr9* (Iwata et al, 2004). The C7-Dnajb1$^{hi}$–activated Tregs express early response inflammatory (*Dnajb1* [Regateiro et al, 2012], *Hspa1a* [Wachstein et al, 2012; Collins et al, 2013], *Tagap* [Arshad et al, 2018], and interferonresponse–related) genes, some of which are associated with enhanced Treg function and autoimmunity (Fig S10A). Given that cells in the gut regularly encounter diverse commensal microbes and food antigens, these Tregs may actively promote tolerance even during steady-state conditions.

## Reorganization of Treg landscape in multiple tissues after treatment with murine IL-2 mutein

We demonstrated the expansion of Tregs after murine IL-2M in vivo (Fig 1), but whether this treatment will expand all Treg subpopulations equally is not clear. Therefore, we compared the frequency of Treg states in each tissue before and after IL-2M to detect changes in molecular phenotypes and function. Treatment with IL-2M shifted the frequency of Treg clusters, reducing C1-resting and C3-activated Tregs while increasing primed/activated (C2) and activated Treg states (C4 and C8) (Fig 5A [distribution of all Tregs from isotype versus IL-2M–treated mice], Fig 5B and C [distribution of specific tissue Tregs from isotype versus IL-2M–treated mice]). These shifts were consistent in all animals (Fig 5C) and reflected differences at the tissue level, as lung and spleen Tregs shared most of the genes after IL-2M treatment (Fig S11A), but they were distinct when compared with gut Tregs (Fig S11B and C). In the spleen, lung, and gut, C2-primed Tregs and C4-activated Tregs all displayed an more than fourfold increase in all animals (Fig 5D). This expansion of C2- and C4-Tregs in the spleen and lung established these two states as the most prominent Treg populations after IL-2M stimulation, coinciding with a drastic reduction in C1-resting Tregs in both tissues and in C3-activated Tregs in the spleen. A twofold increase was also observed in C8-activated Tregs in the spleen and lungs. The selective expansion of C4- and C8-Tregs, which both co-express *Tnfrsf9* and *Il1rl1*, suggests that IL-2M may skew expansion in eligible tissues toward specific activation phenotypes. Gut Tregs still maintained C5-activated Tregs after IL-2M treatment as the most prominent cell state (Fig 5C and D).

## IL-2 mutein skews clonal Treg expansion toward specific activation states

During development in the thymus, each T cell generates a unique TCRα and -β rearrangement (or clonotype), which predicates antigen specificity and influences Treg fate. This clonotype is retained in cells that descend from a common progenitor T cell, enabling characterization of Tregs that share a developmental lineage.

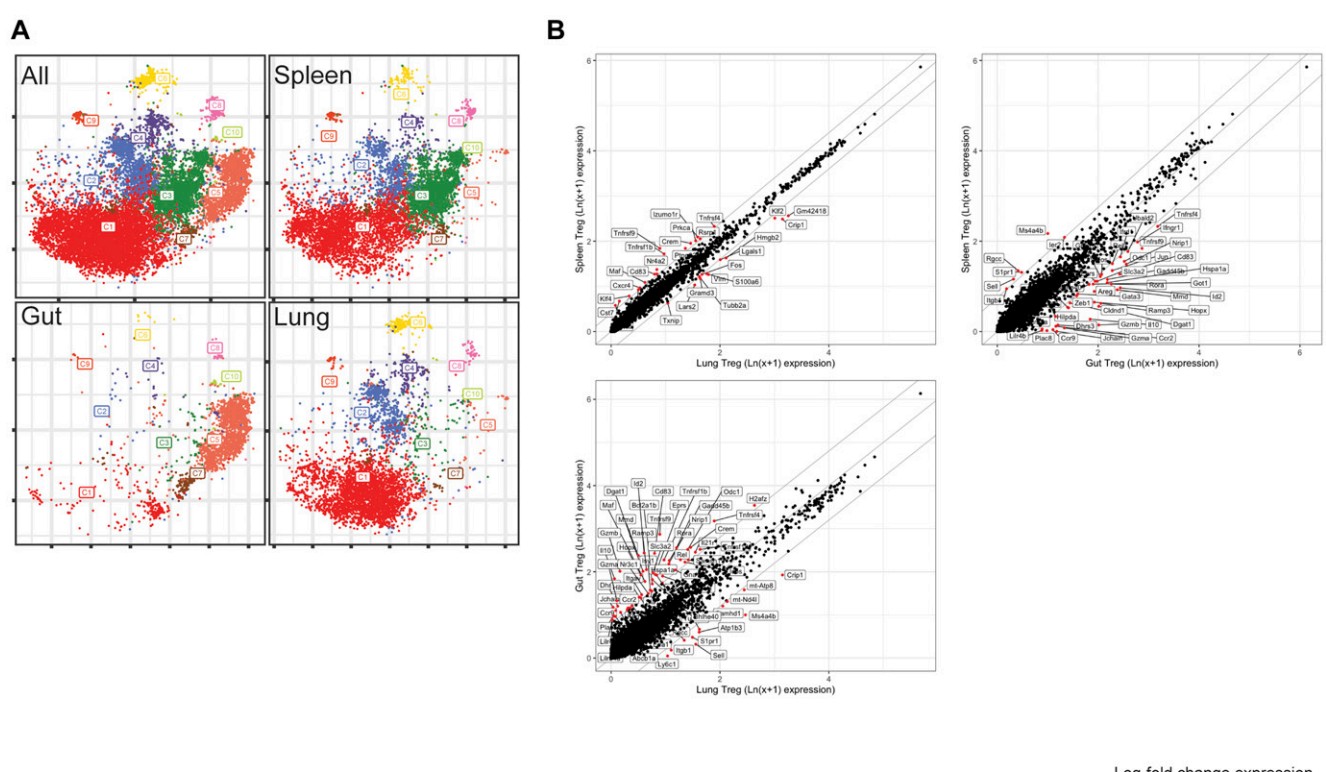

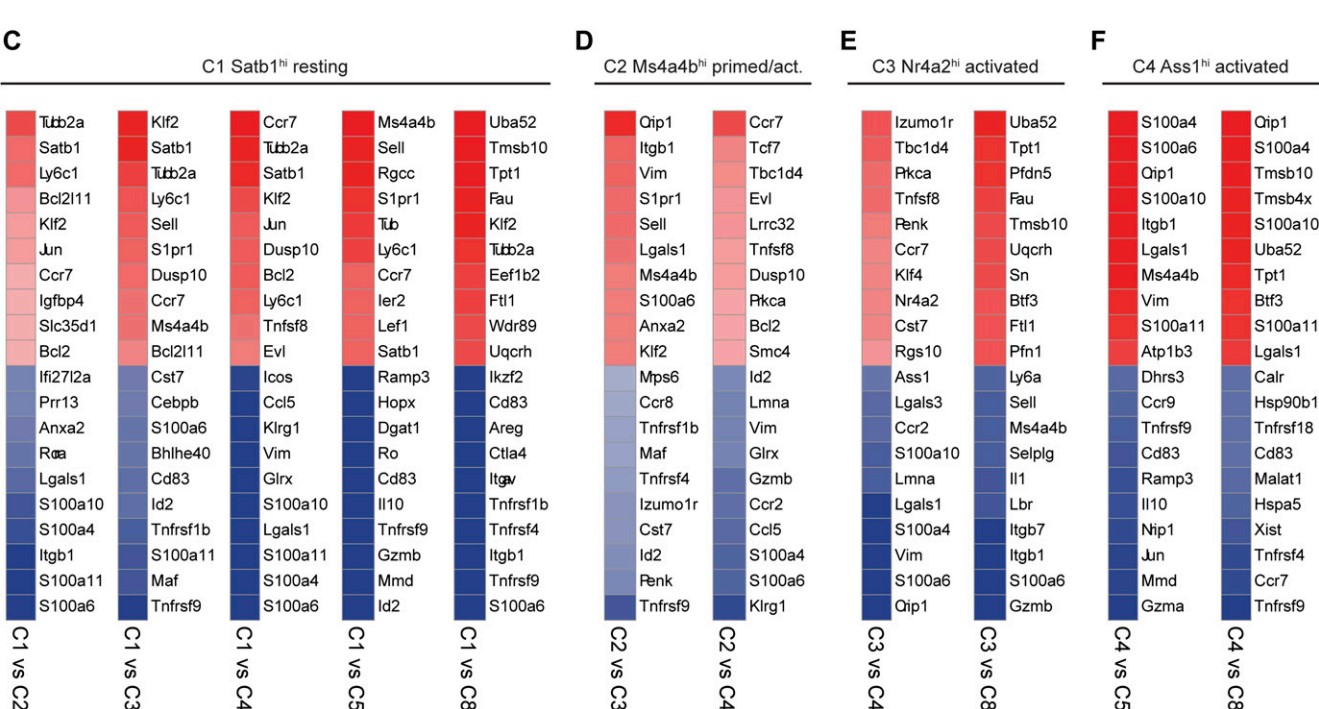

**Figure 4. Heterogeneous landscape of Treg cell states defined between the spleen, lung, and gut.**
**(A)** t-SNE projection of Treg cell states in the spleen, lung, gut, and combined at steady state (isotype control–treated). Individual cells are colored by Treg cell state classification from Fig 3. **(B)** Comparison of Treg signature genes between all tissue-specific cells in the spleen versus lung, spleen versus gut, and lung versus gut at steady state. Significant genes are shown in red (adjusted $P$-value < 0.05, MAST). **(C, D, E, F)** Select heat maps showing the top 10 most differentially expressed genes from all Tregs (spleen, lung, and gut) of isotype control (Iso)–treated mice when comparing between two Treg cell states using MAST. Genes are colored by the difference in log$_2$-fold change expression between cell states.

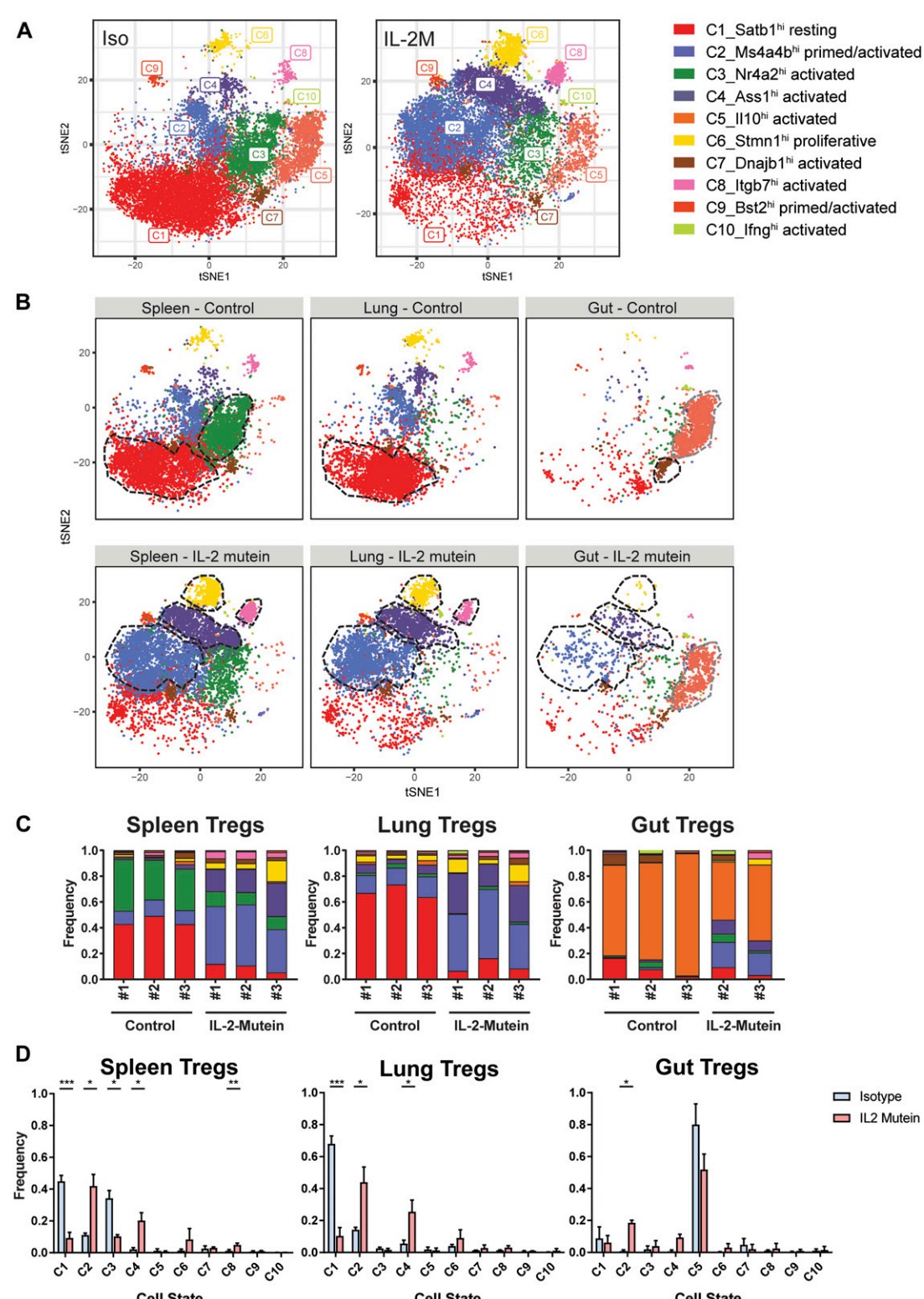

**Figure 5. IL-2 mutein induces a convergent expansion toward distinct Treg states in the spleen, lung, and gut.**
**(A)** t-SNE projection of all Tregs shown by cell state from Iso-treated mice (left) or Tregs from IL-2M–treated mice (right) using K-nearest neighbor graph–based clustering. Individual cells are colored by cell state cluster assignments. **(B)** t-SNE projection of Treg cell states in the spleen, lung, and gut from isotype control–treated (Iso) and IL-2 mutein treated mice. Treg clusters that are enriched at least twofold in isotype control–treated or in IL-2 mutein–treated mice relative to the other treatment condition are outlined in with a black dotted line. Prominent populations (>10% of total Tregs in a given tissue) whose frequencies are consistent across treatment are highlighted in gray. **(C)** The tissue-specific frequency of Treg cell states for each experimental replicate. **(D)** Bar plots showing shift in Treg population in the spleen, lung, and gut. N = 3 for all isotype and IL-2 mutein treatments except for the IL-2 mutein–treated gut Tregs (N = 2). Statistics: Welch's t test. *P < 0.05, **P < 0.01, ***P < 0.001.

Because we observed an expansion in Tregs (1) toward distinct cell states (C2, C4, and C6) in all three tissues, (2) toward the C8 state in the spleen and lungs, and (3) away from the C3 state in the spleen, we analyzed TCR information combined with Treg cell state classifications from single cells to elucidate whether these cell states expand independently of one another or whether they are developmentally related.

From the 3,600 sequenced single-cell transcriptomes where TCRα and TCRβ were also recovered (see the Materials and Methods section), 3,405 unique Treg clonotypes (1,428 in Iso- and 1,977 in IL-2M–treated conditions) were identified. Of these clonotypes, 119 CFs (i.e., clonotypes that were shared by at least two Tregs) consisting of 314 total Tregs were identified. Among the 119 CFs, 67 CFs of Tregs existed in different phenotypically distinct activated states (e.g., C3 and C4) or in primed/activated and activated states (e.g., C2 and C4), at both steady state and after IL-2M stimulation (Fig 6), revealing that different Treg cell states can be derived from the same parental Treg. The identification of transcriptional diversity among Tregs from the same clonal family (CF) was an interesting result because we also find that pairs of T cells belonging to the same clonotype tend to be transcriptionally correlated than randomly sampled pairs of Tregs at the population level ($P = 2.106 \times 10^{-4}$ when comparing equal numbers of cells from same and different clonotypes), although this was a modest effect, as previously reported

(Zemmour et al, 2018) (Fig S12). CF membership also revealed evidence of Treg migratory patterns between tissues because we observed that Tregs of the same clonotype spanned the spleen, lung, and gut (Fig 6). Taken together, we interpret these findings as such that, although the TCR responses may generally influence the transcription of Tregs toward a programmed transcriptional fate, clonally related Tregs still can differentiate into multiple cell states depending on the tissue.

We next characterized cell state classifications of clonotype pairs to identify cell states that may be developmentally linked (which we refer to as "cell state axes"). At the steady state, no cell state axis comprised more than 1% of all clonotypes because the majority of clonotypes (95.6%) did not belong to a CF (Fig 7B, top panel). In contrast, IL-2M–induced clonal proliferation facilitated the detection of prominent cell state axes connecting clonotype pairs by increasing the proportion of Tregs detected in a CF by 105% (211 CF Tregs in IL-2M compared with 103 CF Tregs in isotype), despite only a 41% increase in Tregs with TCRs recovered (2,107 TCRs in IL-2M, 1,493 TCRs in isotype). Multiple axes higher than 1% frequency were identified after IL-2M treatment (Fig 7B, bottom panel). Among those, multiple clonotypic pairs were observed in the C4-Tregs, including proliferation-dependent C4-C4 (3.9%) and C4-C6 (2.7%) clonotype pairs, as well as C2-C4 (1.0%) and C3-C4 (1.1%) pairs, revealing that C2, C3, and C4 are developmentally linked (Fig 7A and

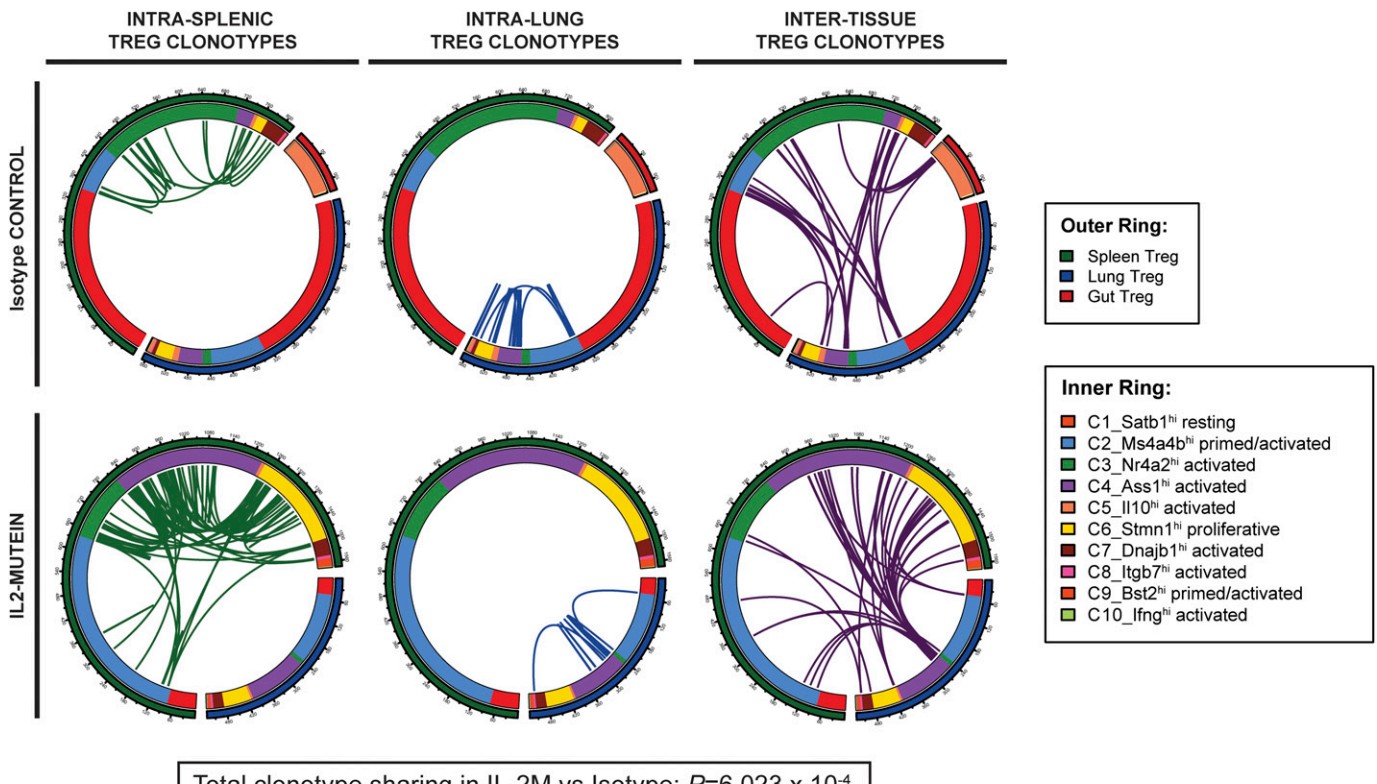

**Figure 6. Tregs from the same progenitor differentiate across multiple states within and across tissues.**
Chord diagrams of TCR clonotype sharing involving Tregs within the spleen (left), lung (middle), and across tissues (right) in control and IL-2M–treated condition. Outer ring denotes the tissue from which a Treg was isolated. Inner ring denotes the cell state identified by Seurat cell clustering. Green lines link splenic Tregs of the same clonotype. Blue lines link lung Tregs of the same clonotype. Purple lines link clonotypes shared between different tissues. *P*-values show Fisher's exact test for independence between clonal family frequency and IL-2 mutein treatment.

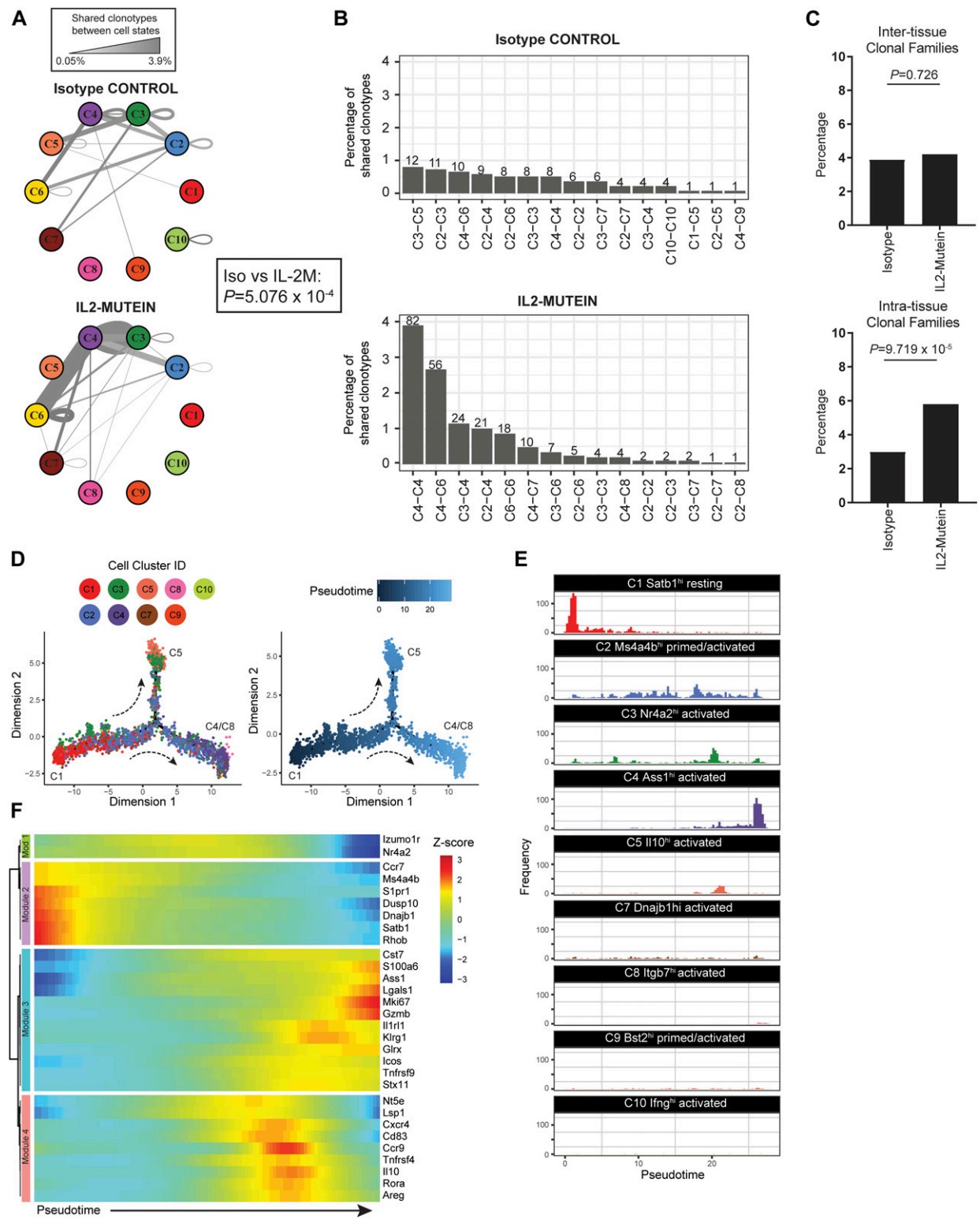

**Figure 7.  Treg lineage tracing and trajectory analyses identify axes of Treg differentiation across cell states.**
**(A)** Network graph representing the percentage of shared clonotype pairs (i.e., pairs of cells belonging to the same clonotype that exist across two cell states) of spleen and lung Tregs under isotype control– or IL-2M–treated conditions. The thickness and darkness of the gray edges connecting two cell states corresponds to the percentage of shared clonotypes between two cell states or within a single state. **(B)** Top 15 pairs of cell states that share clonal family membership. Numbers above each bar indicate the total number of clonotype pairs in a given cell state axis. *P*-values show Fisher's exact test for independence between matched cell state pairs frequency and IL-2 mutein treatment. **(C)** Comparison of the percentage of inter-tissue and intra-tissue clonotypes between isotype control– and IL-2M–treated

B). C4 also shared clonotypes with activated states C7 and C8, albeit at a lower frequency of 0.5% and 0.2%, respectively.

Given that IL-2M increases clonal Treg expansion, we also examined how IL-2M influences the localization of CF Tregs by comparing the frequency of CF Tregs that were shared across tissues versus within the same tissue. Although the percentage of inter-tissue CFs remained the same in both isotype and IL-2M conditions (3.9% versus 4.2%, respectively), the percentage of CFs found only in one tissue was nearly doubled (3.0% versus 5.8%, respectively) (Fig 7C). Although we cannot ascertain whether in situ expansion or migration from other tissues is responsible for this phenomenon, these results indicate that CF Tregs after IL-2M–induced clonal expansion are more frequently observed in the same tissue than across multiple tissues.

### C4 and C8 Tregs expand after IL-2M stimulation through C2 intermediate state

Given the gene expression profiles and robust lineage relationship of the C2-primed states and C3/C4/C8 activated states, we used pseudotime analyses using Tregs with recovered TCRs (n = 3,600 cells) to define their developmental relationship (Figs 7D and E and S13A). Treg cell states occupied distinct territories in pseudotime. We defined the node enriched for C1-resting Tregs as the root node (pseudotime $t$ = 0), and pseudotime values were assigned in an unbiased manner to the manifold based on the distance from that root node. As expected, C1-resting Tregs occupied the earliest period of pseudotime. C2-primed/activated and C3-activated Tregs express markers of previous TCR signaling and were dispersed throughout intermediate points in pseudotime. At the latest periods in pseudotime/differentiation, we observed two distinct differentiation branchpoints consisting of C5-Tregs at one terminus and C4/C8 Tregs at the other (Fig S13B). This suggests that Tregs generally follow a differentiation trajectory from C1 into either C5 Il10-producing Tregs or C4/C8 Treg that express effector molecules such as Klrg1, Gzmb, and Glrx. We obtained similar ordering of cell states when applying pseudotime analyses to the other Tregs in our dataset where no TCR information was obtained (Fig S13C and D).

The analysis of Treg trajectories also revealed gene expression dynamics that advance Tregs toward effector states. Of the 30 most variant genes identified from this analysis, four major gene modules were identified that correspond to cell-state classifications (Fig 7F). Module 1 consisted of Izumo1r and Nr4a2, the key markers of C3-Tregs, which are moderately expressed all throughout pseudotime until terminal differentiation. The expression pattern of Nr4a2 agrees with previous findings that establish its role in tonic TCR signaling to allow Treg persistence and survival (Sekiya et al, 2011). Module 2 consisted of genes most frequently found in resting (C1) and primed/activated (C2/C9) Tregs, which are highly expressed at early cell states and become quickly down-regulated once differentiation initiates. Modules 3 and 4 consisted mainly of genes that

were highly expressed in the C4/C8 and C5 terminal states, respectively. Overall, analysis of clonal Treg differentiation trajectories suggests IL-2M promotes differentiation into the terminally differentiated C4 and C8 Treg state by expanding through an intermediate state such as C2 in the spleen and lung.

### Among IL-2M expanded sub-populations, Tnfrsf9⁺Il1rl1⁺ Tregs demonstrated superiority in in vitro suppression functional assay

Pseudotime analysis demonstrated enrichment of genes in four different modules (Fig 7F). Among those genes, Il1rl1 and Tnfrsf9 are highly expressed in Module 3, and they are highly expressed in C4/C8 clusters (Fig 3E). Likewise, scRNA-seq demonstrated increased expression of Il1rl1 and Tnfrsf9 in the spleen and lung after the IL-2M treatment (Fig 8A). Il1rl1 or Tnfrsf9 encodes proteins ST2 or 4-1BB, respectively, and FACS reagents recognizing these proteins were readily available. To verify protein expression of these two genes, we performed flow cytometry analysis. Flow cytometry analysis confirmed that Foxp3⁺ Tregs expressing both Il1rl1(ST2) and Tnfrsf9 (4-1BB) were increased to ~10-fold from the lung and spleen of the mice treated with IL-2M at protein level (Fig 8A–C). Pan-Tregs isolated after IL-2M treatment increased the suppressive activity (Fig 2A and B), but it is not clear which sub-state Tregs contribute to this enhanced suppressive activity. Because Tregs with expression of Il1rl1 and Tnfrsf9 after IL-2M treatment represent the end stage of the differentiation, we determined suppressive activity of Tregs with differential expression of ST2 and 4-1BB using in vitro suppression assays examining suppression activity of Tregs on proliferation and IFNγ production. To this end, Tregs were expanded in vivo by murine IL-2M administration. 4 d after treatment, four populations of Foxp3⁺ Treg cells based on the expression of ST2 and 4-1BB were sorted, and in vitro suppression assay was performed. Proliferation of effector T cells was suppressed by Tregs expressing either ST2 or 4-1BB, but ST2⁺4-1BB⁺ Foxp3⁺ Tregs displayed the most superior suppression (Figs 8D). and S14A and B). Similarly, ST2⁺4-1BB⁺ Foxp3⁺ Tregs were most efficient in suppressing IFNγ production of T effector cells under Th1 skewing condition (Fig 8E). Combined with increased suppression of Foxp3⁺ Tregs expanded with murine IL-2M in Fig 1, these data demonstrated that murine IL-2 mutein treatment preferentially increased the proportion of ST2⁺4-1BB⁺ Foxp3⁺ Tregs, which possess superior ability to suppress proliferation and IFNγ production of T effector cells. Thus, endogenous expansion of Tregs using in vivo murine IL-2M administration has the potential either to induce or expand Foxp3⁺ Tregs with the highest capacity to suppress autoreactive T cells.

## Discussion

IL-2 plays an indispensable role in immune tolerance by governing the proliferation and maintenance of Foxp3⁺ Tregs. IL-2 GWAS

---

conditions. Fisher's exact test was used to test for the effect of IL-2M treatment on inter-tissue or intra-tissue frequency. **(D)** Pseudotemporal ordering of Tregs with recovered TCRs (n = 3,600 cells). Individual cells are colored by cell state (left) or by pseudotime or progress along Treg differentiation (right). **(E)** Histograms showing the distribution of each Treg cell state along the pseudotime axis. **(F)** Heat map showing the Z-score scaled expression of 30 highly variant genes along pseudotime. Genes are grouped into modules with similar expression by unsupervised hierarchical clustering.

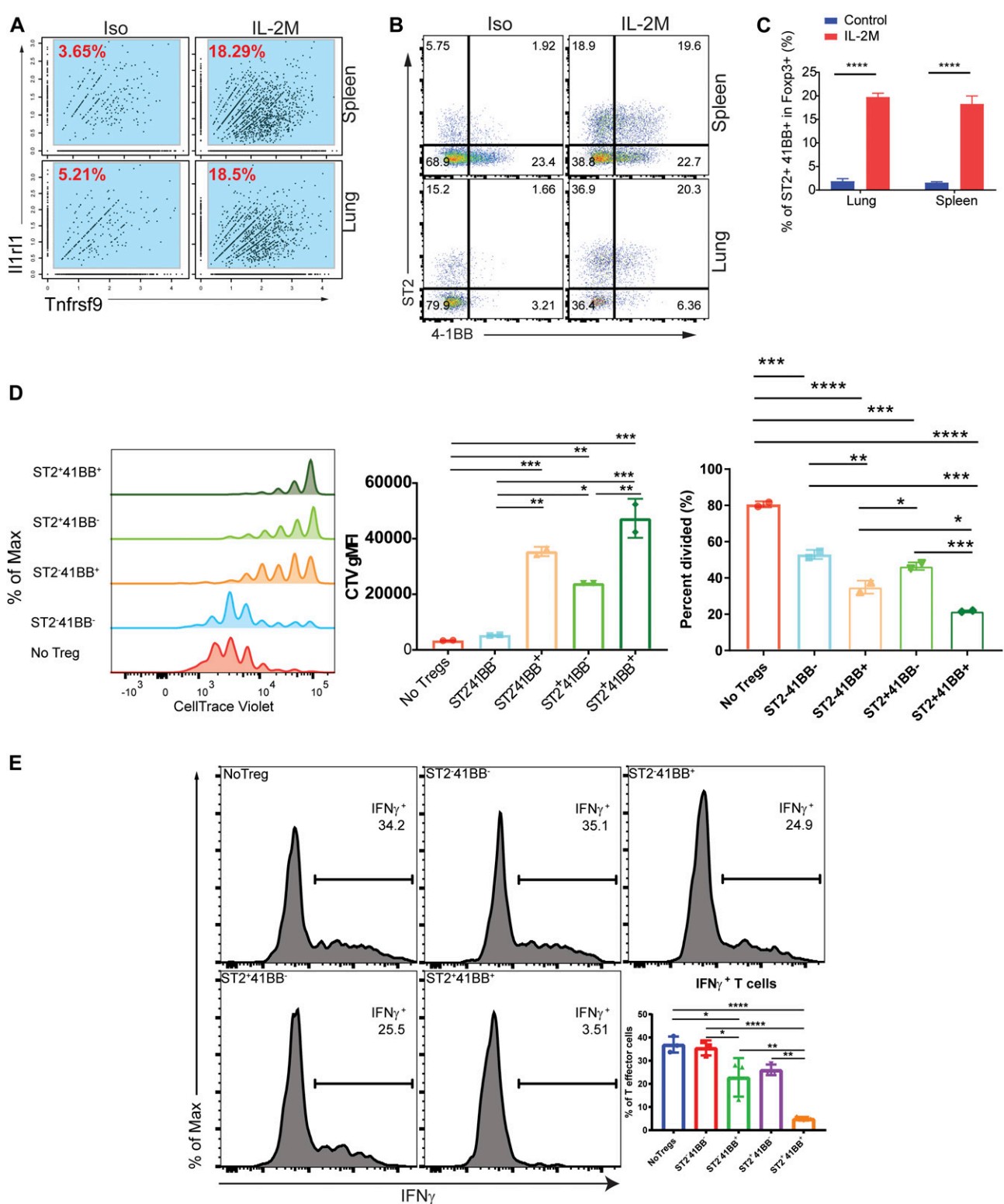

**Figure 8. Tnfrsf9+Il1rl1+ Tregs are superior in suppression functional assay of Tregs in vitro.**
**(A)** Expression of *Il1rl1*(*ST2*) and *Tnfrsf9*(*4-1BB*) in RNA single-cell analysis of Tregs from the spleen (*top*) or lung (*bottom*) of isotype control–treated (Iso) or murine IL-2M–treated mice (IL-2M). **(B)** Expression of ST2 and 4-1BB using FACS analysis from the spleen (*top*) or lung (*bottom*) of Iso or IL-2M–treated mice. **(C)** Quantification of ST2+4-1BB+ cells in Foxp3+ Tregs. **(B, C)** Results are representative of two independent experiments, using three mice in each experiment. **(D)** Fopx3-EGFP+ Tregs were stained based on ST2 and 4-1BB expression. Single-cell suspension from two IL-2M–treated mice was combined and four quadrants of Fopx3-EGFP+ Tregs were sorted based on ST2 and 4-1BB. In vitro suppression assays were performed in duplicate using four populations of sorted Tregs or without Tregs. Experiments were repeated

studies identified multiple polymorphisms affecting IL2 pathway genes encoding IL2Rα (CD25) and IL-2 as key genetic risk alleles in autoimmunity, corroborating IL2 as an essential cytokine for maintaining immune homeostasis (Vella et al, 2005; International Multiple Sclerosis Genetics et al, 2007; Yamanouchi et al, 2007; International Multiple Sclerosis Genetics, 2008; Onengut-Gumuscu et al, 2015; Wallace et al, 2015). Together with IL-2 therapy studies and mouse models defective in IL2 signaling, it is now well accepted that IL2 is the key cytokine necessary for generation, function, and survival of Foxp3⁺ Tregs. Consistent with genetic studies in human and mouse, reduction in numbers and functions of Foxp3⁺ Tregs had been reported as a contributing factor for a variety of autoimmune diseases (International Multiple Sclerosis Genetics, 2008; Long et al, 2011; Garg et al, 2012; Cerosaletti et al, 2013; Carbone et al, 2014; Moulton & Tsokos, 2015; Yang et al, 2015).

Given its role in Foxp3⁺ Treg biology, providing IL2 to Foxp3⁺ Tregs as an intervention for autoimmune diseases had been tested in clinic over the years. Recent clinical trials using low-dose recombinant human IL2 (Proleukin, aldeskeukin) have shown preliminary success in selectively expanding Tregs in chronic graft-versus-host-disease (Koreth et al, 2011; Hartemann et al, 2013; Kennedy-Nasser et al, 2014) and in steroid-refractory moderate-to-severe systemic lupus erythematosus patients (von Spee-Mayer et al, 2016). Additional efforts have focused on novel forms of IL2 with superior pharmacokinetics and Treg selectivity. These include a long-lived bivalent IgG-human IL2 fusion protein with N88D mutation that reduces IL2Rβ binding (Peterson et al, 2018), a polyethylene glycol (PEG)-modified murine IL2 that increases IL2 retention in the body by protection from enzymatic digestion and renal clearance (Wu et al, 2016), and a murine IL2/anti–IL2 antibody (JES6-1) conjugate (Spangler et al, 2018). Despite the effort to develop various forms of IL2 with superior pharmacokinetics and selectivity for treatment of autoimmune and inflammatory diseases, it is not clear how they affect heterogeneity of Tregs, given that Tregs exists in a variety of sub-states at different tissue sites, which are phenotypically and functionally heterogeneous.

To establish an effective Treg-mediated therapy, it is critical to characterize the phenotypic and functional heterogeneity induced by the treatment and define mechanisms that promote functional Treg differentiation. Because of the diverse profiles of proteins expressed in Tregs, it is challenging to define functional Tregs, let alone to identify Tregs with superior effector functions. CD4⁺CD45RA^low CD25^hi CD127^low cells have been previously used as human surface Treg markers to identify the pan-Tregs, and various activated markers of Tregs have been adopted to understand the functional and phenotypic diversity of Tregs. However, the detailed molecular characterization of Treg cell states is limited with a mere handful of activation markers. scRNA-seq overcomes this obstacle by providing a comprehensive view of transcriptional heterogeneity of Tregs, thereby making it possible to better design a targeted Treg therapy.

Using scRNA-seq and ex vivo cellular assays, we mapped the landscape of Treg sub-states at the single cell level during steady state and after murine IL-2M treatment in mice. We used an engineered, half-life extended murine IL2 mutant protein (IL-2M) and demonstrated that it expanded Tregs robustly in vivo. Expanded Tregs by murine IL-2M treatment suppressed proliferation of Tconv as well as production of IFNγ from Th1 effector cells. scRNA-seq study uncovered that IL-2M induces a convergent shift of the Treg landscape toward primed (C2) and highly suppressive activated Treg states (C4 and C8) in addition to proliferative Treg state (C6). Of note, because expansion of Tregs in the C2/C4/C8 clusters after IL-2M treatment is the result of action of IL-2M, we cannot distinguish which cluster of Tregs responded to the IL-2M to give rise to the expansion of the C2/C4/C8 clusters. Likewise, it is possible there is a preferential increase in a specific cluster to respond to IL-2M, resulting in the expansion of the C2/C4/C8 clusters. Furthermore, TCR-seq combined with scRNA-seq revealed a migratory network across tissues. IL-2M–mediated Treg expansion increased shared TCR clonality, and this expansion was more prominent in CFs that were identified in one tissue, whereas the percentage of inter-tissue CFs remained the same in both isotype and IL-2M treatment. Interestingly, shared TCR clones were also found in multiple and distinct cell states, providing us with a molecular snapshot of the transcriptional changes after IL-2M stimulation.

The similarities in single cell profiles between the spleen and lung versus the gut at steady state suggest that the spleen and lung in healthy mice possess a large proportion of resting-Tregs (C1-resting Tregs) that serve as a reservoir of Tregs in preparation for potential inflammatory stimuli. The spleen uniquely contains a high frequency of activated Tregs (C3-activated Tregs) expressing *Nr4a2*, a Foxp3-binding transcription factor (Kamalipour et al), *Cst7* (Hamilton et al, 2008), a marker for previous Treg activation and regulator of cytotoxicity, and *Izumo1r* (Yamaguchi et al, 2007), a marker for natural Tregs in the spleen. Compared with other activated Treg cluster C4, C3-activated cluster also differentially express *Tbc1d4* and *Maf*, transcription factors for follicular regulatory T cells (Wheaton et al, 2017; Wing et al, 2017). Given the role of the spleen in immune surveillance and its lymphatic connection to peripheral tissues as well as germinal center formation (Lim et al, 2004), these splenic Nr4a2⁺ Tregs may include follicular regulatory T cells (Tfr) and recently activated Tregs licensed to exit the spleen and migrate into sites of inflammation. On the other hand, the gut is composed of uniquely activated Treg clusters, IL10-producing suppressive C5 and inflammatory/stress response-associated C7 clusters, likely because of its constant exposure to commensal microbes, pathogens, and food antigens, it requires many activated Tregs that can maintain tissue homeostasis and promote tolerance (Geuking et al, 2011; Lathrop et al, 2011).

IL-2M treatment reduced the proportion of resting C1 Tregs in the spleen and lungs and the C3 Tregs in the spleen, whereas expanding the numbers of proliferating (C6), primed/activated

---

using three mice and in triplicate (data not shown). **(E)** Suppression function of four Treg sub-populations on IFNγ production. Four quadrants of Fopx3-EGFP+ Tregs were sorted based on ST2 and 4-1BB. Tregs were cocultured with effector T cells and APCs differentiated under Th1 skewing condition. IFNγ intracellular staining under the conditions using Tregs from PBS- or IL-2M–treated mice. No Tregs were added in one condition as a control. **(B, C, D, E)** Results are representative of two independent experiments, using 2–3 mice in each experiment. **(C, D)** Statistics: (C, D), one-way ANOVA for multiple comparisons. **$0.001 < P < 0.01$, ***$0.0001 < P < 0.001$, ****$P < 0.00001$.

Tregs (C2) and potently suppressive *Tnfrsf9*[+]*Il1rl1*[+] activated Tregs (C4 and C8). Because we observed a more than twofold expansion in the number of Tregs in all tissues after IL-2M stimulation, the decrease in C1 and C3 Treg percentage was likely driven more by the concomitant expansion of various expanding sub-populations as opposed to a significant reduction in the absolute numbers of the prominent Treg populations during steady state. Both expanded activated Treg populations, C4 and C8, express high *Il1rl1* (*ST2*), which is the receptor for tissue alarmin IL-33 (Siede et al, 2016). ST2[+] Tregs have been previously shown to prevent excessive tissue damage in organs such as the skin, liver, and gut (Matta et al, 2014; Schiering et al, 2014; Gajardo et al, 2015). C4 and C8 Tregs, although both expressing *Tnfrsf9* and *Il1rl1*, as well as effector molecules such as *Klrg1* and *Gzmb*, differ in expression of regulatory mediators such as *Ass1*, *Cd52*, and *Tnfrsf18*. Trajectory analysis suggests that the C4 and C8 Tregs are terminally differentiated. Previously, it has been shown that Klrg1[+] Tregs represent a terminally differentiated Treg subset that is recently activated by antigen and resides in the lamina propria of small intestine (Cheng et al, 2012). Klrg1[+] Tregs are highly activated and express enhanced levels of suppressive Treg molecules. Interestingly, the development of Klrg1[+] Tregs requires extensive IL-2R signaling (Cheng et al, 2012). Based on gene expression, it is highly likely that C4 and C8 Tregs contain Klrg1[+] Tregs. Between the two clusters, the C8 appears to be more terminally differentiated because of its expression of effector cell surface receptors *Cd52* and *Tnfrsf18*-encoded *GITR* (Bandala-Sanchez et al, 2013; Ephrem et al, 2013), which suppress other immune cells by binding to ITIM-containing Siglec10 and GITR-L, respectively. Identification of these expanded clusters suggests that IL-2M may significantly remodel the Treg landscape toward more terminally differentiated effector Tregs and suppressive Tregs, which mediate potent immunosuppression in these tissues.

We provide evidence using TCR analyses and trajectory analyses to suggest that C4 and C8 Tregs differentiate from C2/C3 primed/activated states. TCR analyses identify that C2, C3, C4, and C8 Tregs in the spleen and lung share clonotypes, demonstrating that they can all derive from a common progenitor. Cells from these populations that share the same clonotype are also present across the spleen and the lung, revealing an immune-trafficking axis that exists between the two tissues. Trajectory analysis also identifies a bifurcation in Treg differentiation, which either differentiate into suppressive Il10[+]Rora[+] C5 Tregs, which are most prevalent in the gut, or into C4/C8 Tregs that are prominent in the spleen and lungs. However, the trajectory analysis does not inform about whether IL-2M actually directs this bifurcation.

In this study, we characterize Tregs in the spleen, lung, and gut using scRNA-seq and analyze how the Treg landscape in these tissues is influenced by IL-2M stimulation. We identified an immune axis of clonal TCRs between the spleen and the lung and found an activated, Tnfrsf9[+]Il1rl1[+] Treg population that expand highly upon IL-2M stimulation. Supporting analyses indicate that this population differentiates through multiple primed and recently activated intermediate cell states and it is a highly suppressive Treg state that inhibits Th1 effector cell activity ex vivo. The mechanism of increased suppressive activity by IL-2M remains to be elucidated further. We demonstrated that IL-2M expands Tnfrsf9[+]Il1rl1[+] Tregs

with superior suppressive function. Another non-mutually exclusive possibility is that IL-2M induces transcriptional changes improving suppressor capacity within Tnfrsf9[+]Il1rl1[+] Tregs. Overall, this study reveals a potential mechanism by which IL-2M induces tissue-specific Treg immune modulation and classifies Treg state signatures that may serve as biomarkers for studying Treg responses. Although our analysis focused on genes with previously characterized immune cell functions, previously uncharacterized genes that were associated with Treg states of differentiation, such as certain ribosomal genes, warrant further investigation to elucidate their role in Treg biology.

# Materials and Methods

### Mice and IL-2 mutein treatment

All experimental studies were conducted under protocols approved by the Institutional Animal Care and Use Committee of Amgen. Animals were housed at Association for Assessment and Accreditation of Laboratory Animal Care International–accredited facilities at Amgen in ventilated micro-isolator housing on corncob bedding. Animals had access ad libitum to sterile pelleted food and reverse osmosis–purified water and were maintained on a 12:12 h light:dark cycle with access to environmental enrichment opportunities. Foxp3-EGFP (#006772), C57BL/6J, and B6.SJL-PrprcaPepcb/BoyJ (BoyJ) mice (8–12 wk old) were purchased from Jackson Laboratory. Once purchased, mice were housed under specific pathogen-free conditions in the laboratory animal facility and were handled according to protocols approved by Institutional Animal Care and Use Committee at Amgen. To determine the effect of murine IL-2 mutein (IL-2M), different amounts of IL-2M (0.1, 0.33, and 1 mg/kg) were administered into naïve C57BL/6J mice. After 4 d of IL-2M treatment, flow cytometry was performed to determine its effect on the regulatory and conventional T cells. To determine the effect of murine IL-2M on T-cell proliferation, Ki67 staining was performed. Untreated, PBS- or mouse IgG Fc isotype (31205; Thermo Fisher Scientific)–treated mice were compared and used as negative controls. Administration of murine IL-2M via i.p. or s.c. routes resulted in similar effects. For scRNA-seq experiment, the mice were administered with murine IL-2 mutein (0.33 mg/kg) or mouse IgG Fc isotype control (Iso) by i.p. injection and were sacrificed 4 d after murine IL-2M administration.

### IL-2/anti-IL2 antibody complex (IL-2C)

IL-2 complex (IL-2C) was prepared as described (Webster et al, 2009) by incubating 1 μg recombinant mouse IL-2 (PeproTech) with 5 μg purified antimouse IL-2 antibody (JES6-1; BioXcell) for 30 min at 37°C (molar ratio for IL2:anti-IL-2 is 2:1). IL-2C (total 6 μg) was administered i.p. in a final volume of 200 μl daily for 3 d. 4 d after the last administration of IL-2C, expansion of Tregs and Tconv was determined by FACS analysis.

### Flow cytometry reagents and Treg sorting

Anti-CD11c (N418), anti-F4/80 (BM8), anti-CD19 (6D5), anti-CD11b (M1/70), and anti-CD45.1 (A20) were from BioLegend. Anti-ST2

(U29-93), anti-CD3 (17A2), anti-CD4 (RM4-5), anti-IFNγ (XMG1.2), and anti-CD16/32 (2.4G2) were from BD Biosciences. CellTrace Violet, Fixable Viability Dye eFluor 780, and Foxp3 (FJK-16s) were from Thermo Fisher Scientific. Anti-4-1BB (158332) was from R&D.

## Isolation of murine spleen and lung Treg cells for scRNA-seq

The spleen and lung were collected and processed according to the protocol from Miltenyi murine Lung Dissociation kit (130-095-927). Lungs were perfused via the right ventricle before the process. Briefly, tissues were cut and rinsed in PBS. The tissues were transferred into the gentleMACS C Tube containing an enzyme mix of Enzyme D and A. gentleMACS Program m_lung_01 was run twice and the samples were incubated for 30 min at 37°C under continuous shaking using the gentleMACS dissociator (Miltenyi Biotec) at 150 rpm. Then, gentleMACS Program m_lung_02 was run. The tissue disaggregate was filtered through 70-µm strainers, and the cells were washed with PEB (PBS [pH 7.2] 2 mM EDTA, and 0.5% bovine serum albumin) buffer twice. The samples were ready for downstream flow staining and sorting.

## Isolation of murine gut Treg cells for scRNA-seq

The small intestine (extending from the end of stomach to the cecum) was collected and processed according to the protocol from Miltenyi murine Lamina Propria Dissociation kit (130-097-410). Briefly, the feces and Peyer's patches were removed, and the small intestine was filleted and cut into pieces. The samples were washed in pre-digestion solution, vortexed, and passed through 100-µm filters twice. Then, the samples were washed in HBSS (w/o), vortexed, and passed through 100-µm filters. The samples were transferred to gentleMACS C Tube containing preheated 2.35 ml of digestion solution with Enzyme D, R, and A. The samples were incubated for 30 min at 37°C under continuous shaking using the gentleMACS dissociator (Miltenyi Biotec) at 150 rpm. The C Tube was plugged into gentalMACS and run Program m_intestine_01 twice. The tissue disaggregates were filtered through 100-µm strainers, and the cells were washed with PB (PBS [pH 7.2] and 0.5% bovine serum albumin) buffer twice. The samples were ready for downstream flow staining and sorting.

## Treg suppression assay

Various populations of Treg cells were FACS-sorted on Aria II sorter (BD Biosciences) based on their expression of EGFP reporter. Naïve CD4⁺ T cells were isolated by Miltenyi mouse naive CD4⁺ T cell isolation kit and were used as effector T cells. APCs were prepared by depleting CD90⁺ T cells from splenocytes and irradiated for 3,000 rads (Faxitron). Naïve CD4⁺ T cells and APCs were from BoyJ mice, which is CD45.1+ to differentiate from EGFP+ Treg cells which were in B6 background by flow cytometry. To test the suppression function of Treg cells on proliferation, various populations of Treg cells (1 × 10⁴/well) were cocultured with effector T cells (2 × 10⁴/well) and APCs (3 × 10⁴/well), as indicated, along with anti-CD3 (145-2C11, 1 µg/ml; BioXcell) for 72 h. To test the suppression function of Treg cells on IFNγ production, various populations of Treg cells (2,500/well) were cocultured with effector T cells (2 × 10⁴/well) and APCs (3 × 10⁴/well), along with anti-CD3 (1 µg/ml),

anti-IL-4 (11B11, 10 µg/ml; BioXcell), and IL-12 (6 ng/ml; PeproTech) for 96 h. Then, the whole culture will be stained for intracellular cytokine staining. Briefly, the cells were washed once and were stimulated with cell stimulation cocktail (eBioscience) for 5 h in cell culture medium, and then fixed and stained for IFNγ.

### Single-cell RNA-seq

scRNA-seq was performed in three replicates, with each replicate containing the spleen, lung, and gut Tregs from one isotype control–treated mouse (Iso) and one IL2-mutein treated mouse (IL-2M). Combining different treatment conditions and tissues in each replicate allowed us to identify and regress out gene expression differences driven by batch processing.

CD4⁺Foxp3⁺ single cells were sorted from tissue-dissociated single-cell suspensions after gating on CD14⁻CD19⁻CD11c⁻F4/80⁻ cells. After sorting, the cells were immediately washed and resuspended in chilled PBS + 0.04% BSA at 100–1,000 cells per microliter. Washed cells were then encapsulated in one lane of a 10× Chromium Single Cell Controller, and libraries were constructed using either the 10× Single Cell 3′ Reagent Kit (V2 chemistry) (https://support.10xgenomics.com/single-cell-gene-expression/library-prep/doc/user-guide-chromium-single-cell-3-reagent-kits-user-guide-v2-chemistry) or the 10× Single Cell 5′ Reagent Kit (https://support.10xgenomics.com/single-cell-vdj/library-prep/doc/user-guide-chromium-single-cell-vdj-reagent-kits-v1-chemistry) according to the manufacturer's protocols. Completed libraries were sequenced on the HiSeq 4000 or NovaSeq 6000 platforms (26/8/0/98 Read1/i7/i5/Read2). Data were analyzed using the Cell Ranger 2.0.0 Pipeline by aligning reads to a genome containing mm10 genome and eGFP (Table S1).

**Post-alignment filtering and data aggregation** The Seurat (version 2.4) implementation (Satija et al, 2015) in Scanpy (version 0.94) (Wolf et al, 2018) was used to aggregate data from multiple experimental replicates and cluster Tregs into distinct states. Cells with low-quality transcriptomes (<500 detected genes) and doublets (>8,000 genes) were removed from the analysis. Canonical correlation analysis from the Seurat package was performed to identify common sources of variation and align data from different library preparation batches (each batch consisted of one isotype and one IL-2M treated sample of Tregs from each tissue), thereby mitigating batch effects. After filtering, our dataset detected 18,367 total genes and 31,908 cells across all replicates (12,840 cells in replicate 1, 11,822 cells in replicate 2, and 7,246 cells in replicate 3). We recovered an average of 5,032 unique mRNA molecules after collapsing duplicate unique molecular identifiers and 1,635 genes per cell.

**Unsupervised cell clustering and differential expression analysis of clusters** Unsupervised cell clustering of scRNA-seq data was performed using the FindClusters() function from Seurat. The optimal clustering resolution in Seurat was determined by clustering integrated single-cell expression data at 10 different resolutions from 0.1 to 1.0 using the "*resolution*" parameter in the FindClusters() function. At each increasing resolution, the top marker genes of the cluster containing the fewest cells were evaluated against previously published literature to support inclusion. The determination of 0.6 for the resolution parameter was made because of the

identification of an IFNg<sup>hi</sup> Treg cluster because lower resolutions did not identify this population. Additional clusters at higher resolutions did not reveal additional known immune markers.

Differential expression testing of each cluster (i.e., Treg cell state) versus all other clusters was performed using MAST (Finak et al, 2015) to generate a list of differentially expressed genes for each cluster. A list of genes that were significantly up-regulated genes by both tests were then used to define hallmark marker genes for each cell state. Hallmark genes for each of the Treg sub-populations was determined by selecting from among the top 10 genes with an adjusted $P$-value < 0.01, average $\log_2$-fold change expression >0.3, and statistically significant enrichment in less than half of the total cell clusters. The inclusion of genes significantly enriched in multiple clusters was set under the assumption that whereas the combinatorial marker gene expression is unique to each cluster, individual marker genes may be up-regulated by multiple cell states. Mitochondrial and pseudogenes were excluded from consideration as gene markers. T-stochastic neighbor embedding using RunTSNE() was used from Seurat for dimensionality-reduced visualization of single cell data.

**Single-cell TCR-seq** For samples prepared using the 5′ Reagent Kit, 2 μl of amplified cDNA was set aside to amplify mouse TCRa and TCRb chains in two sequential target enrichment PCRs using custom reverse primers: 5′-TGAAGATATCTTGGCAGGTG-3′ and 5′-TGCTCAGGCAGTAGCTA-TAATTGCT-3′ were used for the first PCR and 5′-GATCTTTTAACTGGTA-CACA-3′ and 5′-TTTGATGGCTCAAACAAGGA-3′ were used for the second PCR. All forward primers, primer concentrations, and PCR conditions and cycles were the same as those specified in the manufacturer's protocol. Completed libraries were sequenced on the HiSeq 4000 platform (150/8/0/150 Read1/i7/i5/Read2). Data were analyzed using the Cell Ranger 2.0 Pipeline by aligning reads to all IMGT mouse TCR variable and constant region sequences (Lefranc et al, 2009). Germline gene segments and CDR3 TCRa and TCRb sequences were identified and clonotypes were defined by single cells containing a recovered TCRa and TCRb with identical germline gene segments and CDR3 sequences. In total, 3,600 cells contained complete TCR and transcriptomic information out of 7,246 TCRs recovered using the 10× Chromium 5′-VDJ method.

### Defining cell state networks using TCR clonotype and transcriptomic data

Custom R scripts were used to define relationships between TCR and gene expression information. Single cell TCR data were merged with transcriptomic data using the common cell nucleotide barcode incorporated during reverse transcription. All clonotype analysis was performed using cells with one single productive TCRa and TCR chain and transcriptomic information (filtered using Scanpy), whereas all other cells were excluded.

To generate chord diagrams to illustrate the clonal relationships and cell states shared by Tregs from the same clonotype, the circlize package in R was used. Tissue of isolation and cell state definitions determined by Seurat were used to construct the outer and inner tracks of the chord diagram. Links were drawn for any two cells sharing a clonotype.

To generate network plots depicting the magnitude of clonotype sharing across cell states, the igraph package in R was used in Fig 5A. The frequency of clonotypes across cell states was first calculated by summing the number of cell pairs shared between two cell states and dividing by the total number of clonotypes. This value was then used to determine edge widths connecting two cell states, or vertices. Only spleen and lung Tregs were included in this analysis, as we recovered low numbers of gut Treg TCRs.

### Pseudotime analysis of Treg differentiation

Monocle2 (2.8.0) was used to order cells according to pseudotime. To generate the input data for the analysis, we combined cells from IL-2M and isotype conditions because (1) cell state classifications were determined independently of treatment and (2) we reasoned that the developmental trajectory should be preserved across treatment condition. We removed cells (i.e., C6 Tregs) whose primary marker genes were driven by proliferative status and not by their relative status along Treg-specific differentiation, although an independent analysis including these cells revealed that they are dispersed evenly throughout the trajectory manifold, which is expected because both resting and activated cell states are capable of proliferation. The gene list used for pseudotime analysis was filtered to include genes expressed in greater than 10% of all cells in the data (5,311 genes). We then used Monocle DDRTree to perform dimension reduction and manifold constructing, representing the trajectory of Treg differentiation. The cells were then ordered along the manifold. Genes that varied the most along the pseudotime axis were determined using the Monocle function: differentialGeneTest(fullModelFormulaStr = "~sm.ns(Pseudotime)").

## Data Availability

The scRNA-seq data from this publication have been deposited to the European Molecular Biology Laboratory's European Bioinformatics Institute public repository (https://www.ebi.ac.uk/ena/browser/home), and assigned the accession code PRJEB36020 (https://www.ebi.ac.uk/ena/browser/view/PRJEB36020).

## Supplementary Information

## Acknowledgements

We thank Olga Pryshchep, Hyun Ra, Nathan Deer, Jennifer Cheung, Laura Dieu, Aaron Fojas, Marisela Killian, Min-Zu Wu, Dev Bhatt, Oliver Homann, and Oh-Kyu Yoon for their technical support and intellectual input.

### Author Contributions

DR Lu: formal analysis and investigation.
H Wu: formal analysis and investigation.
I Driver: data curation, formal analysis, and investigation.
S Ingersoll: validation and methodology.
S Sohn: validation, visualization, and methodology.
S Wang: conceptualization and resources.
C-M Li: resources, software, investigation, and methodology.
H Phee: conceptualization, resources, formal analysis, validation, investigation, and visualization.

## Conflict of Interest Statement

DR Lu, S Sohn, S Wang, C-M Li, and H Phee are employees of Amgen. H Wu is currently employed at Pharmacyclics. I Driver is currently employed at Gordian Biotechnology. S Ingersoll is currently employed at Nektar therapeutics.

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
