## [Reviewer comments · Life Science Alliance]

Life Science Alliance

Dynamic changes in the regulatory T cell heterogeneity and function by murine IL-2 mutein

Daniel Lu, Hao Wu, Ian Driver, Sarah Ingersoll, Sue Sohn, Songli Wang, Chi-Ming Li, and Hyewon Phee

DOI: <https://doi.org/10.26508/lsa.201900520>

Corresponding author(s): Hyewon Phee, Amgen Research, Amgen Inc and Chi-Ming Li, Amgen Inc.

Review Timeline:

Submission Date:	2019-08-07
Editorial Decision:	2019-09-05
Revision Received:	2019-12-05
Editorial Decision:	2019-12-18
Revision Received:	2020-03-23
Editorial Decision:	2020-03-24
Revision Received:	2020-03-30
Accepted:	2020-03-31

Scientific Editor: Andrea Leibfried

Transaction Report:

September 5, 2019

Re: Life Science Alliance manuscript #LSA-2019-00520-T

Dr. Hyewon Phee
Northwestern University Feinberg School of Medicine
Department of Microbiology-Immunology
Chicago

Dear Dr Phee,

Thank you for submitting your manuscript entitled "Dynamic changes in the regulatory T cell heterogeneity and function by murine IL-2 mutein" to Life Science Alliance. The manuscript was assessed by expert reviewers, whose comments are appended to this letter.

As you will see, the reviewers appreciate your analysis but they also express reservation as to the completeness of the study and conclusive support for the claims made. Referee #1 states that more experimental work and in silico analyses on the transcriptome data would be required to improve the technical robustness of the work. This referee also points to discrepancies with published analyses and asks you to consider complementary treatment schemes. Referee #2 is overall more positive but requests better annotation of the novel IgG1-IL2 construct, as well as extended discussion of the findings. Referee #3 has additional concerns regarding data display, quantification and statistics.

We would thus like to invite you to submit a revised version of your manuscript to us, addressing the individual concerns raised by the reviewers. These seem all reasonable to address, but please do get in touch in case you would like to discuss an individual revision point further.

Thank you for this interesting contribution to Life Science Alliance. We are looking forward to receiving your revised manuscript.

Sincerely,

Daniel Klimmeck

Daniel Klimmeck, PhD
Scientific Editor
Life Science Alliance

B. MANUSCRIPT ORGANIZATION AND FORMATTING:

Reviewer #1 (Comments to the Authors (Required)):

Dynamic changes in the regulatory T cell heterogeneity and function by murine IL-2 mutein

Summary:

This manuscript by Lu, D. R. et al describes the effect of IL-2 mutein treatment on murine Tregs. First the authors compare the IL-2 mutein to currently available Treg selective IL-2 treatment modalities (IL-2 complex) and suggest that the IL-2 mutein is more specific for Treg expansion than IL-2 complex treatment, which also causes modest expansion of effector T cells. Using single cell RNA seq analysis, the authors then explore the alterations in Treg phenotype and subset distribution in IL-2 mutein treated mice versus controls. This reveals the expansion of ST2+41BB+ Tregs during IL-2M treatment. In vitro analysis of Treg subset suppressor function demonstrates that ST2+41BB+ Tregs are imbued with increased suppressor activity. The major feature of this manuscript is tracking expansion of Treg subsets during an IL-2M based treatment regimen. Analysis of single cell sequencing from Tregs doesn't reveal novel biology over previous reports however this study is able to identify the effect of IL-2M treatment on Treg subset proportions and provides a rationale for exploring IL-2M treatment clinically. Overall, the general conclusion that an IL2 mutein allows for selective expansion of specific Treg subsets is supported by the data. However, there are a number of concerns about specific studies and clarification is needed for some of the analysis. Finally, there is no indication that the single cell RNA-Seq data will be deposited with a public repository (GEO, etc.). Specific concerns are outlined below.

Major Points:

- The determination of clustering resolutions seems arbitrary. The heatmaps show somewhat undefined transcriptomic signatures between some clusters thus having 10 clusters may represent excessive resolution versus biology. There should be a comparison of different resolutions to show that the chosen resolution accurately represents biological heterogeneity.
- The data suggests that IL-2 mutein treatment leads to a shift between cTreg and eTreg phenotype- this largely disagrees with the bulk of published reports that say IL-2 is less important for eTregs, at least for their maintenance (For example, see work by D. Campbell and colleagues). How does this dataset reconcile these observations?
- Experimental repeats are lacking in a number of experiments (Figure 8 B-C, Supplemental Figure 1) and statistical analysis is lacking in many figures (Figures 3D, 5B, 6, 7B, 8B, Supplemental 3C). The number of separate experiments performed and total n for many figures is not complete.
- Given that the main focus of this manuscript is on generating a more functional ST2+41BB+ Treg subset following IL-2M treatment, other IL-2 based therapies should be analyzed for the expansion of this population (as in figure 8B). This will provide context into the effectiveness of the treatment versus alternatives. It should also be pretty straightforward to do.
- In vitro experiments should be performed to evaluate the effectiveness of IL-2M in stimulating non-Treg T cells and compared to other IL-2 treatment modalities. This should give definitive evidence that the IL-2M has superior on target (Treg) activity compared to other IL-2 therapies.
- There is no analysis of transcriptomic differences between clusters in isotype and IL-2M treated Tregs. It would be relevant to compare, for example, isotype treated cluster X and IL-2M treated cluster X to see what differences are being driven by the treatment.
- In figure 2A, PBS treated Tregs do not appear to suppress proliferation - only IL-2Mutein treated c cells suppress proliferation. Why do control Tregs not suppress as expected?

Minor Points:

- What is the IL-2 mutein? The description provided is relatively vague mentioning only an IL2-Fc fusion. Isotype and pbs are both mentioned as controls for IL-2M- this should be standardized.
- Some cell states are described by speculating on the biology but not backed up by primary experiments in this or other manuscripts- for example, is C7 representative of any known Treg biology? This relates in part to a better explanation of how a resolution yielding 10 subsets was chosen.

- Figure 1B-(CD25+Foxp3-): This should be displayed on a graph from 0-1.0%, 0-40% is impossible to interpret. The inset is too small to see. Likewise, the resolution of figure 3B, 4A, and supplemental figures 10 and 12 is poor - labels of cell subsets are impossible to read in 3 & 4 and gene names cannot be read in Sup figs 10 and 12..
- Figure 1C: There is a significant difference between effector cells with IL-2 mutein treatment. Perhaps a quantification of the fold expansion difference between Treg and Tconv would be helpful to visualize the relative effects on these 2 cell types.
- Is there a rationale behind why CD25- Tregs are also expanding- are these derived from differentiated CD25+ Tregs?
- "Expansion of CD25+Foxp3+ Tregs by IL-2M was comparable to IL-2/anti-IL2 antibody"- Supplemental 1C only quantifies Foxp3+ Tregs, not CD25+Foxp3+ Tregs.
- "After filtering and cross-sample normalization using Seurat(16), we recovered 17,097 spleen Tregs, 10,353 lung Tregs, and 4,458 gut Tregs across three replicates with roughly equivalent Tregs"- In supplemental Fig. 2C there are only 2 replicates for the gut Treg dataset however the text makes it appear as if there are 3 data sets for each organ. This should be corrected.
- "Additionally, Tregs expressed higher transcript levels of established Treg genes such as Foxp3, Il2ra, Ctla4, Ikzf2, and Nr1h3, while both cell types expressed similar levels of Cd4 (Supplemental Figure 3, B and C)."- Have stats been performed to confirm that these are robust differences?
- Figure 2: Could these experiments be performed with IL-2 mutein in vitro instead of in vivo? This could perhaps delineate between subset differences and direct effects of the mutein on Treg functionality.
- "Resting Tregs (C1) have high expression of lymphoid-tissue homing receptors (Ccr7, S1pr1, Sell)"- The data shown suggest this is perhaps true for CCR7 but less so for S1PR1 and not true for Sell.
- Figure 3: It seems surprising that CD62L is more highly expressed in activated cells versus resting Tregs- how does this reconcile with known phenotypes of eTreg vs cTreg?
- "Furthermore, C2- and C9-Tregs could be distinguished from each other, as C2-Tregs express more Nr1h3"- This comparison is not being statistically evaluated in the supplemental figure. Direct comparisons between groups should be made when statements are calling 2 groups different.
- Figure 4: Transcriptomic information should be compared between the same cluster in different organs as well as clusters within the same organ- not just done on a bulk basis. Similarly, transcriptomes should be compared between the same clusters in control and then separately for IL-2 mutein treated Tregs.
- "At the tissue level, the spleen and lung share a >40% frequency of C1-resting and minor frequencies of primed/activated and activated Tregs. Conversely, >80% of gut Tregs are activated and the majority are C5-Tregs (Figure 4, A and B)." This data should be quantified and analyzed statistically.
- "The coexpression of immunomodulatory genes (Cst7...."- The authors state that Cst7 is an immunomodulatory gene. However, Cst7 is a known gene downstream of TCR stimulation (Fassett MS, Jiang W, D'Alise AM, Mathis D, Benoist C. Nuclear receptor Nr4a1 modulates both regulatory T-cell (Treg) differentiation and clonal deletion. Proc Natl Acad Sci USA. 2012;109(10):3891-3896. doi: 10.1073/pnas.1200090109.) as well as a secreted factor from Tregs (Paracrine effect of regulatory T cells promotes cardiomyocyte proliferation during pregnancy and after myocardial infarction. Nature Communications 2018). Given these publications it may be more appropriate to include Cst7 in both the immunomodulatory and activation gene sets in this sentence.
- "Treatment with IL-2M shifted the frequency of Treg clusters, reducing C1-resting and C3-activated Tregs while elevating proliferation (C6),"- In figure 5, C6 is not statistically different between control and IL-2M.
- "The identification of transcriptional diversity among Tregs from the same clonal family was an interesting result, since we also find that pairs of T cells belonging to the same clonotype tend to

be transcriptionally correlated than randomly sampled pairs of Tregs at the population level, although this was a modest effect(20)"- It should be pointed out that this result is recapitulating previous results published in ref-20.

- Figure 6: This figure would be improved by quantitative analysis. How do we know these are really different rather than differences in TCR coverage perhaps?
- Figure 7A-B: This data should be statistically compared between control and IL-2 mutein treated mice. Additionally, it is unclear if this data has been normalized for TCR recovery rates- if not this should be performed prior to quantification.
- "Given that IL-2M increases clonal Treg expansion, we also examined how IL-2M influences the localization of CF Tregs by comparing the frequency of CF Tregs that were shared across tissues versus within the same tissue. While the percentage of inter-tissue CFs remained the same in both isotype and IL-2M conditions (3.9% versus 4.2%, respectively), the percentage of CFs found only in one tissue was nearly doubled (3.0% versus 5.8%, respectively)." A figure should be referenced for the origin of these numbers.
- Figure 7C-E: The signature of the most differentiated cells is somewhat surprising- it seems as though the remaining cycling cells (Mki67hi) are also expressing markers of terminal differentiation (gzmb). Is it thought that the most terminally differentiated cells would be represented by highly proliferative cells?
- "As expected, C1-resting Tregs occupied earliest period of pseudotime."- Is the "resting" Treg subset being chosen as the starting place of the pseudotime or is this being determined in an unbiased way? If chosen one cannot make statements about the analysis "beginning" at resting Tregs.
- "At the latest periods in pseudotime/differentiation, we observed two distinct differentiation branchpoints consisting of C5-Tregs at one terminus and C4/C8 Tregs at the other (Supplemental Figure 12B)." - How are the data sets being integrated? This is not mentioned in the methods and integration methods can affect clustering/pseudotime analysis (see Efficient integration of heterogeneous single-cell transcriptomes using Scanorama, Nature Biotechnology, 2019).
- "Of the thirty most variant genes identified from this analysis, four major gene modules were identified that correspond to cell-state classifications (Figure 7E)." - What analysis is being used to produce the thirty most variant genes- is it the genes that are the most dynamic over pseudotime?
- "Overall, analysis of clonal Treg differentiation trajectories suggests IL-2M promotes differentiation into the terminally differentiated C4 and C8 Tregs state by expanding through C2 and C3 intermediate states in the spleen and lung."- In figure 5C, C3 is decreased in IL-2M treated mice while C2 is increased. From what data is this conclusion being drawn? Pseudotime analysis of isotype and IL-2M treated mice should be compared.
- Figure 7D: Could the scales be changed for each subset so that the distribution can be visualized more clearly? Many of the subsets are imperceptible.
- "Pseudotime analysis demonstrated that bifurcation of Treg differentiation leading into the C4/C8 activated state, showing enrichment of genes in the Module 3 (Figure 7E). Among those genes, Il1rl1 and Tnfrsf9 are highly expressed in the Module 3."- It is unclear what the authors are trying to say here, could this be clarified?
- Figure 8: Does ST2 or 41BB stimulation affect Treg suppressor capability or are these just markers for the most functionally suppressive Treg subsets?
- Figure 8B: The amount of ST2 and/or 41BB+ cells should be quantified over repeated mice.
- Figure 8C: Additional quantification should be performed by calculating "% divided" as a measure of Treg suppressor function. This experiment should be repeated as well given that only 2 data points are being compared.
- "Proliferation of effector T cells was suppressed by Tregs expressing either ST2 or 4-1BB, but ST2+4-1BB+ Foxp3+ Tregs displayed the most superior suppression (Figure 8C)"- It can only be said, from the data in 8C, that ST2+41BB+ are superior to ST2+41BB-. ST2+41BB+ are not

statistically different from ST2-41BB+.

- "Interestingly, the development of Klrp1+ Tregs requires extensive IL-2R signaling."- There is no reference for this statement (although presumably it links to ref 19?). In fact there are other studies that say eTregs are less dependent on IL-2 signaling (KLRG1 expression identifies short-lived Foxp3+ Treg effector cells with functional plasticity in islets of NOD mice, 2017 Autoimmunity; CCR7 provides localized access to IL-2 and defines homeostatically distinct regulatory T cell subsets, 2014 JEM).
- "Trajectory analysis also identifies a bifurcation in Treg differentiation after IL-2M treatment, which either differentiate into suppressive Il10+Rora+ C5 Tregs, which are most prevalent in the gut, or into C4/C8 Tregs that are prominent in the spleen and lungs."- Is this bifurcation dependent on IL-2M treatment? If so, this data is not being displayed.
- Supplemental Figure 1: Is the number of CD8+ T cells not different between PBS and IL-2M? The number of Tregs being detected also seems very low- what organ is this? Spleen?
- Supplemental Figure 4: Perhaps similarity in differentially expressed genes could also be shown here to bolster the argument for data set similarity.
- Supplemental Figure 5A: C2 visually looks quite similar to C1 and C3 looks like C5. C6 also looks like it is a proliferating cluster of C4. It seems as though this data is perhaps over-clustered and producing signatures which are not truly unique. Also, C10 expresses IFN γ and CD8a- could these be contaminating CD8 cells? A better description of how the resolution was chosen to establish 10 clusters is needed.
- Supplemental Figure 5B: These comparisons seem somewhat arbitrary. Could some of these differences between clusters be derived from differences in tissue distribution?
- Supplemental Figure 8&9 are titled the same thing thus these could be combined into 1 figure.
- Supplemental Figure 10: Given that splenic and lung Tregs look almost identical, is there a way to validate that true lung, non-circulating, Tregs were used for the single cell analysis?
- Supplemental Figure 11: Would this analysis be more statistically accurate if the "different clonotype" group had the same number of events as the "Same clonotype" group.
- The exact number and description of single cell library preps for each sample needs to be defined for this study. A supplementary figure or table showing this would be informative. This figure should also show what libraries were then integrated for further analysis. Please include the methods that were used to integrate (ex. Suerat CCA + version, ScanPy aggregation method) - a detailed description of the pipeline would also be a very helpful supplemental figure. There are two different sequencing machines mentioned in the methods - HiSeq4000 and NovaSeq 6000 - please identify what libraries were sequenced on what machine.
- In Figure 7C, you show pseudotime/trajectory analysis of cells from both Il2M and isotype conditions. Why are you not showing these conditions separately as well? I am not sure that the data shown is supporting the statement on page 14 "Overall, analysis of clonal Treg differentiation trajectories suggests IL-2M promotes differentiation into the terminally differentiated C4 and C8 Tregs state by expanding through C2 and C3 intermediate states in the spleen and lung".

In summary, these studies will be of interest and the major conclusions are largely supported by the data but there are many issues (many but not all of which are relatively minor) that should be addressed.

Reviewer #2 (Comments to the Authors (Required)):

The manuscript of Lu et al. describes the effect of a novel IL2 derived drug on Tregs fate. They used a single-cell RNA-seq approach to follow the differentiation of FOXP3+ cells in IL-2 mutein (IL-2M) treated mice. The authors demonstrated several predicted cell states and showed that a

subpopulation of ST2+ 4-1BB+ FOXP3+ cells is induced both in lung and spleen that poses a strong suppressive action in vitro. The manuscript is well written and contains novel data and discuss relevant aspects for the use of IL-2 muteins for autoimmunity control. It may be considered for publishing in Life Science Alliance after major modifications.

Major concerns

The manuscript discusses the effect of a novel construction of IL-2 mutant (N88D) fused to an Fc moiety. There is no mention of the format of this novel immunobiological. In the Introduction and in the Discussion sections, authors refer to a citation (Peterson et al., 2018), that uses a germline coding human IgG1 fused with a mutated IL-2 at the carboxi terminus. Is that same molecule used in this work? There is no mention about that, neither in methods or results. Is that a human IgG/IL-2 used for the mouse experiments. The format and origin of the novel molecule should be described properly. Was that novel molecule already described in another report? Then, it should be properly cited.

The effect of the novel IL-2 mutein is compared to an isotype IgG as control. It would be interesting if the results were also compared with the wild-type IL-2. How comparable are the observed results with the action of the wild-type IL-2? Is there any evidence that the proposed Treg heterogeneity induced with the mutein is different from the wild-type IL-2? Is the observed heterogeneity restricted to the use of this novel mutein? This discussion is missing in the manuscript. Authors could test the effect of IL-2 on key subpopulation to address this questioning.

Results section contains a lot of discussions, making the Discussion section repetitive. Authors should rewrite and reduce redundancy.

Minor points

The authors characterize a few Treg populations and some of them express inflammatory markers. Does it mean that some of those subpopulations are not suppressive regulatory cells? Authors could comment on that, since the appearance of non-suppressive subpopulation may hinder clinical use.

On Figure 5A, IL-2 mutein treated mice are compared to control mice, or Isotype treated control mice. On Figure 6 authors use the term Isotype treated. Please use these terms uniformly.

Reviewer #3 (Comments to the Authors (Required)):

The main question addressed in this manuscript is how a half-life extended mutant form of IL-2 (IL-2M) impacts the phenotypic and functional heterogeneity of Tregs in diverse tissues. This question is relevant and interesting because low dose IL-2 therapies are being tested to induce tolerance in several auto-immune diseases. The authors addressed this question by combining single cell RNA-seq with TCR profiling of Tregs isolated from spleen, lungs or gut of mice injected with IL-2M or IgG Fc isotype as a control. This work revealed unique gene signatures shared between spleen and lungs Tregs as well as distinct activation profiles of gut Tregs. Based on TCR profiling, the authors

uncovered a migratory axis across tissues in response to IL-2M. They also identified a population of activated ST2+Tregs that expands following IL-2M that suppresses T conventional cells robustly in vitro. Overall this work was well performed and provides new insights into the relationships between Foxp3+ Treg activation states and their phenotypic heterogeneity in different tissues during homeostasis and after IL-2M stimulation.

Several issues should be addressed to improve the clarity of this study.

- 1) The authors mentioned that IL-2M has an extended half-life, but this was not defined.
- 2) It appears all studies were performed at day 4 post IL-2M. Why choose this time point? Does this coincide with maximal Treg expansion?
- 3) In Figure 1, most of the panels are too small to easily read and the insert in Fig. 1B is impossible to read. The text within Figs. 2A and 2B are also unreadable.
- 4) The authors should show some representative flow plots for Fig. 1, perhaps as a supplement, in order to understand how the bar graphs were generated.
- 5) For scRNAseq, an adjusted p value of <0.01 was used, which seems reasonable. However, an average log₂-fold change expression >0.3 does not seem very stringent. Please comment on why this was chosen.
- 6) For the legend to Figure 4, it should be corrected to state "individual cells are colored by Treg state classification from Figure 3" instead of Fig. 2, as stated.

RE: Point-by-point response to Reviewer's comments**General comments by reviewers: Provide further information about the murine IL2 mutein used in this study.**

IL2-mutein is a half-life-extended mouse IgG2a Fc fusion protein of a mutant form of mouse IL2, in which mouse IL2 is engineered to improve selective binding towards Tregs. In short, the mouse IL-2 mutein we generated is similar to the human form of long-lived IL2 (human IgG-(human IL-2N88D)₂), which was reported by Peterson et al (1). In this report, authors used an effector-silent human IgG1 to increase half-life and also the N88D mutation in human IL2, which decreased binding to IL2R β and allowed selective binding of the human IL2 mutein to the high affinity IL2R $\alpha\beta\gamma$ on Tregs.

Recombinant wild-type IL2 has very short half-life. The serum half-life of human IL2 in man after i.v. administration is notoriously short, with value of 6.9 min for recombinant human IL-2(2). Clearance of recombinant human IL-2 was even faster in mice, with a serum half-life of about 1.6 min when administered i.v (3). Frequent administration of large amount of recombinant IL2 has been used to maintain therapeutic serum levels, but capillary leak syndrome was emerged as one of the major side effects of this frequent high dose IL-2 therapy (4).

To extend the half-life of murine IL-2, we fused the Fc portion of mouse IgG2a with a linker, which is similar to the effector-silent form of human IgG1 used in human IL2 mutein. The N297G mutation was introduced in muIgG2a to inhibit ADCC (antibody-dependent cell cytotoxicity) activity of the mouse IgG2a Fc in order to generate an "effector-silent" version (5). The last amino acid residue of mouse IgG2a, K, is usually cleaved off by carboxypeptidase activity in monoclonal antibody. To maintain homogeneity, the last K residue of the mouse IgG2a was deleted (**Data Figure 1**).

When we administered murine IL-2M, it showed dose proportional exposure increase. Serum concentration of IL-2M reached its maximum concentration at 6 hours after administration, then gradually decreased over 7 days. During this time, serum concentration of IL-2M was maintained over 0.1 nM. After 4 days of IL-2M administration, the serum concentration of IL-2M was between 0.1 -1 nM, which was concentration of IL-2M expanding Tregs selectively over Tconv *in vitro* experiment. Furthermore, we measured expansion of Tregs at Day 4 and 7 and found that expansion of Tregs by IL-2M was maximum after 4 days following administration.

Data Figure 1
In order to achieve selectivity towards Tregs, we sought for mutations that attenuate interaction between IL2R β (CD122) and murine IL2, similar to the N88D mutation in human half-life extended IL2 mutein(1). It is because the interaction of IL2 with IL2R β in the intermediate IL2R $\beta\gamma$ receptor in conventional T cells expands Tconv cells following IL-2 treatment.

The IL-2R exists in two functional forms. The high affinity IL-2R assembles when IL-2 is captured by the IL2R α (CD25) subunit that in turns facilitates additional binding to signaling receptors- IL-2R β (CD122) and IL-2R γ (CD132)-, forming IL-2R $\alpha\beta\gamma$. The high affinity IL-2R $\alpha\beta\gamma$ is found in Tregs (CD25+Tregs) but also found in lower levels on effector T conventional cells (CD25+ Tconv). The intermediate affinity IL-2R (IL2R $\beta\gamma$) is expressed by multiple hematopoietic lineage cells, including conventional T cells as well as NK cells(6). When high dose of IL2 is used, it interacts with the high affinity IL-2 receptor (IL-2 $\alpha\beta\gamma$) as well as the intermediate affinity IL2 receptor (IL2R $\beta\gamma$) and activates both Tconv and Tregs. Only when IL2 is used in low-dose, selectivity towards Treg can be achieved. However, low-dose IL2 therapy cannot achieve greater expansion of Tregs due to the limitation of the amount can be administered because the window of the dose to achieve Treg selectivity over Tconv is very narrow.

Zurawski et al reported series of papers describing important residues of murine IL2 for interacting with IL2R α and IL2R β (7,8). Among those residues, the D34 and N103 residues of mouse IL2 were shown to be important for IL2R β (CD122) binding. The N103 residue of mouse IL2 mutein is corresponding to the N88 of human IL2(1). Thus, we mutated D34 and N103 residues of murine IL2 to D34S and N103D to reduce the interaction between IL2 and IL2R β .

In addition to D34S and N103D mutations in mouse IL2, we incorporated two additional mutations (C140A and P51T) of mouse IL2 to facilitate manufacturability. The C140A is a mutation corresponding to C125A in human IL2 to avoid aggregation. This mutation was also incorporated in aldesleukin (low-dose IL2) and was reported previously (9). The P51T is a mutation specific for the murine IL2 and it was used to prevent clipping during production.

Intracellular staining of phosphorylated STAT5 was performed to determine the activity of mouse IL-2M *in vitro*. Because IL-2M was mutated to decrease its binding to IL2R β , the activity of IL-2M was attenuated compared with wild-type recombinant IL2. However, IL-2M induced phospho-STAT5 in CD25+Foxp3+ Tregs that express the high affinity IL2R $\alpha\beta\gamma$ in a dose dependent manner, while it did not induce phospho-STAT5 in CD25-Foxp3- Tconv cells that express the intermediate IL2R, IL2R $\beta\gamma$ (**Data Figure 2**). In contrast, wild-type recombinant IL2 induced phosphor-STAT5 in CD25-Foxp3- Tconv as well

Data Figure 2

as Tregs.

Because the mutations we generated were intended to reduce the interaction of mouse IL2 with IL2R β but maintain interaction with IL2R α (CD25) to allow binding to the high affinity IL2R (ILR $\alpha\beta\gamma$) on Tregs, IL-2M slightly activated CD25+Foxp3- Tconv cells, although the degree of activation was markedly reduced compared with wild-type rmlL2 (**Data Figure 3**).

Data Figure 3

We hope this information will answer reviewer's questions about murine IL-2M.

We now added a paragraph describing mouse IL-2M (manuscript pg. 5) and it is highlighted in yellow. In addition, we added Data Figures 1-3 as the revised Supplemental Figure 1A-B.

From here, we would like to respond with detailed response to each reviewer's comments.

Reviewer #1 (Comments to the Authors (Required)):

Summary:

This manuscript by Lu, D. R. et al describes the effect of IL-2 mutein treatment on murine Tregs. First the authors compare the IL-2 mutein to currently available Treg selective IL-2 treatment modalities (IL-2 complex) and suggest that the IL-2 mutein is more specific for Treg expansion than IL-2 complex treatment, which also causes modest expansion of effector T cells. Using single cell RNA seq analysis, the authors then explore the alterations in Treg phenotype and subset distribution in IL-2 mutein treated mice versus controls. This reveals the expansion of ST2+41BB+ Tregs during IL-2M treatment. In vitro analysis of Treg subset suppressor function demonstrates that ST2+41BB+ Tregs are imbued with increased suppressor activity. The major feature of this manuscript is tracking expansion of Treg subsets during an IL-2M based treatment regimen. Analysis of single cell sequencing from Tregs doesn't reveal novel biology over previous reports however this study is able to identify the effect of IL-2M treatment on Treg subset proportions and provides a rationale for exploring IL-2M treatment clinically. Overall, the general conclusion that an IL2 mutein allows for selective expansion of specific Treg subsets is supported by the data. However, there are a number of concerns about specific studies and clarification is needed for some of the analysis. **“Finally, there is no indication that the single cell RNA-Seq data will be deposited with a public repository (GEO, etc.)”** Specific concerns are outlined below.

Response: We will deposit the single cell RNA-seq data into the EMBL-EBI public repository, European Nucleotide Archive (<https://www.ebi.ac.uk/ena>). We will confirm submission of the data upon acceptance of the manuscript.

Major Points:

1. “ The determination of clustering resolutions seems arbitrary. The heatmaps show somewhat undefined transcriptomic signatures between some clusters thus having 10 clusters may represent excessive resolution versus biology. There should be a comparison of different resolutions to show that the chosen resolution accurately represents biological heterogeneity.”

We acknowledge that the determination of optimal clustering resolutions is difficult in single-cell RNA-seq and must balance a cautious approach that minimizes technical, non-biological variances with an approach that can reveal novel cell populations of biological interest. For this reason, we (1) analyzed the data iteratively at different clustering resolutions and (2) used quantitative measures (thresholding p-values and log-fold changes in expression) to identify critical genes.

The optimal clustering resolution in Seurat was determined by clustering integrated single-cell expression data at ten different resolutions from 0.1 to 1.0 using the “*resolution*” parameter in the FindClusters() function. At each resolution, the top marker genes of the cluster containing the fewest cells were evaluated against previously published literature to support inclusion. The determination of 0.6 for the resolution parameter was made because this was the lowest resolution (i.e. smallest cluster number) at which C10 IFN γ ^{hi} Tregs - which express ex-Treg markers and were previously described by Daniel V *et al.* (10) and others- could be identified (**Data Figure 4**). Lower resolutions merged this population with other cell states, masking the distinct gene expression profile of this population (see Fig. 3C, Sup Fig. 5C, Sup. Fig. 9C, and **Data Figure 5**).

DATA FIGURE 4.

DATA FIGURE 5.

Despite the appearance of “undefined transcriptomic signatures between some clusters” in some parts of the heatmap, we believe our cluster definitions represent an appropriate resolution to reveal true biology. First, marker genes for clusters were compared against existing scientific literature for their role in Treg function prior to inclusion. Second, marker genes were quantitatively selected based on statistical significance after differential expression using MAST, as well as the application of a log-fold cut-off to further triage marker genes. Third, the Louvain algorithm identifies clusters based on similarity networks, which are defined by combinations of genes and not by single genes. Thus, while the heatmap may not adequately represent these differences, the differences between clusters can be captured by complementary analyses. For example, while clusters C1 and C2 appear to overlap in a large number of gene signatures (Sup. Fig. 5A), the trajectory analysis clearly places C1 and C2 cells and assigns distinct pseudotime values to these clusters, highlighting their distinction (Sup. Fig. 12B). Based on these multiple lines of evidence, we believe that our clustering resolution best represents the current knowledge of Treg biology and the biological variance present in the dataset.

2. “The data suggests that IL-2 mutein treatment leads to a shift between cTreg and eTreg phenotype- this largely disagrees with the bulk of published reports that say IL-2 is less important for eTregs, at least for their maintenance (For example, see work by D. Campbell and colleagues). How does this dataset reconcile these observations?”

We did not conclude that IL-2 is more important for eTregs. What we reported was that murine IL-2 mutein (IL-2M) treatment increased the proportion of C4- and C8 -Tregs (which mostly resemble eTregs), while reducing the proportion of the C1-Treg (which resemble cTregs). Because we are looking at Day 4 after IL-2M treatment, this is the result of the action of IL-2M on Tregs. Thus, just by looking at the result of the action of IL-2M, it is difficult to determine which population of Tregs is responsible for this result.

Data from Campbell’s lab showed requirement of IL-2 in cTregs by using IL-2R α KO mice as well as blocking IL-2 (11). In the same paper, authors also showed that, when cTregs were transferred to the mice activated by TCR and LPS, cTregs were activated and became eTregs, indicating that cTregs responded to stimuli and changed their state to eTregs.

Based on Monocle analysis, the C1 Tregs (naïve cTregs -like phenotype) occupied the earliest period of pseudotime. The C2 Tregs (primed and activated state) occupied the intermediate state and the C4 and C8 Tregs (eTregs-like phenotype) occupied the terminal state. Because IL-2M treatment increased the proportion of C4 and C8-Tregs while reducing C1-Treg, it is likely that C1-Treg responded to the IL-2M, then differentiated into the C4/C8 state. If this is case, our data is consistent with data from Campbell’s group, in which cTregs changed their state to eTregs following stimulation.

3. “Experimental repeats are lacking in a number of experiments (Figure 8 B-C, Supplemental Figure 1) and statistical analysis is lacking in many figures (Figures 3D, 5B, 6, 7B, 8B, Supplemental 3C). The number of separate experiments performed and total n for many figures is not complete.”

We performed the following statistical analysis and revised the text accordingly.

- 1) Figure 3D: The Wilcoxon rank sum test was applied to Figure 3D to test for significant expression changes.
- 2) Figure 5B: Quantification of Figure 5B is shown in Figure 5C. Statistical analysis using Welch's t-test was included in Figure 5C to test for significance in Treg cell states across treatment groups for all three tissues.
- 3) Figure 6: Fisher's exact test was applied to Figure 6 to test for independence between clonal family frequency and IL-2 mutein treatment. For example, if the number of clonal families we observed in Tregs is "dependent" upon IL-2 mutein treatment, p-value will be <0.05 . If the two variables (# clonal families and IL2M treatment) are "independent", p-value will be > 0.05 .
- 4) Figure 7B: Fisher's exact test was applied to Figure 7B to test for independence between the frequency of clonal cell state pairs and IL-2 mutein treatment.
- 5) Figure 8B: We added Figure 8C to show statistical analysis with multiple replicates.
- 6) Figure 8B-E: We added the following statement in the Figure Legend and highlighted: "Results are representative of two independent experiments, using 2-3 mice in each experiment."
- 7) Supplemental Figure 1B; Data are representative of two independent experiment.
- 8) Supplemental Figure 1D; Results are representative of at least two independent experiment using 3 mice per each group.
- 9) Supplemental Figure 1E-F; Results shown are from three mice from each condition from one experiment.
- 10) Supplemental Figure 3C: Differential expression was performed using MAST, and the results of calculated p-values were added to Supplemental Figure 3C.

4. "Given that the main focus of this manuscript is on generating a more functional ST2+41BB+ Treg subset following IL-2M treatment, other IL-2 based therapies should be analyzed for the expansion of this population (as in figure 8B). This will provide context into the effectiveness of the treatment versus alternatives. It should also be pretty straight forward to do."

The main focus of the manuscript is to describe the effect of IL-2M on heterogenous populations of Tregs and to determine how it alters the landscape of Tregs, but not to compare the effectiveness of the IL-2M over other IL-2 based therapies. In fact, it is difficult to compare the effectiveness of different modalities without information of pharmacokinetics or pharmacodynamics, thus it is beyond the scope of current paper.

For example, recombinant mouse IL-2 has very short half-life *in vivo* as previously described. In addition, we showed that high dose of recombinant mouse IL-2 did expand Tconv as well as Tregs *in vitro*. Thus, at higher concentration, IL-2 will expand Tregs as well as Tconv.

Likewise, comparing effectiveness of the IL-2/anti-IL2 antibody (IL-2C) and IL-2M will not be meaningful without determining pharmacokinetics or pharmacodynamics of the two agents. For example, for IL-2/anti-IL2 (IL-2C) administration *in vivo*, we needed to administer the IL-2C daily for three days as previously reported(12). If we increase the dose or frequency of either IL-2M or IL-2C, it will further increase Tregs.

Having said that, we think that it will be meaningful to show whether the response induced by IL-2M is also induced by IL-2C. We determined the percent of ST2+ cells within Tregs from spleens followed by IL-2M or IL-2C *in vivo* treatment because ST2 was increased in both C4- and C8-clusters following IL-2M treatment. Similar to IL-2M, IL-2C also increased the percent of ST2+ Tregs (**Data Figure 6**).

5. “*In vitro* experiments should be performed to evaluate the effectiveness of IL-2M in stimulating non-Treg T cells and compared to other IL-2 treatment modalities. This should give definitive evidence that the IL-2M has superior on target (Treg) activity compared to other IL-2 therapies.”

We agree that performing *in vitro* experiment comparing IL-2M and murine recombinant WT IL2 (rmIL-2) for their activity on Treg vs. non-Tregs population would clarify the selective effect of IL-2M on Tregs.

Thus, we determined phosphorylated STAT5 using intracellular FACS analysis from CD25+Foxp3+ Tregs and CD25-Foxp3- Tconv cells following increasing dose of recombinant mouse wild-type recombinant mouse IL-2 (rmIL-2) or mouse IL-2M treatment (**Data Figure 2 and 3**). Wild-type rmIL-2 resulted in phosphorylation of STAT5 in both Tregs and Tconv at high dose. In contrast, IL-2M resulted in phosphorylation STAT5 only in Tregs but not in Tconv.

6. “There is no analysis of transcriptomic differences between clusters in isotype and IL-2M treated Tregs. It would be relevant to compare, for example, isotype treated cluster X and IL-2M treated cluster X to see what differences are being driven by the treatment.”

For the cluster analysis, we integrated all single-cell RNA-seq data from isotype control-treated and IL-2M- treated mice prior to cell clustering, and this approach was driven by our biological understanding of the effect of IL-2/IL-2M signaling on Tregs. IL-2M treatment is NOT creating a new Treg cell state, but it results in over-representation or under-representation of existing states among the Treg differentiation continuum. Therefore, we applied clustering to the combined isotype- and IL-2M-treated

cells to properly observe changes in the representation of cell states. This led to such findings as the decrease in C1-Tregs and increase in C4/C8-Tregs following IL-2M treatment.

In our analysis, each cluster is defined by transcriptional differences when all samples are aggregated. Thus, comparing isotype-treated cluster X and IL-2M treated cluster X will not generate meaningful gene sets within that cluster X, because cluster X is already defined by expression of unique gene sets from both isotype- and IL-2M-treated cells.

7. “ In figure 2A, PBS treated Tregs do not appear to suppress proliferation - only IL-2Mutein treated cells suppress proliferation. Why do control Tregs not suppress as expected?”

This question is due to confusion of the labeling of Figure 2A. The first panel of Figure 2A showed that Tregs from PBS-treated mice were added into naïve T cells in *in vitro* Treg suppression assay. Second panel showed that Tregs from IL-2M treated mice were added into naïve T cells. The third panel showed that proliferation of naïve T cells without any Tregs. **Compared with the third panel, Tregs from PBS treated mice (from the first panel) did suppress proliferation of naïve T cells, but not as much as Tregs from IL-2M treat mice (second panel).** The experimental procedure was described in the Figure legend, and we revised labeling in the revised Figure 2 to clarify this.

Minor Points:

1. “What is the IL-2 mutein? The description provided is relatively vague mentioning only an IL2-Fc fusion. Isotype and pbs are both mentioned as controls for IL-2M- this should be standardized”.

We provided information from Pg 1-3.

2. “Some cell states are described by speculating on the biology but not backed up by primary experiments in this or other manuscripts- for example, is C7 representative of any known Treg biology? This relates in part to a better explanation of how a resolution yielding 10 subsets was chosen”.

We provided answers for the resolution from the Major point #1, Pg 4-6.

Because gut Tregs contain primarily the C7-Tregs and the C7-Tregs express early response inflammatory genes, we speculate that these Treg cells may provide tolerance towards gut microbiome or food antigens.

3. “Figure 1B-(CD25+Foxp3-): This should be displayed on a graph from 0-1.0%, 0-40% is impossible to interpret. The inset is too small to see. Likewise, the resolution of figure 3B, 4A, and supplemental figures 10 and 12 is poor - labels of cell subsets are impossible to read in 3 & 4 and gene names cannot be read in Sup figs 10 and 12”.

The low resolution of Figures is due to embedded PNG files in the Word document. To address this concern, we provide the following files.

Figure 1B: We changed the y axis of the graph to 0-1.0% and removed the insert.

Figure 3B and 4A: We provided Tiff files with increased resolution (600 dpi).

Supplemental Figure 10 and 12: We enlarged the figures and provide Tiff files with increased resolution (600 dpi).

4. *“Figure 1C: There is a significant difference between effector cells with IL-2 mutein treatment. Perhaps a quantification of the fold expansion difference between Treg and Tconv would be helpful to visualize the relative effects on these 2 cell types”.*

To address this concern, we now provide the fold increase in the revised Supplemental Figure 1C. Although there was significant difference between CD25⁺Foxp3⁻ effector T conv cells, the fold increase of cell numbers was approximately 1.2 fold in lymph node and spleen and 0.7 fold in the lung while the fold increase of cell numbers of CD25⁺Foxp3⁺ Tregs were approximately 5 fold in lymph node and spleen and 7.5 fold in the lung following IL-2M treatment.

5. *“Is there a rationale behind why CD25⁻ Tregs are also expanding- are these derived from differentiated CD25⁺ Tregs?”*

We also wondered about this. Previously it has been reported that CD25^{low} Foxp3⁺ T cells share phenotypic features resembling conventional CD25^{high} Foxp3⁺ Tregs in human (13). This report concluded that the number of CD25^{low}Foxp3⁺ T cells was correlated with the proportion of CD25^{high}Foxp3⁺ T cells in cell cycle, suggesting that CD25^{low}Foxp3⁺ T cells represent a subset of Tregs that are derived from CD25^{high}Foxp3⁺ T cells. Because IL-2M increased the CD25^{high}Foxp3⁺ T cells in cell cycle, we speculate that the increase in CD25^{low}Foxp3⁺ T cells may be derived from CD25^{high}Foxp3⁺ T cells.

6. *“Expansion of CD25⁺Foxp3⁺ Tregs by IL-2M was comparable to IL-2/anti-IL2 antibody”- Supplemental 1C only quantifies Foxp3⁺ Tregs, not CD25⁺Foxp3⁺ Tregs”.*

We revised the text to Foxp3⁺Tregs and highlighted.

7. *“After filtering and cross-sample normalization using Seurat(16), we recovered 17,097 spleen Tregs, 10,353 lung Tregs, and 4,458 gut Tregs across three replicates with roughly equivalent Tregs”- In supplemental Fig. 2C there are only 2 replicates for the gut Treg dataset however the text makes it appear as if there are 3 data sets for each organ. This should be corrected”.*

We revised it and highlighted:

“After filtering and cross-sample normalization using Seurat(16), we recovered 17,097 spleen Tregs, 10,353 lung Tregs, and 4,458 gut Tregs across three replicates (except gut Tregs treated with IL-2M (n=2)) with roughly equivalent Tregs in mouse IgG Fc isotype control (Iso)- and IL-2M-treated conditions (16,152 and 15,756 cells, respectively).”

8. *“Additionally, Tregs expressed higher transcript levels of established Treg genes such as Foxp3, Il2ra, Ctla4, Ikzf2, and Nrp1, while both cell types expressed similar levels of Cd4 (Supplemental Figure 3, B and C).”- Have stats been performed to confirm that these are robust differences?”*

Yes. This question is similar to Major point #3 from Reviewer 1 (pg. 7), and the concern was addressed there. Differential expression was performed using MAST, and the results of calculated p-values were added to Supplemental Figure 3C.

9. "Figure 2: Could these experiments be performed with IL-2 mutein *in vitro* instead of *in vivo*? This could perhaps delineate between subset differences and direct effects of the mutein on Treg functionality".

We believe that using Tregs from mice treated with IL-2M *in vivo* is better representation of the effect of IL-2M *in vivo*.

10. "Resting Tregs (C1) have high expression of lymphoid-tissue homing receptors (Ccr7, S1pr1, Sell)"- The data shown suggest this is perhaps true for CCR7 but less so for S1PR1 and not true for Sell

Thanks for noticing this error. This statement is true for S1PR1, but not Sell. S1PR1 was significantly upregulated in C1. However, Sell was significantly upregulated only in C2 and C6, but not in C1 cluster (Data Figure 7). Thus, we revised the text to "Resting Tregs (C1) have high expression of lymphoid-tissue homing receptors (Ccr7 and S1pr1) ".

Data Figure 7

11. Figure 3: It seems surprising that CD62L is more highly expressed in activated cells versus resting Tregs- how does this reconcile with known phenotypes of eTreg vs cTreg?

Expression of CD62L(Sell) is significantly high in C2 (primed/activated cluster) and C6 (proliferative) clusters, but it was markedly reduced in the spleen-enriched C3 (activated) and gut-enriched C5 clusters. Thus, downregulation of Sell in activated Tregs (eTregs) holds true at least in C3 and C5 activated Tregs. For the highly proliferative C6-Tregs, active cell cycling of these cells may cause more Sell to be expressed.

Downregulation of Sell on activated eTregs occurs transcriptionally as well as by protein shedding. It is possible that the protein expression of Sell in the C6 cluster may be decreased due to shedding despite higher transcriptional expression.

12. "Furthermore, C2- and C9-Tregs could be distinguished from each other, as C2-Tregs express more Nrp1"- This comparison is not being statistically evaluated in the supplemental figure. Direct comparisons between groups should be made when statements are calling 2 groups different."

We performed differential expression using MAST of C2 versus C9. After differential expression analysis of Nrp1 in C2 versus C9, we find that C2 is significantly upregulated relative to C9 (log2expression=0.33, FDR-adjusted p-value=0.0119). This log-fold change in expression is shown in Supplemental Figure 7B already. We have reported this p-value in Sup. Figure 7A.

13. "Figure 4: Transcriptomic information should be compared between the same cluster in different organs as well as clusters within the same organ- not just done on a bulk basis. Similarly, transcriptomes should be compared between the same clusters in control and then separately for IL-2 mutein treated Tregs."

Discussed in Major point #6. The purpose of this figure is to show global differences between tissues. We already showed differences in transcriptomes in individual clusters.

14. "At the tissue level, the spleen and lung share a >40% frequency of C1-resting and minor frequencies of primed/activated and activated Tregs. Conversely, >80% of gut Tregs are activated and the majority are C5-Tregs (Figure 4, A and B)." This data should be quantified and analyzed statistically."

This paragraph is supposed to depict Figure 5 not Figure 4. Quantification is provided in Figure 5B. We changed the text accordingly.

15. "The coexpression of immunomodulatory genes (Cst7...."- The authors state that Cst7 is an immunomodulatory gene. However, Cst7 is a known gene downstream of TCR stimulation (Fassett MS, Jiang W, D'Alise AM, Mathis D, Benoist C. Nuclear receptor Nr4a1 modulates both regulatory T-cell (Treg) differentiation and clonal deletion. Proc Natl Acad Sci USA. 2012;109(10):3891-3896. doi: 10.1073/pnas.1200090109.) as well as a secreted factor from Tregs (Paracrine effect of regulatory T cells promotes cardiomyocyte proliferation during pregnancy and after myocardial infarction. Nature Communications 2018). Given these publications it may be more appropriate to include Cst7 in both the immunomodulatory and activation gene sets in this sentence.

We removed Cst7 in the text following reviewer's suggestion.

16. "Treatment with IL-2M shifted the frequency of Treg clusters, reducing C1-resting and C3-activated Tregs while elevating proliferation (C6),"- In figure 5, C6 is not statistically different between control and IL-2M.

In Figure 5, C6 is not statistically different between control and IL-2M despite there being an increase in the percentage of C6 cells after IL2-M treatment. This result is shown in the representation of P-value results in Figure 5C. The statement was revised to "Treatment with IL-2M shifted the frequency of Treg

clusters, reducing C1-resting and C3-activated Tregs while increasing primed/activated (C2) and activated Treg states (C4 and C8) (Figure 5B-C)."

17. "The identification of transcriptional diversity among Tregs from the same clonal family was an interesting result, since we also find that pairs of T cells belonging to the same clonotype tend to be transcriptionally correlated than randomly sampled pairs of Tregs at the population level, although this was a modest effect(20)"- It should be pointed out that this result is recapitulating previous results published in ref-20.

We revised the text following reviewer's suggestion and added "as previously reported".

18. Figure 6: This figure would be improved by quantitative analysis. How do we know these are really different rather than differences in TCR coverage perhaps?

This comment was already addressed in Reviewer #1 Major point #3. Regarding the reviewer's comment on the distinction of true differences in clonotype sharing from differences in TCR sampling, Fisher's exact test, which we use to test for independence between clonotype sharing and treatment, accounts for sampling differences by using the hypergeometric distribution to calculate expected frequencies. Therefore, we conclude that the differences in clonotype sharing in Tregs treated by IL2-mutectin are statistically meaningful.

19. Figure 7A-B: This data should be statistically compared between control and IL-2 mutectin treated mice. Additionally, it is unclear if this data has been normalized for TCR recovery rates- if not this should be performed prior to quantification.

Thank you for the insightful comments. The data shown in Figures 7A-B are already normalized by differences in TCR recovery rate to mitigate for sample size differences, which is why the values are shown in percentages.

Based on the reviewer's suggestion, we use Fisher's exact test to test for independence between matched pairs of cell states (*i.e.* C4-to-C4, or C4-to-C6) across treatment groups (IL2-mutectin versus isotype). Since Fisher's exact test is performed on observed data by using the hypergeometric distribution, we utilized the counts (not percentages) to determine statistical significance. We reasoned that comparing each matched pair of cell states provided a more accurate method to assess statistical differences than rank ordering tests on the mean of each population (*e.g.* Wilcoxon-signed rank test), since IL-2 accelerates the differentiation/expansion of Tregs toward specific activated cell states that are inherent to their regulatory circuitry(14). Therefore, IL2-mutectin treatment should not only increase the mean number of cell state pairs with shared clonotypes; it should also increase the frequency of specific pairs of cell states.

20. "Given that IL-2M increases clonal Treg expansion, we also examined how IL-2M influences the localization of CF Tregs by comparing the frequency of CF Tregs that were shared across tissues versus within the same tissue. While the percentage of inter-tissue CFs remained the same in both Isotype and IL-2M conditions (3.9% versus 4.2%, respectively), the percentage of CFs found only in one tissue was nearly doubled (3.0% versus 5.8%, respectively)." A figure should be referenced for the origin of these numbers.

Figure 7C was generated and statistics were calculated using Fisher's exact test to compare the counts between intra/inter-tissue clonal family clonotypes and singletons (non-clonal family clonotypes) for each treatment group.

21. Figure 7C-E: The signature of the most differentiated cells is somewhat surprising- it seems as though the remaining cycling cells (Mki67hi) are also expressing markers of terminal differentiation (gzmb). Is it thought that the most terminally differentiated cells would be represented by highly proliferative cells?

Yes, we believe that the proliferation of Tregs coincides with the expression of terminal differentiation markers. The cells that are selected to reach a terminally differentiated state are recruited by the host immune system to carry out effector functions; this causes these selected cells to rapidly expand as they differentiate in order to rapidly respond to host signals. Therefore, it makes sense that terminally differentiated cells are also expressing markers of cell proliferation.

22. "As expected, C1-resting Tregs occupied earliest period of pseudotime."- Is the "resting" Treg subset being chosen as the starting place of the pseudotime or is this being determined in an unbiased way? If chosen one cannot make statements about the analysis "beginning" at resting Tregs.

As mentioned in the Monocle 2 paper (15), Monocle uses reversed-graph embedding to define a manifold that represents the structure of cell differentiation, but the package allows the user to define the root node (or root cells) based on an understanding of the underlying biology and calculates pseudotime in an unbiased manner based on the distance from that root node.

From the paper:

"Monocle 2 allows users to conveniently select a tip of the tree as the root and then transverses the tree from the root, computing the geodesic distance of each cell to the root cell, which is taken as its pseudotime, and assign branch or segment simultaneously."

This approach can be problematic of cell types of unknown etiology, but we believe this is not the case for Tregs. Given the unbiased approach in structuring the trajectory manifold, the following lines of evidence suggest that the root node was correctly selected:

- 1) Previous studies have shown that Tregs express secondary lymphoid organ-homing genes such as Ccr7 and S1pr1 prior to activation, suggesting that cells expressing these genes appear earlier in pseudotime (16,17). Tregs then downregulate these genes upon activation and extravasation into tissues.
- 2) Previous studies have all found genes such as Tnfrsf9, Gzmb, Il1rl1, Il10, and Areg to be involved in effector Treg functions (18,19). Furthermore, we show in this study that Tregs expressing Tnfrsf9 and Il1rl1 show more suppressive, effector activity than Tregs that do not express these genes, suggesting that cells expressing these genes should occur later in pseudotime.

We revised the text accordingly and it is highlighted.

"Given the gene expression profiles and robust lineage relationship of the C2-primed states and C3/C4/C8 activated states, we used pseudotime analyses using Tregs with recovered TCRs ($n=3,600$ cells) to define their developmental relationship (Figure 7, D and E, Supplemental Figure 12A). Treg cell states occupied distinct territories in pseudotime. We defined the node enriched for C1-resting Tregs as

the root node (pseudotime $t=0$), and pseudotime values were assigned in an unbiased manner to the manifold based on the distance from that root node."

23. "At the latest periods in pseudotime/differentiation, we observed two distinct differentiation branchpoints consisting of C5-Tregs at one terminus and C4/C8 Tregs at the other (Supplemental Figure 12B)."- How are the data sets being integrated? This is not mentioned in the methods and integration methods can affect clustering/pseudotime analysis (see Efficient integration of heterogeneous single-cell transcriptomes using Scanorama, Nature Biotechnology, 2019).

Thank you for the comment and paper reference. We are acutely aware of the negative impact of integrating different datasets without proper sample integration. However, data integration prior to pseudotime analysis in Figures 7C-E was not necessary, since cells used for the analysis were from the same mouse. Therefore, they were prepared in the same library prep batch and by the same library prep method. After identifying the Treg cell differentiation trajectory in this batch of clonally related Tregs (confirmed by TCR sequence), we confirmed that this trajectory pattern was not unique to this batch by repeating the trajectory analyses for the remaining batches of Tregs from which we did not sequence the TCR (shown in Supplemental Figures 12C-D), and we observed the same manifold structure in all batches. We believe that if batch effects were significant across library prep batches and methods, the manifolds would appear drastically different between batches and/or methods.

We would like to note that data integration was performed prior to Louvain clustering analysis using the CCA method in Seurat, since this analysis incorporated cells from three mice, different library prep batches, and two different library prep methods (10x 3' V2 and 10x 5'). This is already described in the Methods section.

24. "Of the thirty most variant genes identified from this analysis, four major gene modules were identified that correspond to cell-state classifications (Figure 7E)."- What analysis is being used to produce the thirty most variant genes- is it the genes that are the most dynamic over pseudotime?

We used the differentialGeneTest() function in Monocle to test for genes that were the most dynamic over pseudotime:

```
differentialGeneTest(fullModelFormulaStr = "~sm.ns(Pseudotime)"))
```

For clarification, we have added the following statement to the methods section:

Genes that varied the most along the pseudotime axis were determined using the Monocle function: differentialGeneTest(fullModelFormulaStr = "~sm.ns(Pseudotime)")).

25. "Overall, analysis of clonal Treg differentiation trajectories suggests IL-2M promotes differentiation into the terminally differentiated C4 and C8 Tregs state by expanding through C2 and C3 intermediate states in the spleen and lung."- In figure 5C, C3 is decreased in IL-2M treated mice while C2 is increased. From what data is this conclusion being drawn? Pseudotime analysis of isotype and IL-2M treated mice should be compared.

The Monocle analysis was used to understand differentiation trajectory of the Treg clusters, specially between the C2-primed state and C3/C4/C8- activated states. Based on this analysis, we found that the

C4 and C8 Treg states, which were expanded by IL-2M, occupied the terminal state in the differentiation pseudotime. Meanwhile, the C2 and C3 were dispersed throughout the intermediate points in the pseudotime. Furthermore, C3-Tregs express immunomodulatory genes (Izumo1r, Nt5e) as well as TNFRSF9 and CD83 at a medium level without expression of effector proteins compared with other activated states. Based on this result, we concluded that the C3-Treg is an intermediate state- they are activated but not terminally differentiated.

However, Monocle analysis cannot distinguish which differentiation events are result of IL-2M treatment. We agree with the reviewer that, because C3-Tregs was decreased following IL-2M treatment, and there is no evidence that differentiation occurs through C3. Thus, we will remove C3 in the text and revise the statement to the following:

“Overall, analysis of clonal Treg differentiation trajectories suggests IL-2M promotes differentiation into the terminally differentiated C4 and C8 Treg state by expanding **through an intermediate state such as C2** in the spleen and lung.”

26. Figure 7D: Could the scales be changed for each subset so that the distribution can be visualized more clearly? Many of the subsets are imperceptible.

More detailed distribution of each subset in Figure 7D (now Figure 7E) can be found in Supplemental Figure 12B.

27. "Pseudotime analysis demonstrated that bifurcation of Treg differentiation leading into the C4/C8 activated state, showing enrichment of genes in the Module 3 (Figure 7E). Among those genes, Il1r1 and Tnfrsf9 are highly expressed in the Module 3."- It is unclear what the authors are trying to say here, could this be clarified?

We revised to “ Pseudotime analysis demonstrated enrichment of genes in 4 different Modules (Figure 7F). Among those genes, Il1r1 and Tnfrsf9 are highly expressed in the Module 3.”.

28. Figure 8: Does ST2 or 41BB stimulation affect Treg suppressor capability or are these just markers for the most functionally suppressive Treg subsets?

We did not stimulate Tregs with ST2 or 41BB. These are used as surface markers to identify the majority of Tregs expanded by IL2M.

29. Figure 8B: The amount of ST2 and/or 41BB+ cells should be quantified over repeated mice.

We now added Figure 8C, which is quantification of Figure 2 from repeated mice. We revised the Figure legend accordingly.

30. Figure 8C: Additional quantification should be performed by calculating "% divided" as a measure of Treg suppressor function. This experiment should be repeated as well given that only 2 data points are being compared.

We now provided Percent divided (%) in the right panel of Figure 8D. These experiments were repeated two times using sorted Treg populations from two individual mice. Figure legend was revised accordingly.

31. "Proliferation of effector T cells was suppressed by Tregs expressing either ST2 or 4-1BB, but ST2+4-1BB+ Foxp3+ Tregs displayed the most superior suppression (Figure 8C)"- It can only be said, from the data in 8C, that ST2+41BB+ are superior to ST2+41BB-. ST2+41BB+ are not statistically different from ST2-41BB+.

Although difference in dilution of CTV gMFI was not significant between ST2+41BB+ vs. ST2-41BB+ subsets, percent divided (%) of ST2+41BB+ was statistically different from ST2-41BB+, thus this statement is correct.

32. "Interestingly, the development of Klrp1+ Tregs requires extensive IL-2R signaling."- There is no reference for this statement (although presumably it links to ref 19?). "

Yes, the reference is ref 19.

"In fact there are other studies that say eTregs are less dependent on IL-2 signaling (KLRG1 expression identifies short-lived Foxp3+ Treg effector cells with functional plasticity in islets of NOD mice, 2017 Autoimmunity; CCR7 provides localized access to IL-2 and defines homeostatically distinct regulatory T cell subsets, 2014 JEM). "

This is the same point raised in Major point # 2 and we addressed there.

33. "Trajectory analysis also identifies a bifurcation in Treg differentiation after IL-2M treatment, which either differentiate into suppressive Il10+Rora+ C5 Tregs, which are most prevalent in the gut, or into C4/C8 Tregs that are prominent in the spleen and lungs."- Is this bifurcation dependent on IL-2M treatment? If so, this data is not being displayed.

At the latest periods in pseudotime/differentiation, we observed two distinct differentiation branchpoints consisting of C5-Tregs at one terminus and C4/C8 Tregs at the other. The pseudotime analysis can show developmental relationship of Treg states and trajectory of differentiation but cannot distinguish whether differentiation is dependent on IL-2M. Thus, we revised the statement to:

"Trajectory analysis also identifies a bifurcation in Treg differentiation after IL-2M treatment, which either differentiate into suppressive Il10⁺Rora⁺ C5 Tregs, which are most prevalent in the gut, or into C4/C8 Tregs that are prominent in the spleen and lungs.

34. Supplemental Figure 1: Is the number of CD8+ T cells not different between PBS and IL-2M?

We performed One-Way ANOVA for multiple comparisons using GraphPad Prism 7.04, and the numbers CD8+ T cells were not statistically significantly different between PBS and IL-2M.

The number of Tregs being detected also seems very low- what organ is this? Spleen?

It is spleen. To calculate the cell numbers, we used percentages of CD4 or CD8 T cells from total cells. For this particular experiment, percentages of live cells within spleen were around 70% among all samples, which resulted in overall smaller numbers of cells. If we use percentages of CD4 or CD8 T cells from live lymphocyte gate to calculate the cell number, the overall numbers are increased, but the trend

remained the same. Although it is interesting to see differential effect on CD8, it is not relevant for the current study. Thus, we removed the CD4 and CD8 data and revised Supplemental Figure 1.

35. Supplemental Figure 4: Perhaps similarity in differentially expressed genes could also be shown here to bolster the argument for data set similarity.

The clusters in Supplemental Figure 4 were determined after integrating all replicates together; therefore, differential expression by each replicate would be redundant and unnecessary. A more robust way to test for similarity between replicates is by testing whether the observed cell frequencies in each cell state are significantly different between replicates. Thus, we applied Fisher's exact test to compare for significant differences in cell state frequencies for each replicate and found that there were no significant differences. These results have been added to Supplemental Figure 4.

36. Supplemental Figure 5A: C2 visually looks quite similar to C1 and C3 looks like C5. C6 also looks like it is a proliferating cluster of C4. It seems as though this data is perhaps over-clustered and producing signatures which are not truly unique. Also, C10 expresses IFN γ and CD8a- could these be contaminating CD8 cells? A better description of how the resolution was chosen to establish 10 clusters is needed.

We addressed this concern about resolution in Major Point #1.

C2 is different from C1 in the sense that they weakly expressed genes that are enriched in C1-resting as well as other activated Treg clusters. Furthermore, Monocle analysis showed the C2-Tregs are dispersed through the manifold as opposed to the C1-resting cluster occupied at the initial starting point.

Moreover, we don't believe the C10- cluster is contamination, because there are other reports demonstrating a small set of CD8+ Tregs that has suppressive effect for the self-reactive CD4 T cells(20,21) In particular, it was reported that a small population of CD8⁺CD25⁺FOXP3⁺ T cells was found both in mice and humans and they can suppress CD4 effector T cell proliferation(21).

37. Supplemental Figure 5B: These comparisons seem somewhat arbitrary. Could some of these differences between clusters be derived from differences in tissue distribution?

We would like to see differential gene expression regardless of the location within the cluster. Some clusters are shown in all organs, but some clusters are over-represented by an organ. Differences in tissue distribution were shown in Figure 4.

38. Supplemental Figure 8&9 are titled the same thing thus these could be combined into 1 figure.

Combining two figures will make figures even smaller and it will be difficult to read each gene.

39. Supplemental Figure 10: Given that splenic and lung Tregs look almost identical, is there a way to validate that true lung, non-circulating, Tregs were used for the single cell analysis?

To take a look at differential expression between lung and spleen, we need to take a look at Figure 4B not Supplemental Figure 10. Figure 4B showed more genes differentially expressed between spleen and

lung following Isotype control treatment. Supplemental Figure 10 showed differential genes between spleen and lung following IL-2M treatment.

Figure 6A upper right graph showed some of the TCR shared between spleen and lung (inter-tissue clonotypes). These Tregs with shared TCR in the lung would be circulating Tregs. If the lung Tregs are contamination of all circulating Tregs, the composition of each cluster should be similar with spleen Tregs. However, the lung Tregs are quite different from spleen Tregs (for example, the lung Tregs lack the C3-Treg cluster).

40. Supplemental Figure 11: Would this analysis be more statistically accurate if the "different clonotype" group had the same number of events as the "Same clonotype" group.

We believe the analysis performed using a larger number of samples represents a more accurate approach, since more replicates are sampled. 10,000 randomly sampled correlations were used in the "different clonotype" group to represent an exceedingly large sampling of that group and remove any ambiguity about sufficient sampling depth. This approach is similar to that used by Zemmour, Nat Imm, 2018 (22), which compares 14 "same clonotypes" to 1000 randomly sampled "different clonotypes".

To explore this analysis further, we also performed statistical analysis between equal numbers of events in both groups and find that the results are the same as the analysis shown in Supp. Figure 11.

41. The exact number and description of single cell library preps for each sample needs to be defined for this study. A supplementary figure or table showing this would be informative. This figure should also show what libraries were then integrated for further analysis. Please include the methods that were used to integrate (ex. Suerat CCA + version, ScanPy aggregation method) - a detailed description of the pipeline would also be a very helpful supplemental figure. There are two different sequencing machines mentioned in the methods - HiSeq4000 and NovaSeq 6000 - please identify what libraries were sequenced on what machine.

We have included an excel spreadsheet table as a Supplemental Table for the number of preps, sample processing batches, library prep methods, and sequencing instruments. Moreover, we will add information about the Seurat version (2.4) and Scanpy version (0.94) to the Methods section as well.

42. In Figure 7C, you show pseudotime/trajectory analysis of cells from both IL2M and isotype conditions. Why are you not showing these conditions separately as well? I am not sure that the data shown is supporting the statement on page 14 "Overall, analysis of clonal Treg differentiation trajectories suggests IL-2M promotes differentiation into the terminally differentiated C4 and C8 Tregs state by expanding through C2 and C3 intermediate states in the spleen and lung".

The clusters were defined by combining both IL-2M and isotype control. By doing so, we could see which clusters are increased by IL-2M treatment. Because we define clusters based on all possible clusters of genes under IL-2M and isotype control, we can define which cluster is increased following treatment. Figure 7 is performed by TCR analysis and Supplemental Figure was performed using all T cells. For the statement, we already discussed in Minor Point #22.

In summary, these studies will be of interest and the major conclusions are largely supported by the

data but there are many issues (many but not all of which are relatively minor) that should be addressed.

Reviewer #2 (Comments to the Authors (Required)):

The manuscript of Lu et al. describes the effect of a novel IL2 derived drug on Tregs fate. They used a single-cell RNA-seq approach to follow the differentiation of FOXP3+ cells in IL-2 mutein (IL-2M) treated mice. The authors demonstrated several predicted cell states and showed that a subpopulation of ST2+ 4-1BB+ FOXP3+ cells is induced both in lung and spleen that posses a strong suppressive action in vitro. The manuscript is well written and contains novel data and discuss relevant aspects for the use of IL-2 muteins for autoimmunity control. It may be considered for publishing in Life Science Alliance after major modifications.

Major concerns

“ The manuscript discusses the effect of a novel construction of IL-2 mutant (N88D) fused to an Fc moiety. There is no mention of the format of this novel immunobiological. In the Introduction and in the Discussion sections, authors refer to a citation (Peterson et al., 2018), that uses a germline coding human IgG1 fused with a mutated IL-2 at the carboxi terminus. Is that same molecule used in this work? There is no mention about that, neither in methods or results. Is that a human IgG/IL-2 used for the mouse experiments. The format and origin of the novel molecule should be described properly. Was that novel molecule already described in another report? Then, it should be properly cited.

We discussed the nature of the murine IL-2M in pg 1-3.

“The effect of the novel IL-2 mutein is compared to an isotype IgG as control. It would be interesting if the results were also compared with the wild-type IL-2. How comparable are the observed results with the action of the wild-type IL-2?”

We provided answers in pg 1-3.

“Is there any evidence that the proposed Treg heterogeneity induced with the mutein is different from the wild-type IL-2? Is the observed heterogeneity restricted to the use of this novel mutein? This discussion is missing in the manuscript. Authors could test the effect of IL-2 on key subpopulation to address this questioning.”

Because of the short pharmacokinetics of the wild type IL-2, the effect of murine IL-2M cannot be compared properly *in vivo*. In addition, as shown in Data Figure 2, wild type IL-2 stimulate Tregs as well as Tconv *in vitro*, so data will be difficult to interpret.

“Results section contains a lot of discussions, making the Discussion section repetitive. Authors should rewrite and reduce redundancy.”

We revised the discussion to reduce redundancy. Removed sections were highlighted and crossed in the text. Removed discussion can be found in pg 19-22 from the manuscript.

Minor points

1. "The authors characterize a few Treg populations and some of them express inflammatory markers. Does it mean that some of those subpopulations are not suppressive regulatory cells? Authors could comment on that, since the appearance of non-suppressive subpopulation may hinder clinical use."

We don't have any evidence that the minor subpopulations of Foxp3+ Tregs are not suppressive. Although they express inflammatory markers, it might be a sign of activation of Tregs. For example, a minor population of CD8+CD25+Foxp3+ Treg resembling the C10-Cluster were found in mice and humans (20,21) and shown to be suppressive towards CD4 effector T cell proliferation (21). We also discussed this point from response to Reviewer 1, Minor point # 22.

2. "On Figure 5A, IL-2 mutein treated mice are compared to control mice, or Isotype treated control mice. On Figure 6 authors use the term Isotype treated. Please use these terms uniformly."

We used the term "Isotype control-treated mice" uniformly.

Reviewer #3 (Comments to the Authors (Required)):

The main question addressed in this manuscript is how a half-life extended mutant form of IL-2 (IL-2M) impacts the phenotypic and functional heterogeneity of Tregs in diverse tissues. This question is relevant and interesting because low dose IL-2 therapies are being tested to induce tolerance in several auto-immune diseases. The authors addressed this question by combining single cell RNA-seq with TCR profiling of Tregs isolated from spleen, lungs or gut of mice injected with IL-2M or IgG Fc isotype as a control. This work revealed unique gene signatures shared between spleen and lungs Tregs as well as distinct activation profiles of gut Tregs. Based on TCR profiling, the authors uncovered a migratory axis across tissues in response to IL-2M. They also identified a population of activated ST2+Tregs that expands following IL-2M that suppresses T conventional cells robustly *in vitro*. Overall this work was well performed and provides new insights into the relationships between Foxp3+ Treg activation states and their phenotypic heterogeneity in different tissues during homeostasis and after IL-2M stimulation.

Several issues should be addressed to improve the clarity of this study.

1. "The authors mentioned that IL-2M has an extended half-life, but this was not defined."

The nature of half-life extension of the murine IL-2M was explained in page 1. When we administered IL-2M, it showed dose-proportional exposure increase. Serum concentration of IL-2M reached its maximum concentration at 6 hours after administration, then gradually decreased over 7 days. During this time, serum concentration of IL-2M was maintained over 0.1 nM. After 4 days of IL-2M administration, the serum concentration of IL-2M was between 0.1 -1 nM, which was concentration of IL-2M expanding Tregs selectively over Tconv *in vitro* experiment. Furthermore, we measured expansion of Tregs at Day 4 and 7 and found that expansion of Tregs by IL-2M was maximum after 4 days following administration.

2. " It appears all studies were performed at day 4 post IL-2M. Why choose this time point? Does this

coincide with maximal Treg expansion?"

Yes, we chose day 4 because it coincided with maximal Treg expansion.

3. " In Figure 1, most of the panels are too small to easily read and the insert in Fig. 1B is impossible to read. The text within Figs. 2A and 2B are also unreadable."

We removed the insert of the Figure 1B and increased the font size of the whole Figure 1. For Figure 2, we provided high resolution Tiff file (600 dpi).

4. " The authors should show some representative flow plots for Fig. 1, perhaps as a supplement, in order to understand how the bar graphs were generated."

We provided this data as new Supplemental Figure 1C.

5. " For scRNAseq, an adjusted p value of <0.01 was used, which seems reasonable. However, an average log2-fold change expression >0.3 does not seem very stringent. Please comment on why this was chosen."

There are several reasons behind the use of this threshold:

(1) The threshold of 0.3 was used for finding cluster markers because the goal was to find markers that denote Treg *states* and not Treg *subsets*. In the study we are primarily concerned with Treg *states*, which we define as distinct activation potentials defined by their cell expression profiles. We define Treg *subsets* as Treg cell types under the control of distinct master transcriptional regulators, which we do not focus on here. Since expression differences in 'cell state' are often more subtle than in 'cell subsets', especially when comparing within the same cell type (i.e. Tregs), we use a lower threshold than studies that compare between different cell types (where the gene expression differences are more distinguishable).

(2) Given the large sample sizes in single-cell data, since one cell = one sample, it is common for genes to be significantly differentially expressed with an FDR-corrected p-value of <0.01 despite changes in log2 expression of >0 and <0.3. The threshold we set was an additional filter on top of the p-value to ensure that marker genes we selected would be more easily observable when testing these genes with other assays that may have less sensitivity.

(3) Other single cell RNA-seq papers utilize similar cut-offs for differential expression, especially those that analyze differences between groups of cells that belong to the same cell type. For example, Miragaia et al. 2019 (23) used an even less stringent log-fold change cutoff of 0.25 and an adjusted p-value cutoff of 0.05 to analyze Tregs from different tissues.

6. "For the legend to Figure 4, it should be corrected to state "individual cells are colored by Treg state classification from Figure 3" instead of Fig. 2, as stated."

We corrected.

References

1. Peterson LB, Bell CJM, Howlett SK, Pekalski ML, Brady K, Hinton H, et al. A long-lived IL-2 mutein that selectively activates and expands regulatory T cells as a therapy for autoimmune disease. *J Autoimmun.* 2018;95:1–14.
2. Lotze MT, Matory YL, Ettinghausen SE, Rayner AA, Sharrow SO, Seipp CA, et al. In vivo administration of purified human interleukin 2. II. Half life, immunologic effects, and expansion of peripheral lymphoid cells in vivo with recombinant IL 2. *J Immunol Baltim Md 1950.* 1985 Oct;135(4):2865–75.
3. Chang AE, Hyatt CL, Rosenberg SA. Systemic administration of recombinant human interleukin-2 in mice. *J Biol Response Mod.* 1984 Oct;3(5):561–72.
4. Rosenstein M, Ettinghausen SE, Rosenberg SA. Extravasation of intravascular fluid mediated by the systemic administration of recombinant interleukin 2. *J Immunol Baltim Md 1950.* 1986 Sep 1;137(5):1735–42.
5. Tao MH, Morrison SL. Studies of aglycosylated chimeric mouse-human IgG. Role of carbohydrate in the structure and effector functions mediated by the human IgG constant region. *J Immunol Baltim Md 1950.* 1989 Oct 15;143(8):2595–601.
6. Liao W, Lin J-X, Leonard WJ. Interleukin-2 at the crossroads of effector responses, tolerance, and immunotherapy. *Immunity.* 2013 Jan 24;38(1):13–25.
7. Zurawski SM, Zurawski G. Mouse interleukin-2 structure-function studies: substitutions in the first alpha-helix can specifically inactivate p70 receptor binding and mutations in the fifth alpha-helix can specifically inactivate p55 receptor binding. *EMBO J.* 1989 Sep;8(9):2583–90.
8. Zurawski SM, Vega F, Doyle EL, Huyghe B, Flaherty K, McKay DB, et al. Definition and spatial location of mouse interleukin-2 residues that interact with its heterotrimeric receptor. *EMBO J.* 1993 Dec 15;12(13):5113–9.
9. Klein C, Waldhauer I, Nicolini VG, Freimoser-Grundschober A, Nayak T, Vugts DJ, et al. Cergutuzumab amunaleukin (CEA-IL2v), a CEA-targeted IL-2 variant-based immunocytokine for combination cancer immunotherapy: Overcoming limitations of aldesleukin and conventional IL-2-based immunocytokines. *Oncoimmunology.* 2017;6(3):e1277306.
10. Daniel V, Trojan K, Adamek M, Opelz G. IFN γ ⁺ Treg in-vivo and in-vitro represent both activated nTreg and peripherally induced aTreg and remain phenotypically stable in-vitro after removal of the stimulus. *BMC Immunol.* 2015 Aug 13;16:45.
11. Smigiel KS, Richards E, Srivastava S, Thomas KR, Dudda JC, Klonowski KD, et al. CCR7 provides localized access to IL-2 and defines homeostatically distinct regulatory T cell subsets. *J Exp Med.* 2014 Jan 13;211(1):121–36.
12. Webster KE, Walters S, Kohler RE, Mrkvan T, Boyman O, Surh CD, et al. In vivo expansion of T reg cells with IL-2-mAb complexes: induction of resistance to EAE and long-term acceptance of islet allografts without immunosuppression. *J Exp Med.* 2009 Apr 13;206(4):751–60.

13. Ferreira RC, Simons HZ, Thompson WS, Rainbow DB, Yang X, Cutler AJ, et al. Cells with Treg-specific FOXP3 demethylation but low CD25 are prevalent in autoimmunity. *J Autoimmun.* 2017 Nov;84:75–86.
14. Zheng Y, Josefowicz S, Chaudhry A, Peng XP, Forbush K, Rudensky AY. Role of conserved non-coding DNA elements in the *Foxp3* gene in regulatory T-cell fate. *Nature.* 2010 Feb 11;463(7282):808–12.
15. Qiu X, Mao Q, Tang Y, Wang L, Chawla R, Pliner HA, et al. Reversed graph embedding resolves complex single-cell trajectories. *Nat Methods.* 2017 Oct;14(10):979–82.
16. Schneider MA, Meingassner JG, Lipp M, Moore HD, Rot A. CCR7 is required for the in vivo function of CD4⁺ CD25⁺ regulatory T cells. *J Exp Med.* 2007 Apr 16;204(4):735–45.
17. Liu G, Burns S, Huang G, Boyd K, Proia RL, Flavell RA, et al. The receptor S1P1 overrides regulatory T cell-mediated immune suppression through Akt-mTOR. *Nat Immunol.* 2009 Jul;10(7):769–77.
18. Vaeth M, Wang Y-H, Eckstein M, Yang J, Silverman GJ, Lacruz RS, et al. Tissue resident and follicular Treg cell differentiation is regulated by CRAC channels. *Nat Commun.* 2019 12;10(1):1183.
19. Magnuson AM, Kiner E, Ergun A, Park JS, Asinovski N, Ortiz-Lopez A, et al. Identification and validation of a tumor-infiltrating Treg transcriptional signature conserved across species and tumor types. *Proc Natl Acad Sci U S A.* 2018 06;115(45):E10672–81.
20. Yu Y, Ma X, Gong R, Zhu J, Wei L, Yao J. Recent advances in CD8⁺ regulatory T cell research. *Oncol Lett.* 2018 Jun;15(6):8187–94.
21. Churlaud G, Pitoiset F, Jebbawi F, Lorenzon R, Bellier B, Rosenzweig M, et al. Human and Mouse CD8⁽⁺⁾CD25⁽⁺⁾FOXP3⁽⁺⁾ Regulatory T Cells at Steady State and during Interleukin-2 Therapy. *Front Immunol.* 2015;6:171.
22. Zemmour D, Zilionis R, Kiner E, Klein AM, Mathis D, Benoist C. Single-cell gene expression reveals a landscape of regulatory T cell phenotypes shaped by the TCR. *Nat Immunol.* 2018;19(3):291–301.
23. Miragaia RJ, Gomes T, Chomka A, Jardine L, Riedel A, Hegazy AN, et al. Single-Cell Transcriptomics of Regulatory T Cells Reveals Trajectories of Tissue Adaptation. *Immunity.* 2019 19;50(2):493-504.e7.

December 18, 2019

Re: Life Science Alliance manuscript #LSA-2019-00520-TR

Dr. Hyewon Phee
Amgen Discovery Research
Department of Oncology and Inflammation
1120 Veterans Blvd
South San Francisco, CA 94080

Dear Dr. Phee,

Thank you for submitting your revised manuscript entitled "Dynamic changes in the regulatory T cell heterogeneity and function by murine IL-2 mutein" to Life Science Alliance. The manuscript was assessed by the original reviewers again, whose comments are appended to this letter.

As you will see, while reviewer #2 appreciates the introduced changes, reviewer #1 thinks that many of her/his concerns have not been adequately addressed. Though addressing the remaining concerns requires a significant effort, it mainly relies on the re-analysis of the data already at hand (as well as a defined, not time-consuming experiment). While we usually only allow a single round of revision, we therefore concluded that we can grant another round in this case. However, it would be important to fully address the reviewer concerns, following the constructive input provided. I attach reviewer #1's report in word docx format to this email, to allow you to more easily follow and address the remaining concerns.

While further revising your work, please also pay attention to the following:

- all corresponding authors should link their profile in our submission system to their ORCID iDs, please make sure that this occurs
- please list 10 authors et al in your reference list
- please add callouts in the manuscript text for Fig. 4B, Supp. Fig. 1C, Supp. Fig. 8B, Supp. Fig. 9B
- please deposit the RNA-seq data and provide the accession code

Thank you for this interesting contribution to Life Science Alliance. We are looking forward to receiving your revised manuscript.

Sincerely,

Reviewer #1 (Comments to the Authors (Required)):

Review:

This resubmission of the manuscript by Lu. et al has made improvements that address many of our concerns. Inclusion of a detailed description of the IL2Mutein (IL2M) reagent was very helpful, along with including more details regarding experimental design, repeats and statistics. However, concerns remain on 2 major points. First, the authors state the goal of this manuscript is not to characterize the IL2M or determine its efficacy, but instead to determine the general effect of IL2-based treatments on Tregs. While it can be appreciated that this is perhaps outside the scope of this study, the mechanism of improved suppressor function is still not clearly demonstrated. The two, non-mutually exclusive, likely candidates for driving immune suppression are improved suppressor capacity induced by IL2M ligation, and/or expansion of a particular Treg subset with a higher suppressor capacity. A simple in vitro experiment suppression assay in the presence or absence of IL2M would allow the authors to determine whether this reagent enhances the suppressor function of Tregs. These experiments are relatively easy to do, take little time (< 1 week) and would provide at least some insights into how the IL2M mutein promotes Treg function. We still think this would be a good experiment to do. Second, several concerns remain about the analysis of the single cell RNA-seq data sets. The determination of the number of distinct subsets should be validated by some known or speculated biology. Some of the clusters appear very similar to each other and may be considered as falling into the same subset. The rationale from the authors for increasing resolution is based on the appearance of a subset of unknown importance that may even be a sort contaminant. Also, a comparison of transcriptomes between IL2M and control treated mice is a necessary component of this study. If IL2M induces transcriptional changes in a particular subset remains an open question that is easily addressable with the presented data. Based on the tSNE graphs presented, it appears like IL2M treated cells do occupy different regions within the same cluster, and thus likely have transcriptional changes induced by IL2M. Note that this second concern does not involve doing any additional experiments but simply analyzing data the authors already have. Addressing this second concern is essential for proper interpretation of the results presented.

RE: Point-by-point response to Reviewer's comments

The first section is very helpful in understanding what the IL2M is and the logic behind its generation.

The new data on intracellular staining of phosphorylated STAT5 is also very helpful.

Input on authors' answer to major point 1:

The rationale for setting the resolution appears to be that it allows one to detect a potentially novel population of CD8+ Tregs. However, it is unclear whether this particular population is relevant to the study at hand as the authors sorted on CD4+CD8- cells, thus the CD8+ population must have arisen via an imperfect sort. It is also unclear whether these cells actually express Foxp3. The described study (10) doesn't identify these cells as ex-Tregs and is done in human PBMCs so to compare this study to that one is a stretch. It seems most likely that this C10 population arises via contamination from CD4 or CD8 effectors- this is quite common in single cell data sets and the cluster in question is very small.

A heatmap should be added displaying the top 10 differentially regulated genes in each cluster compared to every other cluster. Many of the "marker" genes correlate to no known Treg subset or function (i.e. *Ass1*, *Stmn1*, *Dnajb1*). The resolution chosen is going to determine if 2 sets of cells are assigned to different clusters by the algorithm. However, this is implicit and is why they are put into 2 distinct clusters, so it doesn't have any bearing on whether this represents a truly biologically distinct population (circular argument). The pseudotime interpretation could go the other way as well- if a cluster of cells doesn't occupy any particular niche over pseudotime (i.e. clusters 2, 3, 7, 9 and 10) then could you argue that these don't represent any unique population? I don't think you can make this conclusion so the statement that having different pseudotime distributions (would need to be statically validated anyway) has no bearing on whether 2 clusters are distinct or not.

A more concrete description of the biology represented by the subsets should be provided or else another resolution should be chosen for the analysis.

Input on authors' answer to major point 2:

This is confusing because IL2 treatment still doesn't cause conversion of a cTreg into a eTreg- this requires activation of the cTregs. Why are the cTregs being activated during IL2M treatment? Why would they now be receiving TCR stimulation? Alternatively, please include a statement that it is also possible that IL2M preferentially expands distinct Treg subsets.

This is still not consistent with the literature on the effect of IL2 on Treg phenotype. Once again, could it be equally likely that the IL2M is just expanding these preexisting eTregs? This again, seems less likely given the requirements of eTregs. A better mechanistic understanding of this process should be provided to explain the results, rather than speculation based on the single-cell RNAseq datasets. Alternatively, a statement acknowledging that these differences could arise via preferential expansion of existing Treg subsets could be included.

Input on authors' answer to major point 3-4:

OK now.

Input on authors' answer to major point 5:
This is a nice result.

Input on authors' answer to major point 6:

Unless you have done a comparison between unstimulated and IL2M treated Treg subsets then you don't know if IL2M does not alter the transcriptome within subsets. There is precedent that Tregs with stronger STAT5 activation are better suppressors so this treatment certainly could be altering the transcriptome of Tregs of similar overall phenotypes. In fact, this is clearly visible in the tSNE plots when control and IL2M are being compared- IL2M treated cells within the same cluster appear in a distinct space from control cells in the same cluster. This analysis should be included in this manuscript.

Input on authors' answer to major point 7:

I agree that the panels in figure 2A make it look like the untreated control Tregs are suppressing proliferation of naïve T cells compared to no Tregs. However, the same problem remains in figure 2B, which seeks to quantify that difference - the CTV gMFI is still unchanged between the PBS and No Treg groups so this again suggests that the untreated Tregs are not suppressing proliferation. It might be better to compare % divided cells as the authors have done in other figures as that might give a more obvious difference. Otherwise, you are left with a representative example in figure 2A that looks convincing, but aggregate data (figure 2B) which shows no difference. The % IFN γ is mildly lower in PBS vs No Treg, which at least shows they are doing something. % Divided should be calculated here as well, not just CTV gMFI.

Input on authors' answer to minor point 1:

This was a very helpful addition to the paper.

Input on authors' answer to minor point 2:

This statement should be quantified as they are still a minor component of gut Tregs as presented in figures 4A and 5A.

Input on authors' answer to minor point 3:

Figure 1B: We changed the y axis of the graph to 0-1.0% and removed the insert.

=> As such- change the y-axis for the CD25-Foxp3 $^{+}$ graph as well to maybe 10-20% and the y-axis for CD25-Foxp3 $^{-}$ should go down to 0.

Input on authors' answer to minor point 4-8:

OK

Input on authors' answer to minor point 9:

It obviously is a better representation of the effect of IL2M in vivo but does not address mechanism effectively. The in vitro experiments would allow you to determine if one part of the mechanism involves enhanced suppressor function on a per cell basis (see first main concern in initial paragraph).

Input on authors' answer to minor point 10-17:

OK

Input on authors' answer to minor point 18:

There is still no quantitative analysis in this figure. What does total clonotype sharing refer to- this should be analyzed by clonality or diversity within a sample. It would be ideal to have repeated experiments for TCR sequencing so statistics can be drawn on these measures.

Input on authors' answer to minor point 19:

Again- a more robust test would be via diversity or clonality measurements for each experimental repeat.

Unless you know the clonality of the 2 data sets it is unclear if a simple % conversion is normalizing the data. If you say have lower diversity and more TCRs recovered from a data set then the relationship between the 2 data sets is non-linear.

Input on authors' answer to minor point 20-24:

OK

Input on authors' answer to minor point 25:

It would still be informative to compare the pseudotime trajectories for IL2M vs isotype. Certainly, you would expect to see a different distribution of cells over the pseudotime trajectories.

Input on authors' answer to minor point 26-28:

OK

Input on authors' answer to minor point 29:

This is a nice addition; however, all the individual data points should be visible in all bar graphs.

Input on authors' answer to minor point 30:

The figure legends say that this data has 2-3 mice per experiment and the experiment was performed twice. The exact n for each condition should be stated. All data points from all experiments should be displayed to indicate the biological variability in the experiment.

Input on authors' answer to minor point 31:

OK.

Input on authors' answer to minor point 32:

Authors: ...This is the same point raised in Major point # 2 and we addressed there.

=> This is not really the same point as it is a different marker and different set of literature. Can you just confirm that Klrp1 is expressed in these clusters or induced by IL2M? In the supplement it appears as if Klrp1 is expressed somewhat higher in C3,4,5,8.

Input on authors' answer to minor point 33:

The trajectory analysis is still not determining anything about the IL2M treatment- this statement is still just based in the proportional changes between clusters in the IL2M treated mice? If so this should be revised again for accuracy to mention that the trajectory analysis does not inform us about whether IL2M actually directs this bifurcation - just that it could.

Input on authors' answer to minor point 34:

This data should be included in the supplemental figure as displayed previously; all we wanted was for you to include p-values of the multiple comparisons. This is directly relevant and should not be removed.

A normal spleen likely contains 2-4 million Tregs whereas the figure shows maybe a couple hundred thousand (potentially off by a factor of 10). It should just be verified that the numbers are being calculated appropriately.

Input on authors' answer to minor point 35:

Clusters shouldn't be changed by experimental variance, but gene expression could be. The more robust way to ensure that the datasets are reproducible is to show that the same sets of genes

are being differentially regulated in each dataset. This analysis should be performed.

Input on authors' answer to minor point 36:

This would still constitute contamination given that the cells for this experiment were sorted on being CD4+ CD8-. It would be relevant to verify that this cluster does express Foxp3 as well as additional suppressive genes related to CD8 Tregs.

Input on authors' answer to minor point 37:

Yes- but the gene expression patterns for a given cluster could be different based on the ratio of spleen/lung/gut in 1 cluster versus another cluster. These comparisons should be made within a particular tissue to rule out this explanation for differences between clusters.

Input on authors' answer to minor point 38:

This data would probably be better displayed in the form of a table with exact fold changes and p-values.

Input on authors' answer to minor point 39:

Where are the figure legends for the supplemental figures here?

Input on authors' answer to minor point 40:

Where is this data- the only data presented is the previous analysis with 10,000 correlations. If the results are the same then certainly the analysis with equal numbers represents a more powerful finding and should be mentioned in the text.

Input on authors' answer to minor point 41:

OK

Input on authors' answer to minor point 42:

Separate analysis of each condition is still warranted here. Are there TCR's found in both isotype and IL2M treated mice? If so you could use these to bolster claims regarding the shifting of a phenotype following treatment. It is likely these would be quite rare events though.

Final statement:

The authors did well to bolster various aspects of the manuscript including a more thorough description of the IL2M reagent as well as more clear explanations of the methods used for the

single-cell RNAseq and sequencing. However, the report lacks basic mechanistic insight on how IL2 treatments are changing Treg functionality which could be easily discerned from simple experiments. There remain a number of unaddressed concerns regarding the direction of single-cell RNAseq analysis as well as graphical display of repeat data points. These concerns are outlined in the point by point response.

Reviewer #2 (Comments to the Authors (Required)):

The authors had addressed every major and minor point and the revised manuscript is acceptable for publication without any additional review.

RE: 2nd Point-by-point response to Reviewer's comments

General comments by reviewers:

(1) This resubmission of the manuscript by Lu. et al has made improvements that address many of our concerns. Inclusion of a detailed description of the IL2Mutein (IL2M) reagent was very helpful, along with including more details regarding experimental design, repeats and statistics. However, concerns remain on 2 major points. First, the authors state the goal of this manuscript is not to characterize the IL2M or determine its efficacy, but instead to determine the general effect of IL2-based treatments on Tregs. While it can be appreciated that this is perhaps outside the scope of this study, the mechanism of improved suppressor function is still not clearly demonstrated. The two, non-mutually exclusive, likely candidates for driving immune suppression are improved suppressor capacity induced by IL2M ligation, and/or expansion of a particular Treg subset with a higher suppressor capacity. A simple in vitro experiment suppression assay in the presence or absence of IL2M would allow the authors to determine whether this reagent enhances the suppressor function of Tregs. These experiments are relatively easy to do, take little time (< 1 week) and would provide at least some insights into how the IL2M mutein promotes Treg function. We still think this would be a good experiment to do.

Reviewer is asking whether IL2M's effect is due to (1) improved suppressor function induced by IL2M ligation and/or (2) expansion of a particular subset with high suppressive capacity. To answer this question, reviewer suggested to perform in vitro culture of Tregs with IL2M. However, this experiment suggested by reviewer has fundamental problem.

In vitro culture of Tregs requires two signals- IL2 and TCR (CD3/CD28)- and IL2 alone nor TCR alone is not sufficient to maintain Tregs in culture. It is well-demonstrated in below figure we published previously (Choi et al, *Immunology*. 2018 Jun;154(2):309-321. doi: 10.1111/imm.12886). Treating Tregs with IL2 alone in culture substantially reduced CD25 and Foxp3, leaving only 30% of Tregs express CD25 and Foxp3. Because IL-2 mutein signal is even further attenuated compared with wild-type IL2, IL2 mutein signal alone will not maintain Tregs in the culture, thus making the proposed experiment impossible.

Regarding the question whether IL2M's effect is due to (1) improved suppressor function induced by IL2M ligation and/or (2) expansion of particular subset with high suppressive capacity, we demonstrated that IL-2M expanded Tnfrsf9⁺Il1r1⁺ Tregs with superior suppressive capacity in this manuscript.

When we compare the Treg suppressive activity of total Treg population between control or IL-2M treated, we saw the suppressive activity was increased when Tregs from IL-2M treated mice were used (Figure 2A). However, not all populations of IL-2M treated Tregs were superior in suppressive activity. As we demonstrated in Figure 8D, ST2-41BB⁻ population from IL-2M treated mice was not as efficient as the ST2+41BB⁺ population from IL-2M treated mice. Because the ST2-41BB⁻ Tregs also express IL2R $\alpha\beta\gamma$, they respond to IL2M ligation. But these Tregs were not as effective as ST2+41BB⁺ Tregs in suppressing Tconv.

It remains to be answered if IL2M further increases suppressive activity within the particular population expanded. For example, it is not clear whether the ST2+41BB⁺ Tregs from IL2 treated mice are superior to the ST2+41BB⁺ Tregs from isotype-treated mice. This experiment is very difficult to perform because there are very few ST2+41BB⁺ Tregs from isotype-treated mice.

Thus, we will state in the Discussion (pg.23) that the following point and highlighted green.

“The mechanism of increased suppressive activity by IL-2M remains to be elucidated further. We demonstrated that IL2M expands Tnfrsf9⁺Il1r1⁺ Tregs with superior suppressive function. Another non-mutually exclusive possibility is that IL2M induces transcriptional changes improving suppressor capacity within Tnfrsf9⁺Il1r1⁺ Tregs.”

(2) Second, several concerns remain about the analysis of the single cell RNA-seq data sets. The determination of the number of distinct subsets should be validated by some known or speculated biology. Some of the clusters appear very similar to each other and may be considered as falling into the same subset.

We do not believe this request is reasonable. First, this is very vague statement and it is not even clear which validation the reviewer is requesting. The reviewer is proposing to validate the number of distinct subsets based on “known or speculated biology”. For example, it is not clear which speculation is acceptable and which is not; overall, it is a speculation. If we only accept the resolution showing “known biology”, we need to “arbitrary” choose a resolution to only show eTregs and cTregs, which are only two subsets with clear known biology.

To clarify, **cell subset** is not the same as **cell state** in our manuscript. One **Treg subset** (i.e. eTreg) may have multiple **Treg cell states** (i.e. C4, C5). To clarify this, we will include Figure 3B, top panel as a low-resolution clustering to identify known Treg subsets (e.g. cTreg, eTreg) based on existing literature and “known biology”. To further look into more distinct Treg cell states within the Treg subsets, we performed high-resolution clustering which includes the C10 (Figure 3B, bottom panel). Definition of individual state as a distinctive “subset” requires much more validation and out of scope.

To add Figure 3B top panel into the figure, we moved the bottom panels of original Figure 3B into new revised Figure 5A.

Second, this is additional request different from the original request from the first review. Originally, the reviewer requested to provide rationale to choose 10 clusters, which we did. The reviewer has additional question below, and we provide more information to strengthen our argument why we believe the C10 cluster is a real population, not a contaminant. However, requesting “other validation” than providing the rationale to choose the current resolution is beyond the original request.

The rationale from the authors for increasing resolution is based on the appearance of a subset of unknown importance that may even be a sort contaminant.

We still believe that the C10 is not a contaminant population because of two reasons.

First, we did not use CD8 to exclude CD8 positive population for sorting Foxp3-GFP positive Tregs. We used CD4+CD3+ gate from the live gate, then sorted Foxp3-GFP positive population. To clarify this, we now provide expression of Foxp3-GFP, IL2Ra (CD25), IKZF2 (Helios) as three Treg markers between the C10 cluster and Tconv as well as total Tregs. As shown figures below, the C10 as well as total Tregs express Foxp3 much higher compared with Tconv. The level of Foxp3 between the C10 and total Treg was similar. The other two markers-CD25 and Helios also showed the same result.

Second, there are other publications describing CD8+Tregs in mice and in human. The publication by Churlaud et al we referenced in our point-by-point response described this population in mice as well as in human very clearly (Churlaud G, et al. Human and Mouse CD8(+)/CD25(+)/FOXP3(+) Regulatory T Cells at Steady State and during Interleukin-2 Therapy. *Front Immunol.* 2015;6:171.).

Below shows the Figure 1A from this paper. There is a small but distinctive CD8+Foxp3+CD25+ population in mice as well as in human.

- (3) Also, a comparison of transcriptomes between IL2M and control treated mice is a necessary component of this study. If IL2M induces transcriptional changes in a particular subset remains an open question that is easily addressable with the presented data. Based on the tSNE graphs presented, it appears like IL2M treated cells do occupy different regions within the same cluster, and thus likely have transcriptional changes induced by IL2M. Note that this second concern does not involve doing any additional experiments but simply analyzing data the authors already have. Addressing this second concern is essential for proper interpretation of the results presented.

Like we explained in our previous point-by-point response, we first perform clustering of all **combined** data to root the analysis. The clustering therefore defines the differentially expressed genes across populations of Tregs under isotype- AND IL2M treated conditions. We then use these clustering definitions to see how the distribution (not gene expression) of Treg cell states changes across IL2M treatment. Because the clusters are defined by gene expression, if cells from IL2M- or isotype- treated conditions were identified as one cluster, they should express a similar gene set.

Because reviewer requested to demonstrate this by comparison of transcriptomes between IL2M and control treated mice, we are showing examples of this analysis. Below is showing comparison of transcriptomes of the C1 cluster between IL2M treated and Isotype control treated mice. As R^2 value of 0.9878 indicated, the transcriptomes are highly correlated.

As another example, we compared transcriptomes of the C4 cluster between IL2M treated and Isotype control treated mice. Again, R^2 value of 0.9739 indicated, the transcriptomes are highly correlated.

There are multiple questions in the main and major points, such as #35 and #37, where the reviewer wants us to re-analyze the data to look for gene expression differences across batches or tissues within clusters, but this analysis only adds marginal additional knowledge if we use the clustering of cell states to root the analysis.

First, we suspect that these requests are in part from misunderstandings regarding the nature of single cell transcriptomic data versus bulk RNA-seq data. Since the unit of interrogation in single cell transcriptomic data is the *cell* and not the *sample*, the cell assignments after clustering provide the most comprehensive characterization of the dataset, and subsequent analyses of tissue and treatment effects should primarily utilize the cell frequencies generated by the clustering assignments over differential expression lists. In bulk RNA-seq, the major unit of interrogation is the sample, in which expression is an aggregate of all the cells present. Therefore, gene expression is the best possible proxy to measure sample composition, since there is no way to obtain granular, single-cell information from the data.

Second, our analysis is focused at the clustering level of the *cell state*, which provides a higher resolution level than clustering performed using the *cell type* level by revealing more subtle differences in expression levels between

cells. This high-resolution level of analysis explains why there is very similar expression between iso- and IL2m-treated cell states and why this type of analysis requested by the reviewer is unlikely to be informative in this case. Examples where this analysis would be more informative involve comparison across cell types (see stimulated vs control PBMCs example: https://satijalab.org/seurat/v3.1/immune_alignment.html), where the gene expression differences in the cell types exist because it is not accounted for by a higher level of resolution.

These are the analyses reviewers asked and we do not believe it adds meaningful additional knowledge.

- IL2M vs isotype DE analysis within the same cluster, since the highly variant genes are already defined by clustering algorithm >> we showed examples of similarity between iso- and IL2m-treated cells in the C1 and C4 clusters above. While such analyses are interesting, we believe they unnecessarily complicate the manuscript without strengthening the main points.
- Tissue-specific Tregs DE analysis within a defined cluster, since this is redundant (genes are defined by clustering algorithm (see the point above))
- Batch effect DE analysis within the same cluster, since this is redundant (genes are defined by clustering algorithm) >> gene expression is used to determine cluster assignment, so if the gene expression is highly dependent on batch effect instead of true biological variability, the clusters will be distributed unevenly between samples from the same treatment and tissue. Since we see the cluster distribution evenly divided between the 3 replicates in Supp. Fig. 4, this provides a better test for batch effect than a DE analysis for each batch.
- Repeated experiments for TCR sequencing so statistics can be drawn on these measures: While this is an interesting avenue for exploration, this is beyond the scope of paper and it requires more than 6 months of additional single-cell experiments, bulk paired TCR experiments (which yield more sequences and are the preferred method for repertoire analyses) and computational analyses.
- Pseudotime of IL2M vs isotype cells, since the purpose of pseudotime was to look at the development of Treg cell states in relationship to each other. The pseudotime distribution of IL2M vs isotype cells is already defined by the frequency of major cell states in each treatment and their respective pseudotime locations.

RE: Point-by-point response to Reviewer's comments

General comments by reviewers: Provide further information about the murine IL2 mutein used in this study.

IL2-mutein is a half-life-extended mouse IgG2a Fc fusion protein of a mutant form of mouse IL2, in which mouse IL2 is engineered to improve selective binding towards Tregs. In short, the mouse IL-2 mutein we generated is similar to the human form of long-lived IL2 (human IgG-(human IL-2N88D)₂, which was reported by Peterson et al (1). In this report, authors used an effector-silent human IgG1 to increase half-life and also the N88D mutation in human IL2, which decreased binding to IL2R β and allowed selective binding of the human IL2 mutein to the high affinity IL2R $\alpha\beta\gamma$ on Tregs.

Recombinant wild-type IL2 has very short half-life. The serum half-life of human IL2 in man after i.v. administration is notoriously short, with value of 6.9 min for recombinant human IL-2(2). Clearance of recombinant human IL-2 was even faster in mice, with a serum half-life of about 1.6 min when administered i.v (3). Frequent administration of large amount of recombinant IL2 has been used to maintain therapeutic serum levels, but capillary leak syndrome was emerged as one of the major side effects of this frequent high dose IL-2 therapy (4).

To extend the half-life of murine IL-2, we fused the Fc portion of mouse IgG2a with a linker, which is similar to the effector-silent form of human IgG1 used in human IL2 mutein. The N297G mutation was introduced in muIgG2a to inhibit ADCC (antibody-dependent cell cytotoxicity) activity of the mouse IgG2a Fc in order to generate an "effector-silent" version (5). The last amino acid residue of mouse IgG2a, K, is usually cleaved off by carboxypeptidase activity in monoclonal antibody. To maintain

homogeneity, the last K residue of the mouse IgG2a was deleted (**Data Figure 1**).

When we administered murine IL-2M, it showed dose proportional exposure increase. Serum concentration of IL-2M reached its maximum concentration at 6 hours after administration, then gradually decreased over 7 days. During this time, serum concentration of IL-2M was maintained over 0.1 nM. After 4 days of IL-2M administration, the serum concentration of IL-2M was between 0.1 -1 nM, which was concentration of IL-2M expanding Tregs selectively over Tconv *in vitro* experiment. Furthermore, we measured expansion of Tregs at Day 4 and 7 and found that expansion of Tregs by IL-2M was maximum after 4 days following administration.

In order to achieve selectivity towards Tregs, we sought for mutations that attenuate interaction between IL2R β (CD122) and murine IL2, similar to the N88D mutation in human half-life extended IL2 mutein(1). It is because the interaction of IL2 with IL2R β in the intermediate IL2R β γ receptor in conventional T cells expands Tconv cells following IL-2 treatment.

The IL-2R exists in two functional forms. The high affinity IL-2R assembles when IL-2 is captured by the IL2R α (CD25) subunit that in turns facilitates additional binding to signaling receptors- IL-2R β (CD122) and IL-2R γ (CD132)-, forming IL-2R α β γ . The high affinity IL-2R α β γ is found in Tregs (CD25+Tregs) but also found in lower levels on effector T conventional cells (CD25+ Tconv). The intermediate affinity IL-2R (IL2R β γ) is expressed by multiple hematopoietic lineage cells, including conventional T cells as well as NK cells(6). When high dose of IL2 is used, it interacts with the high affinity IL-2 receptor (IL-2R α β γ) as well as the intermediate affinity IL2 receptor (IL2R β γ) and activates both Tconv and Tregs. Only when IL2 is used in low-dose, selectivity towards Treg can be achieved. However, low-dose IL2 therapy cannot achieve greater expansion of Tregs due to the limitation of the amount can be administered because the window of the dose to achieve Treg selectivity over Tconv is very narrow.

Zurawski et al reported series of papers describing important residues of murine IL2 for interacting with IL2R α and IL2R β (7,8). Among those residues, the D34 and N103 residues of mouse IL2 were shown to be important for IL2R β (CD122) binding. The N103 residue of mouse IL2 mutein is corresponding to the N88 of human IL2(1). Thus, we mutated D34 and N103 residues of murine IL2 to D34S and N103D to reduce the interaction between IL2 and IL2R β .

In addition to D34S and N103D mutations in mouse IL2, we incorporated two additional mutations (C140A and P51T) of mouse IL2 to facilitate manufacturability. The C140A is a mutation corresponding to C125A in human IL2 to avoid aggregation. This mutation was also incorporated in aldesleukin (low-dose IL2) and was reported previously (9). The P51T is a mutation specific for the murine IL2 and it was used to prevent clipping during production.

Great. This section is very helpful in understanding what the IL2M is and the logic behind its generation.

Intracellular staining of phosphorylated STAT5 was performed to determine the activity of mouse IL-2M *in vitro*. Because IL-2M was mutated to decrease its binding to IL2R β , the activity of IL-2M was attenuated compared with wild-type recombinant IL2. However, IL-2M induced phospho-STAT5 in CD25+Foxp3+ Tregs that express the high affinity IL2R α β γ in a dose dependent manner, while it did not induce phospho-STAT5 in CD25-Foxp3- Tconv cells that express the intermediate IL2R, IL2R β γ (**Data Figure 2**). In contrast, wild-type recombinant IL2 induced phospho-STAT5 in CD25-Foxp3- Tconv as well as Tregs.

This new data is also very helpful.

Because the mutations we generated were intended to reduce the interaction of mouse IL2 with IL2R β but maintain interaction with IL2R α (CD25) to allow binding to the high affinity IL2R (ILR α β γ) on Tregs, IL-2M slightly activated CD25+Foxp3- Tconv cells, although the degree of activation was markedly reduced compared with wild-type rIL2 (**Data Figure 3**).

We hope this information will answer reviewer's questions about murine IL-2M.

We now added a paragraph describing mouse IL-2M (manuscript pg. 5) and it is highlighted in yellow. In addition, we added Data Figures 1-3 as the revised Supplemental Figure 1A-B.

From here, we would like to respond with detailed response to each reviewer's comments.

Reviewer #1 (Comments to the Authors (Required)):

Summary:

This manuscript by Lu, D. R. et al describes the effect of IL-2 mutein treatment on murine Tregs. First the authors compare the IL-2 mutein to currently available Treg selective IL-2 treatment modalities (IL-2 complex) and suggest that the IL-2 mutein is more specific for Treg expansion than IL-2 complex treatment, which also causes modest expansion of effector T cells. Using single cell RNA seq analysis, the authors then explore the alterations in Treg phenotype and subset distribution in IL-2 mutein treated mice versus controls. This reveals the expansion of ST2+41BB+ Tregs during IL-2M treatment. In vitro analysis of Treg subset suppressor function demonstrates that ST2+41BB+ Tregs are imbued with increased suppressor activity. The major feature of this manuscript is tracking expansion of Treg subsets during an IL-2M based treatment regimen. Analysis of single cell sequencing from Tregs doesn't reveal novel biology over previous reports however this study is able to identify the effect of IL-2M treatment on Treg subset proportions and provides a rationale for exploring IL-2M treatment clinically. Overall, the general conclusion that an IL2 mutein allows for selective expansion of specific Treg subsets is supported by the data. However, there are a number of concerns about specific studies and clarification is needed for some of the analysis. **"Finally, there is no indication that the single cell RNA-Seq data will be deposited with a public repository (GEO, etc.)."** Specific concerns are outlined below.

Response: We will deposit the single cell RNA-seq data into the EMBL-EBI public repository, European Nucleotide Archive (<https://www.ebi.ac.uk/ena>). We will confirm submission of the data upon acceptance of the manuscript.

Major Points:

1. "The determination of clustering resolutions seems arbitrary. The heatmaps show somewhat undefined transcriptomic signatures between some clusters thus having 10 clusters may represent excessive resolution versus biology. There should be a comparison of different resolutions to show that the chosen resolution accurately represents biological heterogeneity."

We acknowledge that the determination of optimal clustering resolutions is difficult in single-cell RNAseq and must balance a cautious approach that minimizes technical, non-biological variances with an approach that can reveal novel cell populations of biological interest. For this reason, we (1) analyzed the data iteratively at different clustering resolutions and (2) used quantitative measures (thresholding p-values and log-fold changes in expression) to identify critical genes.

The optimal clustering resolution in Seurat was determined by clustering integrated single-cell expression data at ten different resolutions from 0.1 to 1.0 using the "resolution" parameter in the FindClusters() function. At each resolution, the top marker genes of the cluster containing the fewest cells were evaluated against previously published literature to support inclusion. The determination of 0.6 for the resolution parameter was made because this was the lowest resolution (i.e. smallest cluster number) at which C10 IFN γ ^{hi} Tregs - which express ex-Treg markers and were previously described by Daniel V *et al.* (10) and others- could be identified (**Data Figure 4**). Lower resolutions merged this population with other cell states, masking the distinct gene expression profile of this population (see Fig. 3C, Sup Fig. 5C, Sup. Fig. 9C, and **Data Figure 5**).

The rationale for setting the resolution appears to be that it allows one to detect a potentially novel population of CD8+ Tregs. However, it is unclear whether this particular population relevant to the study at hand as the authors sorted on CD4+CD8- cells, thus the CD8+ population must have arisen via an imperfect sort. It is also unclear whether these cells actually express Foxp3. The described study

(10) doesn't identify these cells as ex-Tregs and is done in human PBMCs so to compare this study to that one is a stretch. It seems most likely that this C10 populations arises via contamination from CD4 or CD8 effectors- this is quite common in single cell data sets and the cluster in question is very small.

We addressed this concern in the main point (2) in page 2-3.

Despite the appearance of “undefined transcriptomic signatures between some clusters” in some parts of the heatmap, we believe our cluster definitions represent an appropriate resolution to reveal true biology. First, marker genes for clusters were compared against existing scientific literature for their role in Treg function prior to inclusion. Second, marker genes were quantitatively selected based on statistical significance after differential expression using MAST, as well as the application of a log-fold cut-off to further triage marker genes. Third, the Louvain algorithm identifies clusters based on similarity networks, which are defined by combinations of genes and not by single genes. Thus, while the heatmap may not adequately represent these differences, the differences between clusters can be captured by complementary analyses. For example, while clusters C1 and C2 appear to overlap in a large number of gene signatures (Sup. Fig. 5A), the trajectory analysis clearly places C1 and C2 cells and assigns distinct pseudotime values to these clusters, highlighting their distinction (Sup. Fig. 12B). Based on these multiple lines of evidence, we believe that our clustering resolution best represents the current knowledge of Treg biology and the biological variance present in the dataset.

A heatmap should be added displaying the top 10 differentially regulated genes in each cluster compared to every other cluster. Many of the “marker” genes correlate to no known Treg subset or function (i.e. *Ass1*, *Stmn1*, *Dnajb1*). The resolution chosen is going to determine if 2 sets of cells are assigned to different clusters by the algorithm. However, this is implicit and is why they are put into 2 distinct clusters, so it doesn't have any bearing on whether this represents a truly biologically distinct population (circular argument). The pseudotime interpretation could go the other way as well- if a cluster of cells doesn't occupy any particular niche over pseudotime (i.e. clusters 2, 3, 7, 9 and 10) then could you argue that these don't represent any unique population? I don't think you can make this conclusion so the statement that having different pseudotime distributions (would need to be statically validated anyway) has no bearing on whether 2 clusters are distinct or not. A more concrete description of the biology represented by the subsets should be provided or else another resolution should be chosen for the analysis.

Regarding the heatmap: We already provided heatmap displaying top 3-5 differentially regulated marker genes from each cluster in Figure 3C. These marker genes from each cluster were selected by their differential expression compared with all other Tregs. Moreover, we expanded these genes and showed 5-7 differentially regulated genes from each cluster in Supplemental Figure S6A.

Regarding the marker genes such as *ASS1*, *Stmn1* and *Dnajb1*, there are reports describing their roles in Tregs.

ASS1 is a cytosolic enzyme that plays a role in the conversion of citrulline to arginine. Arginine metabolism in Tregs is reported. Unlike T effector cells requires arginine, Tregs can synthesize arginine from citrulline. Elevating L-arginine levels induce global metabolic changes including a shift from glycolysis to oxidative phosphorylation (Geiger et al, *Cell*. 2016 Oct 20; 167(3): 829–842.e13.), which is the main metabolism pathway of Tregs (Howie, *JCI Insight*. 2017 Feb 9; 2(3): e89160). Thus, elevation of *ASS1* in Tregs may convert citrulline to arginine, thus maintaining their oxidative phosphorylation metabolism.

Stmn1, Stathmin 1, is cytoskeletal proteins that regulates microtubule dynamics and is involved in mitosis and cell cycle of activated T cells (Silva et al., *Mol Biol Cell*. 2013 Dec;24(24):3819-31. doi: 10.1091/mbc.E13-02-0108. Epub 2013 Oct 23). Because the C6 cluster is the proliferative cluster, it is not surprising *Stmn1* is highly expressed in the C6 cluster.

Dnajb1 encodes a member of the DnaJ or Hsp40 (heat shock protein 40 kD) family of proteins. The encoded protein is a molecular chaperone that stimulates the ATPase activity of Hsp70 heat-shock proteins in order to promote protein folding and prevent misfolded protein aggregation. HSP70 is known to form a biochemical complex with Foxp3, and its upregulation can significantly enhance Treg survival and suppressive function *in vitro* and *in vivo* (de Zoeten *et al.*, Gastroenterology, 2010 Feb;138(2):583-94. doi: 10.1053/j.gastro.2009.10.037. Epub 2009 Oct 29).

2. “The data suggests that IL-2 mutein treatment leads to a shift between cTreg and eTreg phenotype; this largely disagrees with the bulk of published reports that say IL-2 is less important for eTregs, at least for their maintenance (For example, see work by D. Campbell and colleagues). How does this dataset reconcile these observations?”

We did not conclude that IL-2 is more important for eTregs. What we reported was that murine IL-2 mutein (IL-2M) treatment increased the proportion of C4- and C8 -Tregs (which mostly resemble eTregs), while reducing the proportion of the C1-Treg (which resemble cTregs). Because we are looking at Day 4 after IL-2M treatment, this is the result of the action of IL-2M on Tregs. Thus, just by looking at the result of the action of IL-2M, it is difficult to determine which population of Tregs is responsible for this result.

Data from Campbell’s lab showed requirement of IL-2 in cTregs by using IL-2Ralpha KO mice as well as blocking IL-2 (11). In the same paper, authors also showed that, when cTregs were transferred to the mice activated by TCR and LPS, cTregs were activated and became eTregs, indicating that cTregs responded to stimuli and changed their state to eTregs.

This is confusing because IL2 treatment still doesn’t cause conversion of a cTreg into a eTreg- this requires activation of the cTregs. Why are the cTregs being activated during IL2M treatment? Why would they now be receiving TCR stimulation? Alternatively, please include a statement that it is also possible that IL2M preferentially expands distinct Treg subsets.

I included the statement the following statement in page 19 and highlighted in green.

“Of note, because expansion of Tregs in the C2/C4/C8 clusters following IL-2M treatment is the result of action of IL-2M, we cannot distinguish which cluster of Tregs responded to the IL-2M to give rise the expansion of the C2/C4/C8 clusters. Likewise, it is possible there is a preferential increase of a specific cluster to respond to IL-2M, resulting in the expansion of the C2/C4/C8 clusters.”

Based on Monocle analysis, the C1 Tregs (naïve cTregs -like phenotype) occupied the earliest period of pseudotime. The C2 Tregs (primed and activated state) occupied the intermediate state and the C4 and C8 Tregs (eTregs-like phenotype) occupied the terminal state. Because IL-2M treatment increased the proportion of C4 and C8-Tregs while reducing C1-Treg, it is likely that C1-Treg responded to the IL-2M, then differentiated into the C4/C8 state. If this is case, our data is consistent with data from Campbell’s group, in which cTregs changed their state to eTregs following stimulation.

This is still not consistent with the literature on the effect of IL2 on Treg phenotype. Once again, could it be equally likely that the IL2M is just expanding these preexisting eTregs? This again, seems less likely given the requirements of eTregs. A better mechanistic understanding of this process should be provided to explain the results, rather than speculation based on the single-cell RNAseq datasets. Alternatively, a statement acknowledging that these differences could arise via preferential expansion of existing Treg subsets could be included.

See above statement in green highlight.

3. “Experimental repeats are lacking in a number of experiments (Figure 8 B-C, Supplemental Figure 1)

and statistical analysis is lacking in many figures (Figures 3D, 5B, 6, 7B, 8B, Supplemental 3C). The number of separate experiments performed and total n for many figures is not complete.”

We performed the following statistical analysis and revised the text accordingly.

7

- 1) Figure 3D: The Wilcoxon rank sum test was applied to Figure 3D to test for significant expression changes.
- 2) Figure 5B: Quantification of Figure 5B is shown in Figure 5C. Statistical analysis using Welch's ttest was included in Figure 5C to test for significance in Treg cell states across treatment groups for all three tissues.
- 3) Figure 6: Fisher's exact test was applied to Figure 6 to test for independence between clonal family frequency and IL-2 mutein treatment. For example, if the number of clonal families we observed in Tregs is "dependent" upon IL-2 mutein treatment, p-value will be <0.05. If the two variables (# clonal families and IL2M treatment) are "independent", p-value will be > 0.05.
- 4) Figure 7B: Fisher's exact test was applied to Figure 7B to test for independence between the frequency of clonal cell state pairs and IL-2 mutein treatment.
- 5) Figure 8B: We added Figure 8C to show statistical analysis with multiple replicates.
- 6) Figure 8B-E: We added the following statement in the Figure Legend and highlighted: "Results are representative of two independent experiments, using 2-3 mice in each experiment."
- 7) Supplemental Figure 1B; Data are representative of two independent experiment.
- 8) Supplemental Figure 1D; Results are representative of at least two independent experiment using 3 mice per each group.
- 9) Supplemental Figure 1E-F; Results shown are from three mice from each condition from one experiment.
- 10) Supplemental Figure 3C: Differential expression was performed using MAST, and the results of calculated p-values were added to Supplemental Figure 3C.

OK now.

4. "Given that the main focus of this manuscript is on generating a more functional ST2+41BB+ Treg subset following IL-2M treatment, other IL-2 based therapies should be analyzed for the expansion of this population (as in figure 8B). This will provide context into the effectiveness of the treatment versus alternatives. It should also be pretty straight forward to do."

The main focus of the manuscript is to describe the effect of IL-2M on heterogenous populations of Tregs and to determine how it alters the landscape of Tregs, but not to compare the effectiveness of the IL-2M over other IL-2 based therapies. In fact, it is difficult to compare the effectiveness of different modalities without information of pharmacokinetics or pharmacodynamics, thus it is beyond the scope of current paper.

For example, recombinant mouse IL-2 has very short half-life *in vivo* as previously described. In addition, we showed that high dose of recombinant mouse IL-2 did expand Tconv as well as Tregs *in vitro*. Thus, at higher concentration, IL-2 will expand Tregs as well as Tconv. Likewise, comparing effectiveness of the IL-2/anti-IL2 antibody (IL-2C) and IL-2M will not be meaningful without determining pharmacokinetics or pharmacodynamics of the two agents. For example, for IL-2/anti-IL2 (IL-2C) administration *in vivo*, we needed to administer the IL-2C daily for three days as previously reported(12). If we increase the dose or frequency of either IL-2M or IL-2C, it will further increase Tregs.

Having said that, we think that it will be meaningful to show whether the response induced by IL-2M is also induced by IL-2C. We determined the percent of ST2+ cells within Tregs from spleens followed by IL-2M or IL-2C *in vivo* treatment because ST2 was increased in both C4- and C8-clusters following IL-2M treatment. Similar to IL-2M, IL-2C also increased the percent of ST2+ Tregs (**Data Figure 6**).

OK

5. “In vitro experiments should be performed to evaluate the effectiveness of IL-2M in stimulating non-Treg T cells and compared to other IL-2 treatment modalities. This should give definitive evidence that the IL-2M has superior on target (Treg) activity compared to other IL-2 therapies.”

We agree that performing *in vitro* experiment comparing IL-2M and murine recombinant WT IL2 (rmIL-2) for their activity on Treg vs. non-Tregs population would clarify the selective effect of IL-2M on Tregs. Thus, we determined phosphorylated STAT5 using intracellular FACS analysis from CD25⁺Foxp3⁺ Tregs and CD25⁺Foxp3⁻ Tconv cells following increasing dose of recombinant mouse wild-type recombinant mouse IL-2 (rmIL-2) or mouse IL-2M treatment (**Data Figure 2 and 3**). Wild-type rmIL-2 resulted in phosphorylation of STAT5 in both Tregs and Tconv at high dose. In contrast, IL-2M resulted in phosphorylation STAT5 only in Tregs but not in Tconv.

This is a nice result.

6. “There is no analysis of transcriptomic differences between clusters in isotype and IL-2M treated Tregs. It would be relevant to compare, for example, isotype treated cluster X and IL-2M treated cluster X to see what differences are being driven by the treatment.”

For the cluster analysis, we integrated all single-cell RNA-seq data from isotype control-treated and IL-2M- treated mice prior to cell clustering, and this approach was driven by our biological understanding of the effect of IL-2/IL-2M signaling on Tregs. IL-2M treatment is NOT creating a new Treg cell state, but it results in over-representation or under-representation of existing states among the Treg differentiation continuum. Therefore, we applied clustering to the combined isotype- and IL-2M-treated cells to properly observe changes in the representation of cell states. This led to such findings as the decrease in C1-Tregs and increase in C4/C8-Tregs following IL-2M treatment.

In our analysis, each cluster is defined by transcriptional differences when all samples are aggregated. Thus, comparing isotype-treated cluster X and IL-2M treated cluster X will not generate meaningful gene sets within that cluster X, because cluster X is already defined by expression of unique gene sets from both isotype- and IL-2M-treated cells.

Unless you have done a comparison between unstimulated and IL2M treated Treg subsets then you don't know if IL2M does not alter the transcriptome within subsets. There is precedent that Tregs with stronger STAT5 activation are better suppressors so this treatment certainly could be altering the transcriptome of Tregs of similar overall phenotypes. In fact, this is clearly visible in the tSNE plots when control and IL2M are being compared- IL2M treated cells within the same cluster appear in a distinct space from control cells in the same cluster. This analysis should be included in this manuscript.

We addressed this point in the main point (3) in page 3-4.

7. “ In figure 2A, PBS treated Tregs do not appear to suppress proliferation - only IL-2Mutein treated cells suppress proliferation. Why do control Tregs not suppress as expected?”

This question is due to confusion of the labeling of Figure 2A. The first panel of Figure 2A showed that Tregs from PBS-treated mice were added into naïve T cells in *in vitro* Treg suppression assay. Second panel showed that Tregs from IL-2M treated mice were added into naïve T cells. The third panel showed that proliferation of naïve T cells without any Tregs. **Compared with the third panel, Tregs from PBS treated mice (from the first panel) did suppress proliferation of naïve T cells, but not as much as Tregs from IL-2M treated mice (second panel).** The experimental procedure was described in the Figure legend, and we revised labeling in the revised Figure 2 to clarify this.

I agree that the panels in figure 2A make it look like the untreated control Tregs are suppressing proliferation of naïve T cells compared to no Tregs. However, the same problem remains in figure 2B, which seeks to quantify that difference - the CTV gMFI is still unchanged between the PBS and No Treg groups so this again suggests that the untreated Tregs are not suppressing proliferation. It might be

better to compare % divided cells as the authors have done in other figures as that might give a more obvious difference. Otherwise, you are left with a representative example in figure 2A that looks convincing, but aggregate data (figure 2B) which shows no difference. The % IFN γ is mildly lower in PBS vs No Treg, which at least shows they are doing something. % Divided should be calculated here as well, not just CTV gMFI.

This is a good point, and we showed % divided cells as well in Figure 2A, right panel and revised the Figure legend (shown in green) in pg 38.

Minor Points:

1. "What is the IL-2 mutein? The description provided is relatively vague mentioning only an IL2-Fc fusion. Isotype and pbs are both mentioned as controls for IL-2M- this should be standardized".

We provided information from Pg 1-3.

This was a very helpful addition to the paper.

2. "Some cell states are described by speculating on the biology but not backed up by primary experiments in this or other manuscripts- for example, is C7 representative of any known Treg biology? This relates in part to a better explanation of how a resolution yielding 10 subsets was chosen".

We provided answers for the resolution from the Major point #1, Pg 4-6.

Because gut Tregs contain primarily the C7-Tregs and the C7-Tregs express early response inflammatory genes, we speculate that these Treg cells may provide tolerance towards gut microbiome or food antigens.

This statement should be quantified as they are still a minor component of gut Tregs as presented in figures 4A and 5A.

In average, the C7 cluster Tregs comprises 5.8% of gut Tregs from isotype-treated mice, 2.2% in spleen and 1.4% in lung.

3. "Figure 1B-(CD25+Foxp3-): This should be displayed on a graph from 0-1.0%, 0-40% is impossible to interpret. The inset is too small to see. Likewise, the resolution of figure 3B, 4A, and supplemental figures 10 and 12 is poor - labels of cell subsets are impossible to read in 3 & 4 and gene names cannot be read in Sup figs 10 and 12".

The low resolution of Figures is due to embedded PNG files in the Word document. To address this concern, we provide the following files.

Figure 1B: We changed the y axis of the graph to 0-1.0% and removed the insert.

As such- change the y-axis for the CD25-Foxp3+ graph as well to maybe 10-20% and the y-axis for CD25-Foxp3- should go down to 0.

Changed.

Figure 3B and 4A: We provided Tiff files with increased resolution (600 dpi).

Supplemental Figure 10 and 12: We enlarged the figures and provide Tiff files with increased resolution (600 dpi).

4. "Figure 1C: There is a significant difference between effector cells with IL-2 mutein treatment. Perhaps a quantification of the fold expansion difference between Treg and Tconv would be helpful to visualize the relative effects on these 2 cell types".

To address this concern, we now provide the fold increase in the revised Supplemental Figure 1C.

Although there was significant difference between CD25+Foxp3- effector T conv cells, the fold increase

of cell numbers was approximately 1.2 fold in lymph node and spleen and 0.7 fold in the lung while the fold increase of cell numbers of CD25⁺Foxp3⁺ Tregs were approximately 5 fold in lymph node and spleen and 7.5 fold in the lung following IL-2M treatment.

OK

5. "Is there a rationale behind why CD25⁻ Tregs are also expanding- are these derived from differentiated CD25⁺ Tregs?"

We also wondered about this. Previously it has been reported that CD25^{low} Foxp3⁺ T cells share phenotypic features resembling conventional CD25^{high} Foxp3⁺ Tregs in human (13). This report concluded that the number of CD25^{low}Foxp3⁺ T cells was correlated with the proportion of CD25^{high}Foxp3⁺ T cells in cell cycle, suggesting that CD25^{low}Foxp3⁺ T cells represent a subset of Tregs that are derived from CD25^{high}Foxp3⁺ T cells. Because IL-2M increased the CD25^{high}Foxp3⁺ T cells in cell cycle, we speculate that the increase in CD25^{low}Foxp3⁺ T cells may be derived from CD25^{high}Foxp3⁺ T cells.

OK

6. "Expansion of CD25⁺Foxp3⁺ Tregs by IL-2M was comparable to IL-2/anti-IL2 antibody"- Supplemental 1C only quantifies Foxp3⁺ Tregs, not CD25⁺Foxp3⁺ Tregs".

We revised the text to Foxp3⁺Tregs and highlighted.

OK

7. "After filtering and cross-sample normalization using Seurat(16), we recovered 17,097 spleen Tregs, 10,353 lung Tregs, and 4,458 gut Tregs across three replicates with roughly equivalent Tregs"- In supplemental Fig. 2C there are only 2 replicates for the gut Treg dataset however the text makes it appear as if there are 3 data sets for each organ. This should be corrected".

We revised it and highlighted:

"After filtering and cross-sample normalization using Seurat(16), we recovered 17,097 spleen Tregs, 10,353 lung Tregs, and 4,458 gut Tregs across three replicates (except gut Tregs treated with IL-2M (n=2)) with roughly equivalent Tregs in mouse IgG Fc isotype control (Iso)- and IL-2M-treated conditions (16,152 and 15,756 cells, respectively)."

OK

8. "Additionally, Tregs expressed higher transcript levels of established Treg genes such as Foxp3, Il2ra, Ctla4, Ikzf2, and Nrp1, while both cell types expressed similar levels of Cd4 (Supplemental Figure 3, B and C)."- Have stats been performed to confirm that these are robust differences?"

11

Yes. This question is similar to Major point #3 from Reviewer 1 (pg. 7), and the concern was addressed there. Differential expression was performed using MAST, and the results of calculated p-values were added to Supplemental Figure 3C.

OK

9. "Figure 2: Could these experiments be performed with IL-2 mutein in vitro instead of in vivo? This could perhaps delineate between subset differences and direct effects of the mutein on Treg functionality".

We believe that using Tregs from mice treated with IL-2M *in vivo* is better representation of the effect of IL-2M *in vivo*.

It obviously is a better representation of the effect of IL2M in vivo but does not address mechanism effectively. The in vitro experiments would allow you to determine if one part of the mechanism involves enhanced suppressor function on a per cell basis (see first main concern in initial paragraph).

We provide answers for this from the main point (1) in page 1-2.

10. "Resting Tregs (C1) have high expression of lymphoid-tissue homing receptors (Ccr7, S1pr1, Sell)"- The data shown suggest this is perhaps true for CCR7 but less so for S1PR1 and not true for Sell

Thanks for noticing this error. This statement is true for S1PR1, but not Sell. S1PR1 was significantly upregulated in C1. However, Sell was significantly upregulated only in C2 and C6, but not in C1 cluster (Data Figure 7). Thus, we revised the text to "Resting Tregs (C1) have high expression of lymphoid tissue homing receptors (Ccr7 and S1pr1)".

OK

11. Figure 3: It seems surprising that CD62L is more highly expressed in activated cells versus resting Tregs- how does this reconcile with known phenotypes of eTreg vs cTreg?

Expression of CD62L(Sell) is significantly high in C2 (primed/activated cluster) and C6 (proliferative) clusters, but it was markedly reduced in the spleen-enriched C3 (activated) and gut-enriched C5 clusters. Thus, downregulation of Sell in activated Tregs (eTregs) holds true at least in C3 and C5 activated Tregs. For the highly proliferative C6-Tregs, active cell cycling of these cells may cause more Sell to be expressed.

Downregulation of Sell on activated eTregs occurs transcriptionally as well as by protein shedding. It is possible that the protein expression of Sell in the C6 cluster may be decreased due to shedding despite higher transcriptional expression.

OK

12. "Furthermore, C2- and C9-Tregs could be distinguished from each other, as C2-Tregs express more Nrp1"- This comparison is not being statistically evaluated in the supplemental figure. Direct comparisons between groups should be made when statements are calling 2 groups different."

We performed differential expression using MAST of C2 versus C9. After differential expression analysis of Nrp1 in C2 versus C9, we find that C2 is significantly upregulated relative to C9 (log₂expression=0.33, FDR-adjusted p-value=0.0119). This log-fold change in expression is shown in Supplemental Figure 7B already. We have reported this p-value in Sup. Figure 7A.

OK

13. "Figure 4: Transcriptomic information should be compared between the same cluster in different organs as well as clusters within the same organ- not just done on a bulk basis. Similarly, transcriptomes should be compared between the same clusters in control and then separately for IL-2 mutein treated Tregs."

Discussed in Major point #6. The purpose of this figure is to show global differences between tissues. We already showed differences in transcriptomes in individual clusters.

OK

14. "At the tissue level, the spleen and lung share a >40% frequency of C1-resting and minor frequencies of primed/activated and activated Tregs. Conversely, >80% of gut Tregs are activated and the majority are C5-Tregs (Figure 4, A and B)." This data should be quantified and analyzed statistically."

This paragraph is supposed to depict Figure 5 not Figure 4. Quantification is provided in Figure 5B. We

changed the text accordingly.

OK

15. "The coexpression of immunomodulatory genes (Cst7...."- The authors state that Cst7 is an immunomodulatory gene. However, Cst7 is a known gene downstream of TCR stimulation (Fassett MS, Jiang W, D'Alise AM, Mathis D, Benoist C. Nuclear receptor Nr4a1 modulates both regulatory T cell (Treg) differentiation and clonal deletion. Proc Natl Acad Sci USA. 2012;109(10):3891-3896. doi: 10.1073/pnas.1200090109.) as well as a secreted factor from Tregs (Paracrine effect of regulatory T cells promotes cardiomyocyte proliferation during pregnancy and after myocardial infarction. Nature Communications 2018). Given these publications it may be more appropriate to include Cst7 in both the immunomodulatory and activation gene sets in this sentence.

We removed Cst7 in the text following reviewer's suggestion.

OK

16. "Treatment with IL-2M shifted the frequency of Treg clusters, reducing C1-resting and C3-activated Tregs while elevating proliferation (C6),"- In figure 5, C6 is not statistically different between control and IL-2M.

In Figure 5, C6 is not statistically different between control and IL-2M despite there being an increase in the percentage of C6 cells after IL-2M treatment. This result is shown in the representation of P-value results in Figure 5C. The statement was revised to "Treatment with IL-2M shifted the frequency of Treg clusters, reducing C1-resting and C3-activated Tregs while increasing primed/activated (C2) and activated Treg states (C4 and C8) (Figure 5B-C)."

OK

17. "The identification of transcriptional diversity among Tregs from the same clonal family was an interesting result, since we also find that pairs of T cells belonging to the same clonotype tend to be transcriptionally correlated than randomly sampled pairs of Tregs at the population level, although this was a modest effect(20)"- It should be pointed out that this result is recapitulating previous results published in ref-20.

We revised the text following reviewer's suggestion and added "as previously reported".

OK

18. Figure 6: This figure would be improved by quantitative analysis. How do we know these are really different rather than differences in TCR coverage perhaps?

This comment was already addressed in Reviewer #1 Major point #3. Regarding the reviewer's comment on the distinction of true differences in clonotype sharing from differences in TCR sampling, Fisher's exact test, which we use to test for independence between clonotype sharing and treatment, accounts for sampling differences by using the hypergeometric distribution to calculate expected frequencies. Therefore, we conclude that the differences in clonotype sharing in Tregs treated by IL2-mutectin are statistically meaningful.

There is still no quantitative analysis in this figure. What does total clonotype sharing refer to- this should be analyzed by clonality or diversity within a sample. It would be ideal to have repeated experiments for TCR sequencing so statistics can be drawn on these measures.

"Clonotype sharing" (a.k.a. expanded clonotypes) refers to the number of Tregs that were identified in a clonotype consisting of two or more cells. Given the available sequencing data we generated for 1 isotype-

treated and 1 IL2m-treated mouse, the Fisher's exact test shows statistical evidence for a meaningful relationship between treatment and the frequency of Tregs involved in an expanded clonotype.

While we agree that it would be ideal to repeat this single-cell experiment, it would require more than 6 months of additional experimental work and analysis to investigate a claim that is beyond the scope of this paper. Furthermore, we do not believe that repeating the TCR sequencing will answer the reviewer's original question beyond what our existing analyses show.

19. Figure 7A-B: This data should be statistically compared between control and IL-2 mutein treated mice. Additionally, it is unclear if this data has been normalized for TCR recovery rates- if not this should be performed prior to quantification.

Thank you for the insightful comments. The data shown in Figures 7A-B are already normalized by differences in TCR recovery rate to mitigate for sample size differences, which is why the values are shown in percentages.

Based on the reviewer's suggestion, we use Fisher's exact test to test for independence between matched pairs of cell states (*i.e.* C4-to-C4, or C4-to-C6) across treatment groups (IL2-mutein versus isotype). Since Fisher's exact test is performed on observed data by using the hypergeometric distribution, we utilized the counts (not percentages) to determine statistical significance. We reasoned that comparing each matched pair of cell states provided a more accurate method to assess statistical differences than rank ordering tests on the mean of each population (*e.g.* Wilcoxon signed rank test), since IL-2 accelerates the differentiation/expansion of Tregs toward specific activated cell states that are inherent to their regulatory circuitry(14). Therefore, IL2-mutein treatment should not only increase the mean number of cell state pairs with shared clonotypes; it should also increase the frequency of specific pairs of cell states.

Again- a more robust test would be via diversity or clonality measurements for each experimental repeat.

Unless you know the clonality of the 2 data sets it is unclear if a simple % conversion is normalizing the data. If you say have lower diversity and more TCRs recovered from a data set then the relationship between the 2 data sets is non-linear.

We answered this above in response to the Minor point 18.

20. "Given that IL-2M increases clonal Treg expansion, we also examined how IL-2M influences the localization of CF Tregs by comparing the frequency of CF Tregs that were shared across tissues versus within the same tissue. While the percentage of inter-tissue CFs remained the same in both Isotype and IL-2M conditions (3.9% versus 4.2%, respectively), the percentage of CFs found only in one tissue was nearly doubled (3.0% versus 5.8%, respectively)." A figure should be referenced for the origin of these numbers.

Figure 7C was generated and statistics were calculated using Fisher's exact test to compare the counts between intra/inter-tissue clonal family clonotypes and singletons (non-clonal family clonotypes) for each treatment group.

OK

21. Figure 7C-E: The signature of the most differentiated cells is somewhat surprising- it seems as though the remaining cycling cells (Mki67hi) are also expressing markers of terminal differentiation (gzmb). Is it thought that the most terminally differentiated cells would be represented by highly proliferative cells?

Yes, we believe that the proliferation of Tregs coincides with the expression of terminal differentiation markers. The cells that are selected to reach a terminally differentiated state are recruited by the host immune system to carry out effector functions; this causes these selected cells to rapidly expand as they

differentiate in order to rapidly respond to host signals. Therefore, it makes sense that terminally differentiated cells are also expressing markers of cell proliferation.

OK

22. "As expected, C1-resting Tregs occupied earliest period of pseudotime."- Is the "resting" Treg subset being chosen as the starting place of the pseudotime or is this being determined in an unbiased way? If chosen one cannot make statements about the analysis "beginning" at resting Tregs.

As mentioned in the Monocle 2 paper (15), Monocle uses reversed-graph embedding to define a manifold that represents the structure of cell differentiation, but the package allows the user to define the root node (or root cells) based on an understanding of the underlying biology and calculates pseudotime in an unbiased manner based on the distance from that root node.

From the paper:

"Monocle 2 allows users to conveniently select a tip of the tree as the root and then transverses the tree from the root, computing the geodesic distance of each cell to the root cell, which is taken as its pseudotime, and assign branch or segment simultaneously."

This approach can be problematic of cell types of unknown etiology, but we believe this is not the case for Tregs. Given the unbiased approach in structuring the trajectory manifold, the following lines of evidence suggest that the root node was correctly selected:

1) Previous studies have shown that Tregs express secondary lymphoid organ-homing genes such as *Ccr7* and *S1pr1* prior to activation, suggesting that cells expressing these genes appear earlier in pseudotime (16,17). Tregs then downregulate these genes upon activation and extravasation into tissues.

2) Previous studies have all found genes such as *Tnfrsf9*, *Gzmb*, *Il1rl1*, *Il10*, and *Areg* to be involved in effector Treg functions (18,19). Furthermore, we show in this study that Tregs expressing *Tnfrsf9* and *Il1rl1* show more suppressive, effector activity than Tregs that do not express these genes, suggesting that cells expressing these genes should occur later in pseudotime.

We revised the text accordingly and it is highlighted.

"Given the gene expression profiles and robust lineage relationship of the C2-primed states and C3/C4/C8 activated states, we used pseudotime analyses using Tregs with recovered TCRs ($n=3,600$ cells) to define their developmental relationship (Figure 7, D and E, Supplemental Figure 12A). Treg cell states occupied distinct territories in pseudotime. We defined the node enriched for C1-resting Tregs as the root node (pseudotime $t=0$), and pseudotime values were assigned in an unbiased manner to the manifold based on the distance from that root node."

OK now. The revised statement is more transparently worded and clarifies this issue appropriately.

23. "At the latest periods in pseudotime/differentiation, we observed two distinct differentiation branchpoints consisting of C5-Tregs at one terminus and C4/C8 Tregs at the other (Supplemental Figure 12B)." - How are the data sets being integrated? This is not mentioned in the methods and integration methods can affect clustering/pseudotime analysis (see Efficient integration of heterogeneous single-cell transcriptomes using Scanorama, Nature Biotechnology, 2019).

Thank you for the comment and paper reference. We are acutely aware of the negative impact of integrating different datasets without proper sample integration. However, data integration prior to pseudotime analysis in Figures 7C-E was not necessary, since cells used for the analysis were from the same mouse. Therefore, they were prepared in the same library prep batch and by the same library prep method. After identifying the Treg cell differentiation trajectory in this batch of clonally related Tregs (confirmed by TCR sequence), we confirmed that this trajectory pattern was not unique to this batch by repeating the trajectory analyses for the remaining batches of Tregs from which we did not sequence the TCR (shown in Supplemental Figures 12C-D), and we observed the same manifold structure in all batches. We believe that if batch effects were significant across library prep batches and methods, the manifolds would appear drastically different between batches and/or methods.

We would like to note that data integration was performed prior to Louvain clustering analysis using the

CCA method in Seurat, since this analysis incorporated cells from three mice, different library prep batches, and two different library prep methods (10x 3' V2 and 10x 5'). This is already described in the Methods section.

OK

24. "Of the thirty most variant genes identified from this analysis, four major gene modules were identified that correspond to cell-state classifications (Figure 7E)." - What analysis is being used to produce the thirty most variant genes- is it the genes that are the most dynamic over pseudotime?

We used the differentialGeneTest() function in Monocle to test for genes that were the most dynamic over pseudotime:

```
differentialGeneTest(fullModelFormulaStr = "~sm.ns(Pseudotime)")
```

For clarification, we have added the following statement to the methods section:

Genes that varied the most along the pseudotime axis were determined using the Monocle function: differentialGeneTest(fullModelFormulaStr = "~sm.ns(Pseudotime)").

OK

25. "Overall, analysis of clonal Treg differentiation trajectories suggests IL-2M promotes differentiation into the terminally differentiated C4 and C8 Tregs state by expanding through C2 and C3 intermediate states in the spleen and lung." - In figure 5C, C3 is decreased in IL-2M treated mice while C2 is increased. From what data is this conclusion being drawn? Pseudotime analysis of isotype and IL-2M treated mice should be compared.

The Monocle analysis was used to understand differentiation trajectory of the Treg clusters, specially between the C2-primed state and C3/C4/C8- activated states. Based on this analysis, we found that the C4 and C8 Treg states, which were expanded by IL-2M, occupied the terminal state in the differentiation pseudotime. Meanwhile, the C2 and C3 were dispersed throughout the intermediate points in the pseudotime. Furthermore, C3-Tregs express immunomodulatory genes (Izumo1r, Nt5e) as well as TNFRSF9 and CD83 at a medium level without expression of effector proteins compared with other activated states. Based on this result, we concluded that the C3-Treg is an intermediate state- they are activated but not terminally differentiated.

However, Monocle analysis cannot distinguish which differentiation events are result of IL-2M treatment. We agree with the reviewer that, because C3-Tregs was decreased following IL-2M treatment, and there is no evidence that differentiation occurs through C3. Thus, we will remove C3 in the text and revise the statement to the following:

"Overall, analysis of clonal Treg differentiation trajectories suggests IL-2M promotes differentiation into the terminally differentiated C4 and C8 Treg state by expanding **through an intermediate state such as C2** in the spleen and lung."

It would still be informative to compare the pseudotime trajectories for IL2M vs isotype. Certainly, you would expect to see a different distribution of cells over the pseudotime trajectories.

The purpose of pseudotime was to look at Treg cell states following IL2M treatment in relationship to each other. The trajectory of isotype alone is beyond the scope of this paper.

26. Figure 7D: Could the scales be changed for each subset so that the distribution can be visualized more clearly? Many of the subsets are imperceptible.

More detailed distribution of each subset in Figure 7D (now Figure 7E) can be found in Supplemental Figure 12B.

OK

27. "Pseudotime analysis demonstrated that bifurcation of Treg differentiation leading into the C4/C8 activated state, showing enrichment of genes in the Module 3 (Figure 7E). Among those genes, Il1rl1 and Tnfrsf9 are highly expressed in the Module 3." - It is unclear what the authors are trying to say here, could this be clarified?

We revised to " Pseudotime analysis demonstrated enrichment of genes in 4 different Modules (Figure 7F). Among those genes, Il1rl1 and Tnfrsf9 are highly expressed in the Module 3."

It is still unclear what the importance of these genes in this context is- what point is trying to be made here? A sentence describing why you are highlighting these two genes would be helpful.

Il1rl1 and Tnfrsf9 were chosen as a marker for module 3, because we want to identify the cells expressing proteins in module 3 using FACS analysis. FACS agents recognizing protein expression of these two proteins were readily available. To clarify this, we added the following statement in pg 16.

"Il1rl1 or Tnfrsf9 encodes proteins ST2 or 4-1BB respectively and FACS reagents recognizing these proteins were readily available. To verify protein expression of these two genes, we performed Flow cytometry analysis."

28. Figure 8: Does ST2 or 41BB stimulation affect Treg suppressor capability or are these just markers for the most functionally suppressive Treg subsets?

We did not stimulate Tregs with ST2 or 41BB. These are used as surface markers to identify the majority of Tregs expanded by IL2M.

OK.

29. Figure 8B: The amount of ST2 and/or 41BB+ cells should be quantified over repeated mice.

We now added Figure 8C, which is quantification of Figure 2 from repeated mice. We revised the Figure legend accordingly.

This is a nice addition; however, all the individual data points should be visible in all bar graphs.

30. Figure 8C: Additional quantification should be performed by calculating "% divided" as a measure of Treg suppressor function. This experiment should be repeated as well given that only 2 data points are being compared.

We now provided Percent divided (%) in the right panel of Figure 8D. These experiments were repeated two times using sorted Treg populations from two individual mice. Figure legend was revised accordingly.

The figure legends say that this data has 2-3 mice per experiment and the experiment was performed twice. The exact n for each condition should be stated. All data points from all experiments should be displayed to indicate the biological variability in the experiment.

This experiment was performed twice. First experiment was performed using two IL-2M treated mice. Single cell suspension from two mice were combined and each population indicated was sorted. Treg suppression assay was set up in duplicate, thus there are two data points. This data is shown in Figure 8D.

We repeated this experiment for the second time using three mice and we will show this data in Supplemental Figure 14. Single cell suspension from three IL2M-treated mice was combined and each population was sorted. In vitro Treg suppression assay was performed in triplicate using sorted Tregs (Supplemental Figure 14A below). Because proliferation of Tconv is different in each experiment, adding all data points from all experiments will not be feasible. As you see below, Tconv in the absence of Tregs in the second repeat did not proliferate as much compared with the first experiment. However, ST2+41BB+ Tregs still showed superior suppressive capacity compared with ST2-41BB-Tregs. The difference between ST2+41BB+ vs ST2-41BB- population was also apparent when we performed *in vitro* Treg suppression assay under Th1 skewing condition (Supplemental Figure 14B).

We revised the legend of Figure 8D in pg 39 accordingly.

“(D) Fopx3-EGFP+ Tregs were stained based on ST2 and 4-1BB expression. Single cell suspension from two IL2M- treated mice was combined and four quadrants of Fopx3-EGFP+ Tregs were sorted based on ST2 and 4-1BB. *In vitro* suppression assays were performed in duplicate using four populations of sorted Tregs or without Tregs. Data were repeated independently using three mice and in triplicate (Supplemental Figure 14).”

Fig. S14

31. "Proliferation of effector T cells was suppressed by Tregs expressing either ST2 or 4-1BB, but ST2+4-1BB+ Foxp3+ Tregs displayed the most superior suppression (Figure 8C)"- It can only be said, from the data in 8C, that ST2+41BB+ are superior to ST2+41BB-. ST2+41BB+ are not statistically different from ST2-41BB+.

Although difference in dilution of CTV gMFI was not significant between ST2+41BB+ vs. ST2-41BB+ subsets, percent divided (%) of ST2+41BB+ was statistically different from ST2-41BB+, thus this statement is correct.

OK.

32. "Interestingly, the development of Klrp1+ Tregs requires extensive IL-2R signaling."- There is no reference for this statement (although presumably it links to ref 19?). "

Yes, the reference is ref 19.

OK

"In fact there are other studies that say eTregs are less dependent on IL-2 signaling (KLRG1 expression

identifies short-lived Foxp3+ Treg effector cells with functional plasticity in islets of NOD mice, 2017 Autoimmunity; CCR7 provides localized access to IL-2 and defines homeostatically distinct regulatory T cell subsets, 2014 JEM). “

This is the same point raised in Major point # 2 and we addressed there.

This is not really the same point as it is a different marker and different set of literature. Can you just confirm that Klrp1 is expressed in these clusters or induced by IL2M? In the supplement it appears as if Klrp1 is expressed somewhat higher in C3,4,5,8.

Klrp1 expression was high in C4, C5, C6 and C8, which are four clusters that are increased following IL2M treatment.

33.

“Trajectory analysis also identifies a bifurcation in Treg differentiation after IL-2M treatment, which either differentiate into suppressive Il10+Rora+ C5 Tregs, which are most prevalent in the gut, or into C4/C8 Tregs that are prominent in the spleen and lungs.”- Is this bifurcation dependent on IL-2M treatment? If so, this data is not being displayed.

At the latest periods in pseudotime/differentiation, we observed two distinct differentiation branchpoints consisting of C5-Tregs at one terminus and C4/C8 Tregs at the other. The pseudotime analysis can show developmental relationship of Treg states and trajectory of differentiation but cannot distinguish whether differentiation is dependent on IL-2M. Thus, we revised the statement to:

“Trajectory analysis also identifies a bifurcation in Treg differentiation after IL-2M treatment, which either differentiate into suppressive Il10+Rora+ C5 Tregs, which are most prevalent in the gut, or into C4/C8 Tregs that are prominent in the spleen and lungs.

The trajectory analysis is still not determining anything about the IL2M treatment- this statement is still just based in the proportional changes between clusters in the IL2M treated mice? If so this should be revised again for accuracy to mention that the trajectory analysis does not inform us about whether IL2M actually directs this bifurcation – just that it could.

We added the text in pg 23 and shown in green highlight.

“However, the trajectory analysis does not inform about whether IL2M actually directs this bifurcation.”

34. Supplemental Figure 1: Is the number of CD8+ T cells not different between PBS and IL-2M?

We performed One-Way ANOVA for multiple comparisons using GraphPad Prism 7.04, and the numbers CD8+ T cells were not statistically significantly different between PBS and IL-2M.

This data should be included in the supplemental figure as displayed previously; all we wanted was for you to include p-values of the multiple comparisons. This is directly relevant and should not be removed.

We included the data in the new Supplemental Figure 2 because there was no space to include all data to the Supplemental Figure 1.

The number of Tregs being detected also seems very low- what organ is this? Spleen?

It is spleen. To calculate the cell numbers, we used percentages of CD4 or CD8 T cells from total cells. For this particular experiment, percentages of live cells within spleen were around 70% among all samples, which resulted in overall smaller numbers of cells. If we use percentages of CD4 or CD8 T cells from live lymphocyte gate to calculate the cell number, the overall numbers are increased, but the trend remained the same. Although it is interesting to see differential effect on CD8, it is not relevant for the current study. Thus, we removed the CD4 and CD8 data and revised Supplemental Figure 1.

A normal spleen likely contains 2-4 million Tregs whereas the figure shows maybe a couple hundred thousand (potentially off by a factor of 10). It should just be verified that the numbers are being calculated appropriately.

A spleen from healthy mice contains around 60-100 millions of total cells and among them, there are about 10 millions of live CD3+CD4+ T cells in average. Tregs comprised of 5-10% of Tregs, thus leaving 0.5-1 millions of Tregs from one spleen. Our initial calculation was correct, but total live cells were substantially low due to poor viability of that particular experiment. Thus, we repeated this experiment again using three mice for each condition. Now, Treg numbers of Tregs from this experiment showed normal range of the live cells from the spleen. Overall, trend of experiment showing as a percentage was similar to the previous experiment. Using this data, we generated new Supplemental Figures. Because there was no room in the Supplemental Figure 1 to show all of the data, all of these data are now shown as Supplemental Figure 2.

35. Supplemental Figure 4: Perhaps similarity in differentially expressed genes could also be shown here to bolster the argument for data set similarity.

The clusters in Supplemental Figure 4 were determined after integrating all replicates together; therefore, differential expression by each replicate would be redundant and unnecessary. A more robust way to test for similarity between replicates is by testing whether the observed cell frequencies in each cell state are significantly different between replicates. Thus, we applied Fisher's exact test to compare for significant differences in cell state frequencies for each replicate and found that there were no significant differences. These results have been added to Supplemental Figure 4.

Clusters shouldn't be changed by experimental variance, but gene expression could be. The more robust way to ensure that the datasets are reproducible is to show that the same sets of genes are being differentially regulated in each dataset. This analysis should be performed.

While gene expression may vary slightly between replicates due to technical differences sample preparation and sequencing depth, the utilization of batch correction tools such as the CCA algorithm in the Seurat package (Butler, Nature Biotech. 2018) are well-established in the single cell field for harmonizing those technical differences across replicates. The similarity in cell cluster distributions across replicates (which we have already shown) demonstrates this better than differential expression testing, since (1) in single-cell transcriptomic data, the single cell is the major unit for interrogation (in bulk transcriptomic data, it would be more appropriate to do DE testing by batch, since single-cell granularity doesn't exist in bulk); (2) clustering accounts for the variance in expression from multiple genes, while differential expression testing only accounts for single-gene differences per test.

36. Supplemental Figure 5A: C2 visually looks quite similar to C1 and C3 looks like C5. C6 also looks like it is a proliferating cluster of C4. It seems as though this data is perhaps over-clustered and producing

signatures which are not truly unique. Also, C10 expresses IFN γ and CD8 α - could these be contaminating CD8 cells? A better description of how the resolution was chosen to establish 10 clusters is needed.

We addressed this concern about resolution in Major Point #1.

C2 is different from C1 in the sense that they weakly expressed genes that are enriched in C1-resting as well as other activated Treg clusters. Furthermore, Monocle analysis showed the C2-Tregs are dispersed through the manifold as opposed to the C1-resting cluster occupied at the initial starting point.

Moreover, we don't believe the C10- cluster is contamination, because there are other reports demonstrating a small set of CD8+ Tregs that has suppressive effect for the self-reactive CD4 T cells(20,21) In particular, it was reported that a small population of CD8 \cdot CD25 $^{+}$ FOXP3 $^{+}$ T cells was found both in mice and humans and they can suppress CD4 effector T cell proliferation(21).

This would still constitute contamination given that the cells for this experiment were sorted on being CD4 $^{+}$ CD8 $^{-}$. It would be relevant to verify that this cluster does express Foxp3 as well as additional suppressive genes related to CD8 Tregs.

We showed expression of Foxp3, Helios and CD25 in page 2.

37. Supplemental Figure 5B: These comparisons seem somewhat arbitrary. Could some of these differences between clusters be derived from differences in tissue distribution?

We would like to see differential gene expression regardless of the location within the cluster. Some clusters are shown in all organs, but some clusters are over-represented by an organ. Differences in tissue distribution were shown in Figure 4.

Yes- but the gene expression patterns for a given cluster could be different based on the ratio of spleen/lung/gut in 1 cluster versus another cluster. These comparisons should be made within a particular tissue to rule out this explanation for differences between clusters.

This is an unnecessary request that is more fitting for bulk RNA-seq data (where gene expression is heavily influenced by the ratio of cell types per sample and there is no way to account for this). In the case of single-cell RNA-seq data, each single cell serves as its own 'sample' for cluster assignment, so the gene expression of a cluster is not driven by number of cells from a tissue. Cells were assigned to the specific cluster *a priori* because they exhibit similar gene expression patterns together with other cells assigned to that cluster regardless of tissue site. The purpose of Supplemental Figure 5B is to understand differential expression of genes of specific clusters compared with C1 resting state, or between specific clusters, to better understand the uniqueness of the cluster compared with other cluster within all Tregs, not within specific tissue Tregs.

38. Supplemental Figure 8&9 are titled the same thing thus these could be combined into 1 figure.

Combining two figures will make figures even smaller and it will be difficult to read each gene.

This data would probably be better displayed in the form of a table with exact fold changes and p-values.

We believe volcano plot is better representation of the data to show the p value and fold changes at the same time. Original comment was about the titles of Supplemental Figure 8&9 are the same. Thus, we changed the title of Supplemental Figure 8&9 accordingly. The changed titles are shown in green highlight.

39. Supplemental Figure 10: Given that splenic and lung Tregs look almost identical, is there a way to validate that true lung, non-circulating, Tregs were used for the single cell analysis?

To take a look at differential expression between lung and spleen, we need to take a look at Figure 4B not Supplemental Figure 10. Figure 4B showed more genes differentially expressed between spleen and lung following Isotype control treatment. Supplemental Figure 10 showed differential genes between spleen and lung following IL-2M treatment.

Figure 6A upper right graph showed some of the TCR shared between spleen and lung (inter-tissue clonotypes). These Tregs with shared TCR in the lung would be circulating Tregs. If the lung Tregs are contamination of all circulating Tregs, the composition of each cluster should be similar with spleen Tregs. However, the lung Tregs are quite different from spleen Tregs (for example, the lung Tregs lack the C3-Treg cluster).

Where are the figure legends for the supplemental figures here?

We provided Figure legend in the previous revision and I am not sure what the reviewer is asking. Previous Supplemental Figure 10 is now Supplemental Figure 11 for the current revision and the legend is shown below.

Supplemental Figure 11. Transcriptional profiles converge in the spleen and lung after IL2-mutein treatment, while gut retains a distinct transcriptional profile from the spleen and lung.

(A) Comparison of Treg signature genes between all Tregs in the **(A)** spleen vs lung, **(B)** spleen vs gut, and **(C)** lung vs gut after IL-2M treatment. Significant genes are shown in red (adjusted p-value < 0.05, MAST).

40. Supplemental Figure 11: Would this analysis be more statistically accurate if the "different clonotype" group had the same number of events as the "Same clonotype" group.

We believe the analysis performed using a larger number of samples represents a more accurate approach, since more replicates are sampled. 10,000 randomly sampled correlations were used in the "different clonotype" group to represent an exceedingly large sampling of that group and remove any ambiguity about sufficient sampling depth. This approach is similar to that used by Zemmour, Nat Imm, 2018 (22), which compares 14 "same clonotypes" to 1000 randomly sampled "different clonotypes". To explore this analysis further, we also performed statistical analysis between equal numbers of events in both groups and find that the results are the same as the analysis shown in Supp. Figure 11.

Where is this data- the only data presented is the previous analysis with 10,000 correlations. If the results are the same then certainly the analysis with equal numbers represents a more powerful finding and should be mentioned in the text.

We performed this correlation analysis with equal numbers of TCR clones and obtained a p-value of 2.106E-4. We amended the text to include this analysis in pg 13 and highlighted in green:

"The identification of transcriptional diversity among Tregs from the same clonal family was an interesting result, since we also find that pairs of T cells belonging to the same clonotype tend to be transcriptionally correlated than randomly sampled pairs of Tregs at the population level ($p=2.106E-4$ when comparing equal numbers of cells from same and different clonotypes), although this was a modest effect, as previously reported (Zemmour et al, 2018) (Supplemental Figure 11)."

41. The exact number and description of single cell library preps for each sample needs to be defined for this study. A supplementary figure or table showing this would be informative. This figure should also show what libraries were then integrated for further analysis. Please include the methods that were used to integrate (ex. Suerat CCA + version, ScanPy aggregation method) - a detailed description of the pipeline would also be a very helpful supplemental figure. There are two different sequencing machines mentioned in the methods - HiSeq4000 and NovaSeq 6000 - please identify what libraries were sequenced on what machine.

We have included an excel spreadsheet table as a Supplemental Table for the number of preps, sample processing batches, library prep methods, and sequencing instruments. Moreover, we will add information about the Seurat version (2.4) and Scanpy version (0.94) to the Methods section as well.

OK

42. In Figure 7C, you show pseudotime/trajectory analysis of cells from both IL2M and isotype conditions. Why are you not showing these conditions separately as well? I am not sure that the data shown is supporting the statement on page 14 "Overall, analysis of clonal Treg differentiation trajectories suggests IL-2M promotes differentiation into the terminally differentiated C4 and C8 Tregs state by expanding through C2 and C3 intermediate states in the spleen and lung".

The clusters were defined by combining both IL-2M and isotype control. By doing so, we could see which clusters are increased by IL-2M treatment. Because we define clusters based on all possible clusters of genes under IL-2M and isotype control, we can define which cluster is increased following treatment. Figure 7 is performed by TCR analysis and Supplemental Figure was performed using all T cells. For the statement, we already discussed in Minor Point #22.

Separate analysis of each condition is still warranted here. Are there TCR's found in both isotype and IL2M treated mice? If so you could use these to bolster claims regarding the shifting of a phenotype following treatment. It is likely these would be quite rare events though.

We did not find any clonotypes shared between isotype and IL2M treated cells.

In summary, these studies will be of interest and the major conclusions are largely supported by the data but there are many issues (many but not all of which are relatively minor) that should be addressed.

The authors did well to bolster various aspects of the manuscript including a more thorough description of the IL2M reagent as well as more clear explanations of the methods used for the single-cell RNAseq and sequencing. However, the report lacks basic mechanistic insight on how IL2 treatments are changing Treg functionality which could be easily discerned from simple experiments. There remain a number of unaddressed concerns regarding the direction of single-cell RNAseq analysis as well as graphical display of repeat data points. These concerns are outlined in the point by point response.

March 24, 2020

RE: Life Science Alliance Manuscript #LSA-2019-00520-TRR

Dr. Hyewon Phee
Amgen Research, Amgen Inc
Department of Oncology and Inflammation
1120 Veterans Blvd
South San Francisco, CA 94080

Dear Dr. Phee,

Thank you for submitting your revised manuscript entitled "Dynamic changes in the regulatory T cell heterogeneity and function by murine IL-2 mutein". I appreciate the introduced changes and would thus be happy to publish your paper in Life Science Alliance pending final revisions necessary to meet our formatting guidelines:

Please add a "data availability" section in the Material & Method part of your manuscript to provide the reader with the accession code for the deposited RNA-seq data. Please also add the information on the repository used and make sure that the data are accessible.

A. FINAL FILES:

-- Summary blurb (enter in submission system): A short text summarizing in a single sentence the study (max. 200 characters including spaces). This text is used in conjunction with the titles of papers, hence should be informative and complementary to the title. It should describe the context and significance of the findings for a general readership; it should be written in the present tense

and refer to the work in the third person. Author names should not be mentioned.

B. MANUSCRIPT ORGANIZATION AND FORMATTING:

Sincerely,

March 31, 2020

RE: Life Science Alliance Manuscript #LSA-2019-00520-TRRR

Dr. Hyewon Phee
Amgen Research, Amgen Inc
Department of Oncology and Inflammation
1120 Veterans Blvd
South San Francisco, CA 94080

Dear Dr. Phee,

Thank you for submitting your Research Article entitled "Dynamic changes in the regulatory T cell heterogeneity and function by murine IL-2 mutein". It is a pleasure to let you know that your manuscript is now accepted for publication in Life Science Alliance. Congratulations on this interesting work.

*****IMPORTANT:** If you will be unreachable at any time, please provide us with the email address of an alternate author. Failure to respond to routine queries may lead to unavoidable delays in publication.*******

DISTRIBUTION OF MATERIALS:

Again, congratulations on a very nice paper. I hope you found the review process to be constructive and are pleased with how the manuscript was handled editorially. We look forward to future exciting submissions from your lab.

Sincerely,
